# Phase matrix characterization of long-range transported Saharan dust using multiwavelength polarized polar imaging nephelometry

Elena Bazo[1,2], Daniel Pérez-Ramírez[1,2], Antonio Valenzuela[1,2], J. Vanderlei Martins[3], Gloria Titos[1,2], Alberto Cazorla[1,2], Fernando Rejano[4], Diego Patrón[1,2], Arlett Díaz-Zurita[1,2], Francisco José García-Izquierdo[4], David Fuertes[5], Lucas Alados-Arboledas[1,2] and Francisco José Olmo[1,2]

[1]Andalusian Institute for Earth System Research (IISTA-CEAMA), Granada 18006, Spain

[2]Department of Applied Physics, University of Granada, Granada 18071, Spain

[3]JDepartment of Physics and Earth and Space Institute, University of Maryland, Baltimore County, Baltimore, Maryland, USA

[4]Instituto de Astrofísica de Andalucía (IAA-CSIC), Granada 18008

[5]GRASP-SAS, Remote Sensing Developments, Lille, France

*Correspondence to*: Daniel Pérez-Ramírez (dperez@ugr.es)

**Abstract.** This work investigates the scattering matrix elements during different Saharan dust outbreaks over Granada (South-East Spain) in 2022 using the Polarized Imaging Nephelometer (PI-Neph PIN100, GRASP-Earth). The PI-Neph is capable of measuring continuously the phase function ($F_{11}$) and the polarized phase function ($-F_{12}/F_{11}$) at three different wavelengths (405, 515 and 660 nm) in the range 5° - 175° with 1° resolution for ambient aerosol samples. Extreme dust events ($PM_{10} > 1000$ µgm$^{-3}$) occurring in March 2022 are compared with more frequent and moderate events registered in summer 2022 ($PM_{10}$ between 50 and 100 µgm$^{-3}$). These intercomparisons allow the evaluation of $F_{11}$ and $-F_{12}/F_{11}$ when dust particles predominate in the aerosol sample, but there is a possible mixture with other anthropogenic particles. For $F_{11}$ there are no remarkable differences between extreme and moderate events. However, results of $-F_{12}/F_{11}$ show large differences between extreme and moderate events: for 660 nm the $-F_{12}/F_{11}$ pattern is characterized by a bell-shape with a positive maximum in the 90°-120° scattering region, and this pattern is observed both in the extreme and moderate dust events. However, there are remarkable differences in $-F_{12}/F_{11}$ at 405 nm showing a very similar pattern with 660 nm during the peaks of the extreme dust events while for moderate events it shows a different pattern characterized by values around zero up to ~ 50°, decreasing later to negative values ~120° and increasing again to values close to zero in the backward scattering region. For 515 nm we found out intermedia patterns. The temporal evolutions during the extreme dust events reveal that $-F_{12}/F_{11}$ is very sensitive to the particle concentration at 405 nm. For the peak of the events, $F_{11}$ and $-F_{12}/F_{11}$ agree with the laboratory measurements available in the Amsterdam-Granada Light Scattering database at all wavelengths. The combination of PI-Neph measurements with additional in-situ instrumentation allowed to obtain scattering (SAE) and absorption (AAE) Ångström exponents and to conduct a typing classification that revealed extreme dust events as pure dust, while moderate dust events were classified as a mixture of dust with urban background pollution. In addition, simulations with the Generalized Retrieval of Atmosphere and Surface Properties (GRASP) code explain the different patterns in $-F_{12}/F_{11}$ with changes in the refractive indexes and the contributions of the fine and coarse mode. Therefore, our results confirm that differences in the phase matrix elements of Saharan dust outbreaks of varying intensity can be explained by the mixing conditions of dust with the background particles.

## 1   Introduction

The imprecise determination of atmospheric aerosol microphysical properties is currently the main source of uncertainty in climate projections, as stated by the latest Intergovernmental Panel on Climate Change (IPCC – Forster et al., (2021)). Particularly, aerosol particles can scatter and absorb solar radiation, known as the direct effect (Haywood et al., 2000). Moreover, aerosol particles can interact with clouds in different ways: aerosol absorption can modify the energy balance in the atmosphere affecting cloud development

and properties (the semi-direct effect – Fan et al., (2016)). They can also serve as cloud condensation nuclei (CCN) and ice nucleating particles (INP) upon which cloud droplets and ice crystals form (the aerosol indirect effect on clouds – Rosenfeld et al., (2014)).

Advancing in aerosol knowledge faces complex challenges due to the large variability of aerosol types and to aerosol sources and transformation processes in the atmosphere during their transport. More specifically, mineral dust is the most important source of primary particles in the atmosphere with an estimated emission rate of 1000-3000 Tg/year representing about half of the annual particle mass emission at the global scale (Kok et al., 2021). Mineral dust sources extend over a wide area on the planet highlighting the global dust belt that extends from the arid regions of the west coast of northern Africa through the Middle East and Central Asia. Such a belt includes the Sahara Desert, which is the largest in the world being responsible for almost 50% of the global dust emissions (Kok et al., 2017). In this sense, recent studies (Kok et al., 2018) estimate the direct dust-climate feedback parameter associated with the direct radiative effect in the range -0.04 to + 0.02 Wm$^{-2}$°C$^{-1}$ (net, short + long wavelength), but being highly dependent on the model used. The problem with understanding the role of dust in climate becomes even more complex due to the changes in arid lands since the pre-industrial era, which is producing an increase of global dust mass loading (Kok et al., 2017). Most of these uncertainties are due to the challenges in better understanding mineral dust composition and variability with size and sources (Gonçalves Ageitos et al., 2023).

Mineral dust particles are typically considered as large particles in the coarse (1-10 µm) and super-coarse (> 10 µm) modes (J. B. Renard et al., 2018), although recent studies have also shown the presence of a fine mode (ranges below 1 µm diameter) in mineral dust (Huang et al., 2019). The current discrepancies about the roles of fine, coarse and super-coarse modes in the dust sample (González-Flórez et al., 2023) imply difficulties in dust modelling that add uncertainties to the climate modelling (A. A. Adebiyi & Kok, 2020). One critical point is the modelling of coarse mode because of the non-sphericity of these types of particles (Mishchenko et al., 2002), and also inferring the complex refractive index (Formenti et al., 2003) that ultimately depends on particle size, shape and chemical composition (González-Romero et al., 2023). For example, iron oxides are the key to understanding the mineral dust absorption properties in the UV (e.g. hematite, goethite) whilst Ca-rich carbonates become important in the infrared region (Formenti et al., 2014). These variabilities in size-related and absorption parameters difficult modelling accurately the response of mineral dust to direct radiative effect (A. Adebiyi et al., 2023). The problem becomes even more complex because the interactions of dust particles with other precursor gases and aerosol particles already presented in the atmosphere (Ooki & Uematsu, 2005). For example, the variabilities in dust size, shape and chemical composition are also related to emerging questions such as the role of big mineral dust particles in new particle formation in the atmosphere (Casquero-Vera et al., 2023).

Remote sensing techniques are widely used to infer dust properties. For example, passive remote sensing techniques such as sun-photometry by the Aerosol Robotic Network (AERONET – (Holben et al., 1998)) or star/moon photometry (i.e Pérez-Ramírez et al., 2008, Pérez-Ramírez et al., 2011;Berkoff et al., 2011) allow to have a representation of column-integrated values, particularly aerosol optical depth (AOD). But to infer other aerosol optical (e.g. aerosol complex refractive index and single scattering albedo) and microphysical (e.g. aerosol size distribution) properties it is necessary to solve ill-posed problems where the information content is low (Dubovik & King, 2000; King et al., 1978; Nakajima et al., 1996; Olmo et al., 2006, 2008; Pérez-Ramírez et al., 2015). These algorithms use the Mie theory for the internal computation of particles phase functions, but in the case of dust particles more complex approaches such as T-Matrix are needed because of the non-sphericity of dust particles (Mischenko & Travis, 1994, 1997) . Nevertheless, several inversion algorithms have been developed incorporating T-Matrix modeling, being one of the most popular algorithms developed within the AERONET network (Dubovik et al., 2006).

Ground-based remote sensing techniques are only representative of the measurement site, and to face these limitations satellite measurements are ideal because they can cover wide regions of the world. However, passive remote sensing space platforms deal with additional complexity in the retrieval of aerosol properties because of the influence of surface reflectance (Kahn et al., 1998; Levy et al., 2007). The simplest retrievals use look-up tables with a priori aerosol types with great success in obtaining AOD, but limited capacity for obtaining other aerosol parameters because of the difficulties to separate the signals corresponding to the atmosphere and surface (Dubovik et al., 2019). To solve these limitations, the use of multiwavelength and multi-angle polarization measurements is ideal to improve the information content (Mishchenko et al., 2007). Some of the first polarized-based measurements for aerosol studies were carried

out by the POLDER instrument (POlarization and Directionality of the Earth's Reflectances – (Deuzé et al., 1993)) that acquired 9 years of data. These measurements were used as inputs in the Generalized Retrieval of Atmosphere and Surface Properties algorithm (GRASP –(Dubovik et al., 2014, 2021)) for obtaining extended aerosol optical and microphysical properties. Algorithms such as GRASP are becoming the operational algorithms in new satellite missions (Remer et al., 2019; Fuertes et al., 2022; Hasekamp et al., 2024), but these algorithms need phase matrix measurements that allow to optimize the kernels used internally, particularly for non-spherical particles.

The main difficulties lie in measuring aerosol phase matrix of ambient air are in the design and development of appropriate polar nephelometry capable of measuring light scattered with appropriate angular resolution. The first polar nephelometry developments were based on moveable detectors, but they must be mechanically stable and require a constant population of aerosol particles that does not change appreciably during the detector sweep (Holland & Gagne, n.d.; Hovenier et al., 2003; Jaggard et al., 1981; Kuik et al., 1991; Perry et al., 1978; Volten et al., 2001a). Other polar nephelometry designs use arrays of many detectors placed on representative scattering angles (Barkey et al., 1999; Gayet et al., 1998; Pope et al., 1992; West et al., 1997; Wyatt et al., 1988), but this technique requires careful calibration of the detectors and generally suffers from low angular resolution (~2°). Those instrumental limitations have implied that the usual study of scattering matrix elements of dust particles is done in the laboratory for synthetic samples minerals that compose dust particles (Curtis et al., 2008; Huang et al., 2020; Meland et al., 2010; Muñoz et al., 2010a; J. B. Renard et al., 2014; J.-B. Renard et al., 2010) or with collected dust samples (Muñoz et al., 2007a; J. B. Renard et al., 2014; J.-B. Renard et al., 2010, 2024). Actually, the parametrizations of mineral dust phase matrix used for AERONET algorithm were calculated by fitting the laboratory measurements of different non-spherical particles samples (i.e. Dubovik et al., 2006). Such measurements were performed at a few wavelengths, and what is more important, they might be non-representative of real aerosol measurements because of the different transformations and interactions of dust particles since they were emitted in their source regions. There is therefore a current challenge in having an extended database of measurements of dust phase matrix elements for different dust types and mixtures, particularly at different stages of dust evolution after their emission from the remote desert areas.

The latest developments use imaging techniques (Bian et al., 2017; Curtis et al., 2007; Dolgos & Martins, 2014) to determine phase matrix with single detector and relatively compact design that does not require moveable parts. The Polarized Imaging Nephelometer (PI-Neph) was one of the first designs of a polar nephelometer that used imaging techniques, developed by the University of Maryland, Baltimore County (UMBC). This first prototype of the PI-Neph was capable of acquiring aerosol phase matrix at 473, 532 and 671 nm with 0.5° resolution. The instrument was deployed on the NASA DC8 aircraft and operated during special field (Espinosa et al., 2018; Reed Espinosa et al., 2017). Other PI-Neph instruments based on the first UMBC design are operated by NOAA (Ahern et al., 2022; Manfred et al., 2018). The main novelty of these prototypes is that they measure phase matrix elements of ambient air, where conditions can be very different to laboratory measurements. However, to date none of these instruments have been operating continuously and reported any multiwavelength measurements of Saharan dust. The imaging technique is being expanded worldwide with further designs although limited to laboratory operation yet (Moallemi et al., 2023). All designs in polar nephelometry present physical limitations that limit the measurements to the range 3°-178°, but synthetic tests have revealed that multi-wavelength polarimetric PI-Neph measurements improve the information content for the retrieval of aerosol optical and microphysical properties (Moallemi et al., 2022) . Therefore, measurements of dust phase matrix elements for ambient aerosol samples in the atmosphere will serve to advance in the understanding of mineral dust absorption properties and chemical composition (Di Biagio et al., 2017, 2019).

This work presents phase matrix measurements of ambient Saharan dust particles by the GRASP-Earth's (https://www.grasp-earth.com/) multi-wavelength PI-Neph. The instrument was developed using the heritage of previous PI-Neph developments made by UMBC and can provide aerosol phase matrix elements at 405, 515 and 660 nm of ambient samples in the range 5° - 175° with 1° resolution. Measurements were acquired in the urban background station (UGR) of the Andalusian Global ObseRvatory of the Atmosphere (AGORA) located in the Southeast of the Iberian Peninsula where the main source of natural particles is the Sahara Desert's transported particles (Querol et al., 2019). We present the results for extreme outbreaks that occurred in March 2022 (Rodríguez & López-Darias, 2024) with $PM_{10}$ (particulate matter with diameter < 10 µm) concentrations over 1000 µgm$^{-3}$, and for more typical situations of moderate dust events with

PM$_{10}$ concentrations around 100 µgm$^{-3}$. The measurements presented of the phase matrix for Saharan dust are unique and are and step forward from the ancillary measurements performed in the region by Horvath et al., (2018) with a single wavelength polar nephelometry (no polarization was available).

This work is structured as follows: in Section 2 we describe the experimental station and the instrumentation, Section 3 gives an overview of the extreme dust events, Section 4 analyzes the results of the optical properties during different dust events, in Section 5 we discuss the results obtained and Section 6 is devoted to the main conclusions and keys for future work.

## 2  Experimental station and Instrumentation
### 2.1  Experimental station

Experimental measurements were carried out at the UGR station of the AGORA Observatory in the city of Granada (37.18°N, 3.58°W, 680 m asl) in Southern Spain. The main local source of aerosol particles in the UGR station is road traffic (Titos et al., 2014, 2017), with sporadic presence of biomass burning aerosol (Casquero-Vera et al., 2021; Titos et al., 2017). Air-mass stagnation also favors the accumulation of pollution (Lyamani et al., 2012; Patrón et al., 2017). The city is located 200 km away from the African continent, so long-range transport of Saharan dust to the UGR station is quite common (Lyamani et al., 2010b; Valenzuela et al., 2012a, 2012b). These dust intrusions have mean aerosol optical depths (AOD) of 0.25 ± 0.12 (Pérez-Ramírez et al., 2016) and PM$_{10}$ concentrations ranging between 25 and 200 µgm$^{-3}$ (Párraga et al., 2021), although extreme Saharan dust events with AOD above 1.0 also affect the station (Guerrero-Rascado et al., 2009). Most of these intrusions typically occur in summer but in the last years they are becoming more frequent during the winter season (Cazorla et al., 2017; Cuevas-Agulló et al., 2024; Fernández et al., 2019; Titos et al., 2017).

In this study, particles were sampled using a total inlet (no size cut) that consist of a 5 m long stainless-steel tube with a 20 cm diameter (Lyamani et al., 2008). Inside the stainless-steel tube there are several pipes that split the aerosol flow into the different instruments. The inlet system is completely vertical to minimize deposition losses. The final connection to the instruments is performed with conductive tubing avoiding bends. Additionally, all the measurements were at ambient conditions (no aerosol dryer was used). Given the flowrate used in the measurements we can assume that particles are randomly oriented, avoiding the limitations in polarization results of super-coarse particles with particle speed that can orient the particle in particular orientations (Daugeron et al., 2006).

### 2.2  Instrumentation
#### 2.2.1    Polarized Imaging Nephelometer (PI-Neph)

The PI-Neph (GRASP-Earth PIN-100) is used to obtain direct measurements of two aerosol phase matrix elements, the phase function (F$_{11}$) and the polarized phase function (-F$_{12}$/F$_{11}$), at three different wavelengths (405, 515 and 660 nm). The instrument uses previous heritage in PI-Neph developments in the University of Maryland, Baltimore County (Dolgos and Martins, 2014), where the novelty in the PIN-100 is the use of one beam instead of a mirror system to fold the laser beam, as was in previous models. This feature minimizes internal reflection and loss of energy within the laser beam and guarantees that all points along the laser beam will have the same scattering plane orientation, assuring the optimal input polarization state at all scattering angles simultaneously. The optical system counts with a wire grid polarizer and two liquid crystal retarders (LCVRs) that control the state of linear polarization. In this sense, polarized light (parallel or perpendicular) reaches the sample chamber. The light scattered by the aerosol particles is recorded with a 185° field of view CMOS camera, giving the scattered light by the particles in the sample chamber in the range 5°-175°, with 1° angular resolution. More details of the instrument are in Bazo et al., (2024).

An extensive analysis of the error sources in the PI-Neph was performed in Bazo et al., (2024) but an overview is given here: an exhaustive calibration of the instrument is performed consisting of two different steps.  The first is a geometric correction that corrects from the different light paths to the different pixels in the CMOS camera. Later the absolute calibration permits to obtain phase matrix elements in physical units. In each step we used known scatterers (CO$_2$ and particle free air) whose parallel and perpendicular signals can be computed analytically using the Rayleigh theory (Anderson et al., 1996). Evaluation of the calibration with time did reveal great stability (variations around 3%). Instrument stability was evaluated with CO$_2$ measurements at a constant flow rate of 10 Lmin$^{-1}$ during 15 min. These measurements revealed

constant values of scattering coefficients with differences below 1% versus theoretical values from (Bodhaine et al., 1991). Finally, inherent aspects of the imaging technique were evaluated such as the impact of the exposure time. The largest noise is found for exposure times below 5 s, while the smoother values are obtained for exposure times of 10-20 s. However, large exposure times can yield to more angles that are saturated, and the software must find a compromise between noise and saturation. Thus, the typical exposure time is of 10 s and with that we estimate that uncertainties in measured parallel and perpendicular signals are around 5% in laboratory conditions. The evaluation of the instrument versus known scattered (monodisperse polystyrene spheres - PSL) showed good agreements, being the RMSE around 0.10 for both $F_{11}$ and $-F_{12}/F_{11}$.

The uncertainties in direct measurements of the instrument (parallel and perpendicular signals) are 5% that imply uncertainties below 10% in $F_{11}$ and below 20% in $-F_{12}/F_{11}$. However, in-situ measurements present natural variability of the aerosol sampled and the differences can be enhanced because of the short exposure times (~ 10s). Effects during the measurements such as saturation or low signal to noise ratios (SNR) of some pixels can happen. Other issues such as the passage of an individual super-coarse particle can have an impact on certain angles of the phase matrix. Therefore, we apply a data-quality check procedure that accounts for all these issues and provide an effective phase matrix representative of an average time of 30 min or 1 hour, depending on the specific conditions of natural aerosol variability. Note that standard deviations during these periods might be larger than the uncertainties of the instrument. Details of this quality check procedure are in Bazo et al., (2024).

The direct measurements of $F_{11}$ and $-F_{12}/F_{11}$ with the PI-Neph allow to obtain other aerosol optical parameters such as the scattering coefficient ($\sigma$), the asymmetry parameter (g) and the fraction of backscattered light (Bs) using the following equations (Horvath et al., 2018):

$$\sigma_{sca}(\lambda) = \frac{1}{2} \int_0^{180} F_{11}(\theta, \lambda) \sin \theta \cdot d\theta \ , \tag{1}$$

$$P_{11}(\theta, \lambda) = \frac{F_{11}(\theta, \lambda)}{\sigma_{sca}(\lambda)} \ , \tag{2}$$

$$g(\lambda) \ = \ \int_0^{\pi} P_{11}(\theta, \lambda) \cdot \sin \theta \cdot \cos \theta \cdot d\theta \ , \tag{3}$$

$$B_s(\lambda) = \frac{1}{2} \int_{\frac{\pi}{2}}^{\pi} P_{11}(\theta, \lambda) \cdot \sin \theta \cdot d\theta \ , \tag{4}$$

where data from 0 to 5° and from 175 to 180° have been linearly extrapolated to obtain the complete phase function. Stepwise extrapolations might be more consistent (i.e. Horvath, 2015), but our additional computations remarked that differences between linear and stepwise extrapolations were below 1% for $\sigma_{sca}$, g and $B_s$. For particle sizes above 4 µm the uncertainties can yield 5% (Horvath, 2015).

### 2.2.2 Additional in-situ instrumentation at AGORA

AGORA operates other in-situ instruments within the Aerosols, Clouds and Trace gases Research Infrastructure (ACTRIS; https://www.actris.eu/). The integrating nephelometer (TSI model 3563) was used to measure the aerosol particle light scattering coefficient ($\sigma_{sca}$) at 450, 550 and 700 nm with a flow rate of 15 Lmin$^{-1}$ and a time resolution of 1 minute. As with the PI-Neph, the integrating nephelometer is calibrated with particle free air and $CO_2$ and Rayleigh subtraction is applied to measure particle scattering only. Due to experimental limitations the scattered light at the complete forward (0°) and backward (180°) regions cannot be detected, so the angular range for integration is 7-170°. However, results used in this work have been corrected to the entire angular range with the correction proposed by Anderson & Ogren, (1998). On the other hand, we also used the multiwavelength aethalometer (AE33, Magee Scientific) that measures aerosol light absorption coefficient ($\sigma_{abs}$) at seven different wavelengths (370, 470, 520, 590, 660, 880 and 950 nm) with a flow rate of 4 Lmin$^{-1}$ and a time resolution of 1 minute. Equivalent black carbon (eBC) concentration is inferred by measuring the absorption coefficient at 880 nm using a mass absorption cross section of 7.77 m$^2$g$^{-1}$ (Titos et al., 2017). Measurements of the absorption coefficient by the aethalometer are corrected with the one measured by the Multi Angle Absorption Photometer (MAAP,

model 5012, Thermo Fisher) at 637 nm. More details of the instruments are in Drinovec et al., (2015) and Petzold and Schönlinner, (2004), respectively.

The scattering and absorption coefficients measured by the integrating nephelometer and the aethalometer, respectively, have been used to calculate the scattering Ångström Exponent (SAE) and the absorption Ångström Exponent (AAE):

$$SAE_{\lambda_1-\lambda_2} = -\frac{\ln\left(\frac{\sigma_{sca}(\lambda_1)}{\sigma_{sca}(\lambda_2)}\right)}{\ln\left(\frac{\lambda_1}{\lambda_2}\right)},$$  (5)

$$AAE_{\lambda_1-\lambda_2} = -\frac{\ln\left(\frac{\sigma_{abs}(\lambda_1)}{\sigma_{abs}(\lambda_2)}\right)}{\ln\left(\frac{\lambda_1}{\lambda_2}\right)}.$$  (6)

The wavelengths used in this work to calculate both SAE and AAE have been 405-660 nm and 450-700 nm, respectively. Moreover, measurements of $\sigma_{abs}$ combined with those of $\sigma_{sca}$ permit the computation of the extinction coefficient $\sigma_{ext} = \sigma_{abs}+\sigma_{sca}$ and thus obtaining aerosol single scattering albedo (SSA) as:

$$SSA = \frac{\sigma_{sca}}{\sigma_{ext}},$$  (7)

where the Ångström law is used to calculate $\sigma_{abs}$ for the PI-Neph wavelengths. On the other hand, if $F_{11}(180°)$ is computed the extinction-to-backscattering ratio, widely known as lidar ratio (LRs) in the lidar community, can be computed as:

$$LR = \frac{\sigma_{ext}}{F_{11}(180°)},$$  (8)

The computation of $F_{11}(180°)$ has been made using the interpolation method used for completing the entire angular range in $\sigma_{sca}$. Other more robust methods can be used (i.e. Gomez-Martin et al., 2021), that can imply differences in $F_{11}(180°)$ of up to 20-30%. Therefore, LRs estimations will serve as an illustration of how this parameter varies under different conditions. We highlight that PI-Neph is not designed to accurately measure $F_{11}(180°)$ and there are other specific instruments that serve for that purpose (Järvinen et al., 2016; Miffre et al., 2023; Sakai et al., 2010).

## 3    Overview of extreme dust events during March 2022

During March 2022 the Iberian Peninsula, and particularly its southeast region, was affected by two intense Saharan dust outbreaks, especially during the 15th - 16th and the 24th - 25th of March. Figure 1 shows the geopotential height maps at 850 mb for 15th March 18:00 UTC and 25th March 12:00 UTC, which are close to the peaks of the event in each case. Data shown are from the NCEP/NCAR model (Kalnay et al., 1996; Kanamitsu et al., 2002) – https://tropic.ssec.wisc.edu/archive). On 15th March, Fig. 1a indicates the low-pressure system centered in the southwest of the Iberian Peninsula and northern Morocco associated with low values of geopotential heights. A high-pressure system is present in the central Mediterranean associated with high values of geopotential height. This high-pressure system covers wide regions from central Europe to the Sahara Desert in Libya and Tunisia. The interaction between these low-pressure and high-pressure systems favors strong southeastern winds to the south of the Iberian Peninsula. On 25th March, Fig. 1 reveals very similar patterns, although the low-pressure system is less intense (more sparse lines) and is displaced a little bit towards the east, centered over northern Morocco.

The synoptic situations on the 15th and the 25th of March described in Fig. 1 implied the advection of hot and dry air from the Sahara Desert region in Argelia and southern Morocco. The low-pressure system favored wind gusts and thus the injection of dust particles in the atmosphere, which are later transported long distances by the low-pressure system. This is one of the classical transport patterns of Saharan dust to the Iberian Peninsula (Escudero et al., 2005; Rodríguez et al., 2001; Salvador et al., 2014).

Figures 2 and 3 show total aerosol optical depth (AOD) at 550 nm generated by the CAMS model (Benedetti et al., 2009; Morcrette et al., 2009) over Europe. The wind field at the surface is also represented

in these Figures, and different times selected serve to understand how the dust was transported and affected different regions. Figure 2 clearly shows the counterclockwise winds associated with the low-pressure system and how this hot and dry air enters through the southeast reaching the north of the Iberian Peninsula and southern France. That airmass also transports large amounts of mineral dust particles as can be observed by the high values of simulated AODs. It is observed that the largest intensity of dust particles in the southeast of the Iberian Peninsula happened on the evening of 15ᵗʰ March. The wind pattern reveals how dust enters through the southeast of the Iberian Peninsula reaching later the northwest latitudes. The weak wind pattern in the North of the Iberian Peninsula favors the transport to southern France. For the 25ᵗʰ of March (Figure 3), the wind pattern is very similar to that on 15ᵗʰ March (Figure 2), although it does not reach the same northern locations. Indeed, the low-level system configuration seems to facilitate the transport of this hot and dry air to western locations in Portugal and the Atlantic Ocean. The CAMs model also predicts a larger amount of dust particles as indicated by the high AODs.

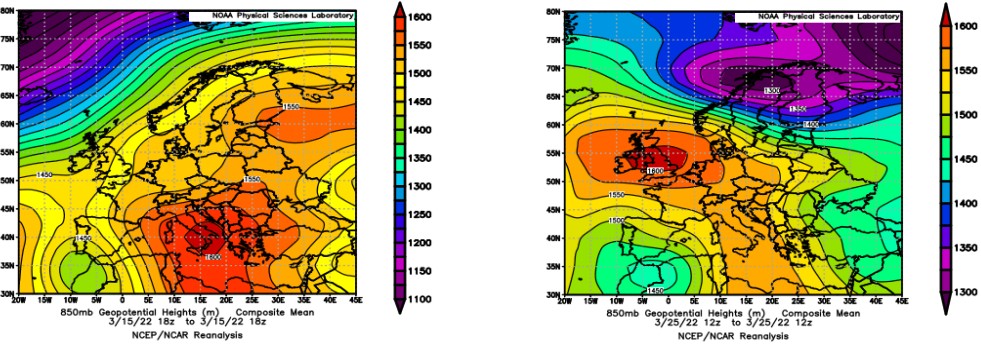

**Figure 1: Geopotential height at 850 mb for a) 15ᵗʰ March 2022 at 18:00 UTC and (b) 25ᵗʰ March 2022 at 12:00 UTC. Data are from NCEP/NCAR - https://tropic.ssec.wisc.edu/archive.**

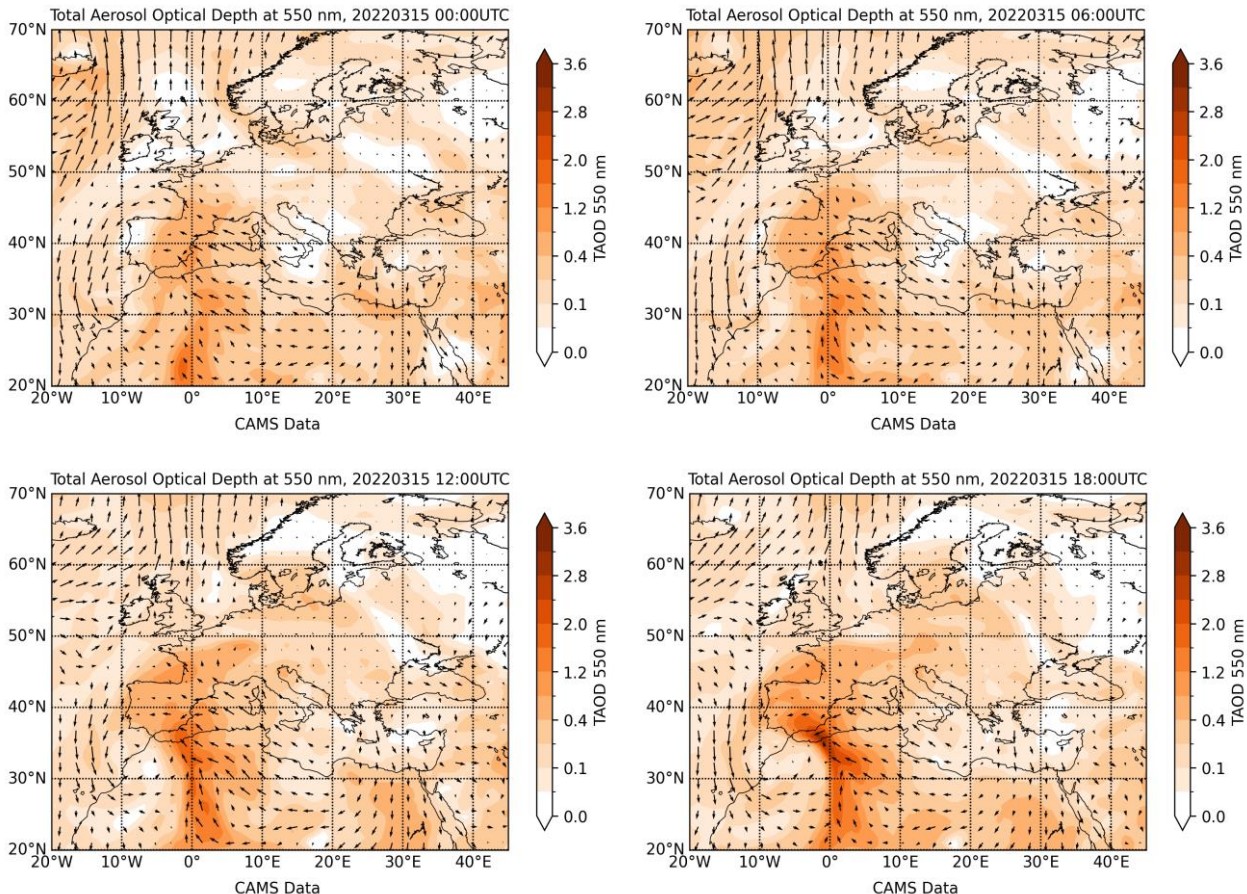

**Figure 2: CAMS model simulations of total aerosol optical depth (TAOD) and wind field for different times on 15th March 2022.**

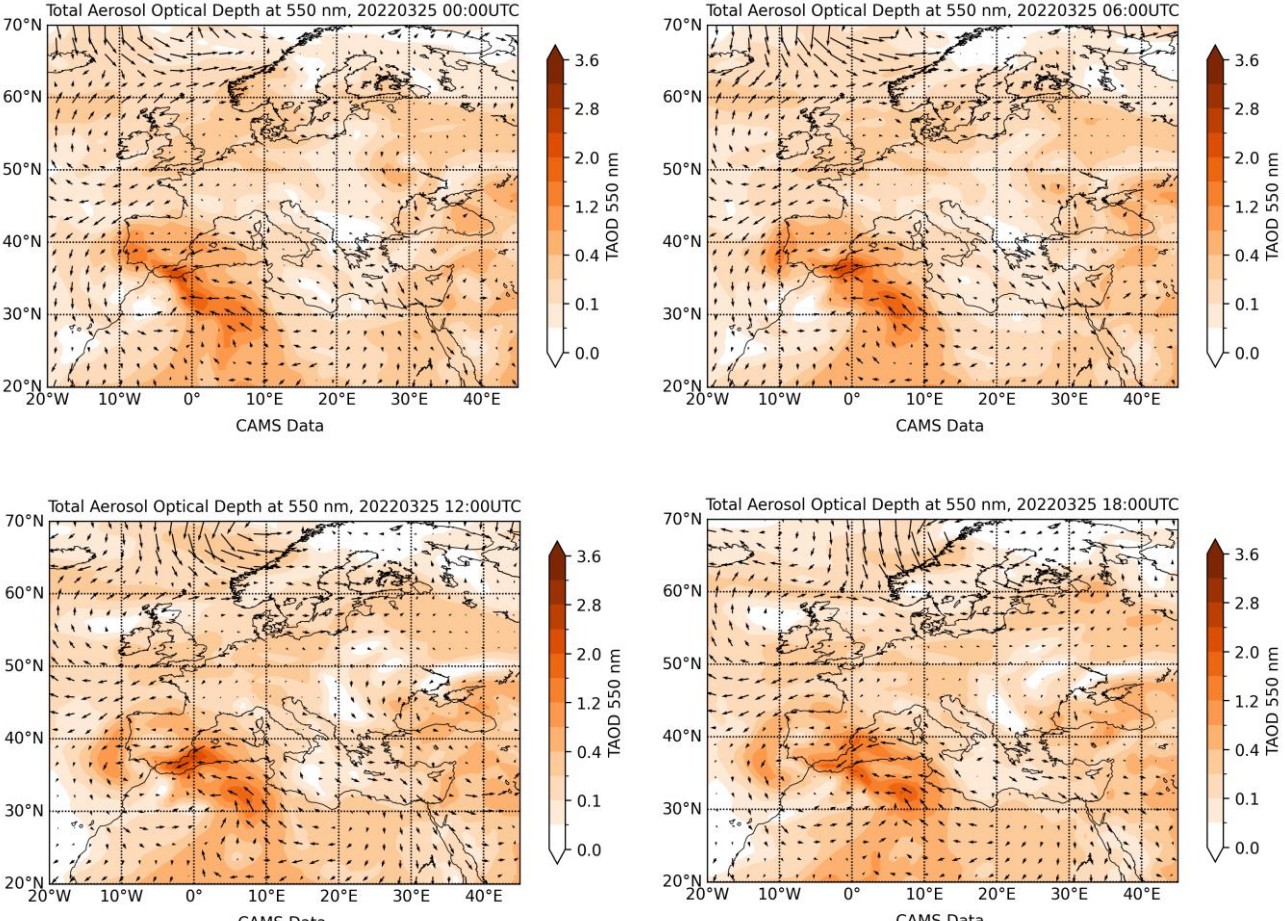

**Figure 3: CAMS model simulations of total aerosol optical depth (TAOD) and wind field for different times on 25th March 2022.**

Figure 4 shows satellite images provided by NASA worldview (https://wvs.earthdata.nasa.gov) that allow to have a visualization of the intensity of the dust outbreaks over the Iberian Peninsula. On both days, there were presence of clouds in the Iberian Peninsula because of the advection of humidity from the Atlantic by the low-pressure system (more intense on 25th which explains that almost all the Iberian Peninsula was covered by clouds). The image for 15th March clearly shows high presence of dust in the north of the Iberian Peninsula, the Cantabrian Sea and southern France. For the 25th of March the cloud cover hinders dust visualization, but it is observed in the Atlantic in the region between the Canary and Madeira islands. We highlight that such extreme events are not typical in winter season in the Iberian Peninsula, although it is not the first time that similar dust events have been registered in this season (Cazorla et al., 2017; Fernandez et al., 2019; Titos et al., 2017).

The events on 15th and 25th March 2022 can be considered as extreme Saharan dust outbreaks because of the large area covered, but especially because of the large amount of mineral dust particles transported. A more in-depth analysis of satellite images revealed that for the cloud-free pixel registered AODs were between 1.9 - 2.5 on 15th March and between 0.3 – 1.5 on 25th March, which are very high for these locations. The extremely high AODs values on 15th March associated with dust particles were confirmed by AERONET observations with AODs above 1.0 for the stations in northern Spain (A Coruña and Palencia) and southern France (Aubiere, Agen, Archaon and Momuy) where data of Level 2.0 Version 3 are available – graphs not shown for clarity but can be visualized in AERONET webpage (https://aeronet.gsfc.nasa.gov/). These high value of AODs agrees with those reported by CAMS model in Figure 2 and 3. Unfortunately, there is no AERONET data Level 2.0 Version 3.0 available on 25th March due to cloud-coverage in the AERONET stations. For both events, the urban background air-quality surface station in Granada (Palacio de Congresos (PAL) http://juntadeandalucia.es/medioambiente) registered 10-minute $PM_{10}$ concentrations, being up to 2500 $\mu gm^{-3}$ on the 15th of March and up to 800 $\mu gm^{-3}$ on the 25th of March, values that are way above the usual $PM_{10}$ values (~100 $\mu gm^{-3}$) registered at Granada during usual

dust outbreaks (Párraga et al., 2021). In this day, lidar measurements in AGORA in the framework of the European Aerosol Research Lidar Network (EARLINET; https://www.earlinet.org/) were saturated in the first 1-2 km and avoided any kind of retrieval of aerosol optical properties. Nevertheless, these measurements served to illustrate that most of the transport occurred in the first two kilometers above the ground.

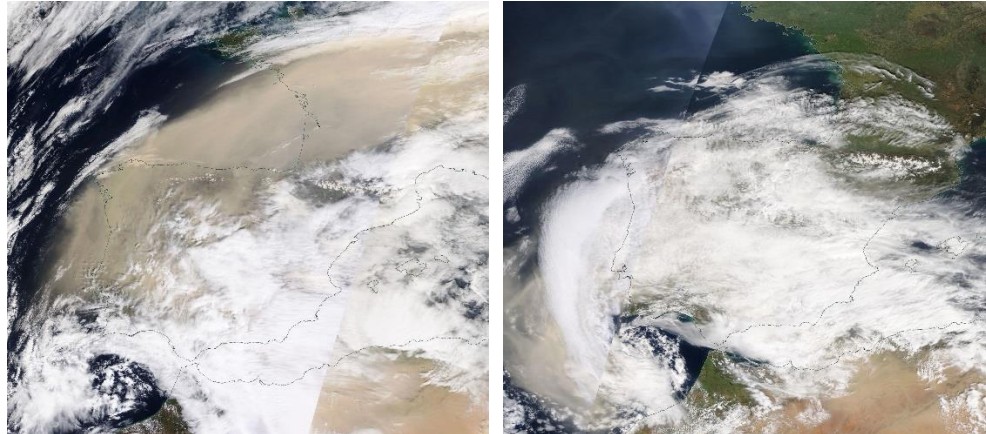

**Figure 4. (a) Satellite image from 15th March 2022. (b) Satellite image from 25th March 2022. Images from https://wvs.earthdata.nasa.gov, obtained with MODIS (Moderate Resolution Imaging Spectroradiometer).**

## 4     Results of aerosol phase matrix from different dust scenarios

### 4.1 Extreme events

For the two extreme events registered on 15th and 25th March 2022, Fig. 5 shows hourly averages of different aerosol properties obtained by the in-situ instruments at UGR station. In particular, we show eBC concentrations measured by the AE33, $\sigma_{sca}(\lambda)$ measured by the PI-Neph, and SAE, AAE (both calculated between 405 and 660 nm), $g(\lambda)$, $Bs(\lambda)$, $SSA(\lambda)$ and $LR(\lambda)$ derived from measurements of both instruments. $PM_{10}$ concentrations are also shown, which were obtained from the PAL air-quality station ($\sim$ 600 m distance from UGR station). Panel (a) in Fig. 5 shows the results for the event on 15th - 16th March while Panel (b) are for the event on 24th - 25th March. Note that Panel (a) in Figure 5 does not cover the beginning of the Saharan dust event due to lack of data related to supersaturation of the PI-Neph's measurements.

For the event on 15th - 16th March, Fig. 5a.1 reveals extremely high $PM_{10}$ with an average of 794 $\mu gm^{-3}$ being over the regulatory daily limit value of 50 $\mu gm^{-3}$ established by the Ambient Air Quality Directive (2008/50/CE European Directive). Maximum values of hourly $PM_{10}$ concentrations are registered at around 16 UTC, with values up to 1800 $\mu gm^{-3}$ approximately. The scattering coefficient time series shows the same behavior as the $PM_{10}$, with maximum values of $\sigma_{sca}$ at the same time as the peak of the $PM_{10}$ concentration. The mean value ($\pm std$) of SAE is $0.01 \pm 0.15$, which increases to 0.20 when $PM_{10}$ concentrations reach their maximum. Such values of SAE suggest clear predominance of large particles. For the g parameter (Fig. 5a.4) approximately constant values are observed with mean values of $0.672 \pm 0.010$, $0.701 \pm 0.006$ and $0.729 \pm 0.007$ for 405, 515 and 660 nm, respectively, which are typical values for transported dust particles (Horvath et al., 2018). Also, Horvath et al. (2018) found an averaged Bs of 0.094 at 532 nm for Saharan dust, which agrees with the results shown in Fig. 5a.5 for the 515 nm wavelength ($0.098 \pm 0.003$). The low values of the Bs observed at the three wavelengths are associated with small values of $F_{11}$ at the backward scattering angles, which is common for non-spherical particles (Horvath et al., 2018). The lack of aethalometer data did not allow eBC, AAE, SSA and LR analyses for this day.

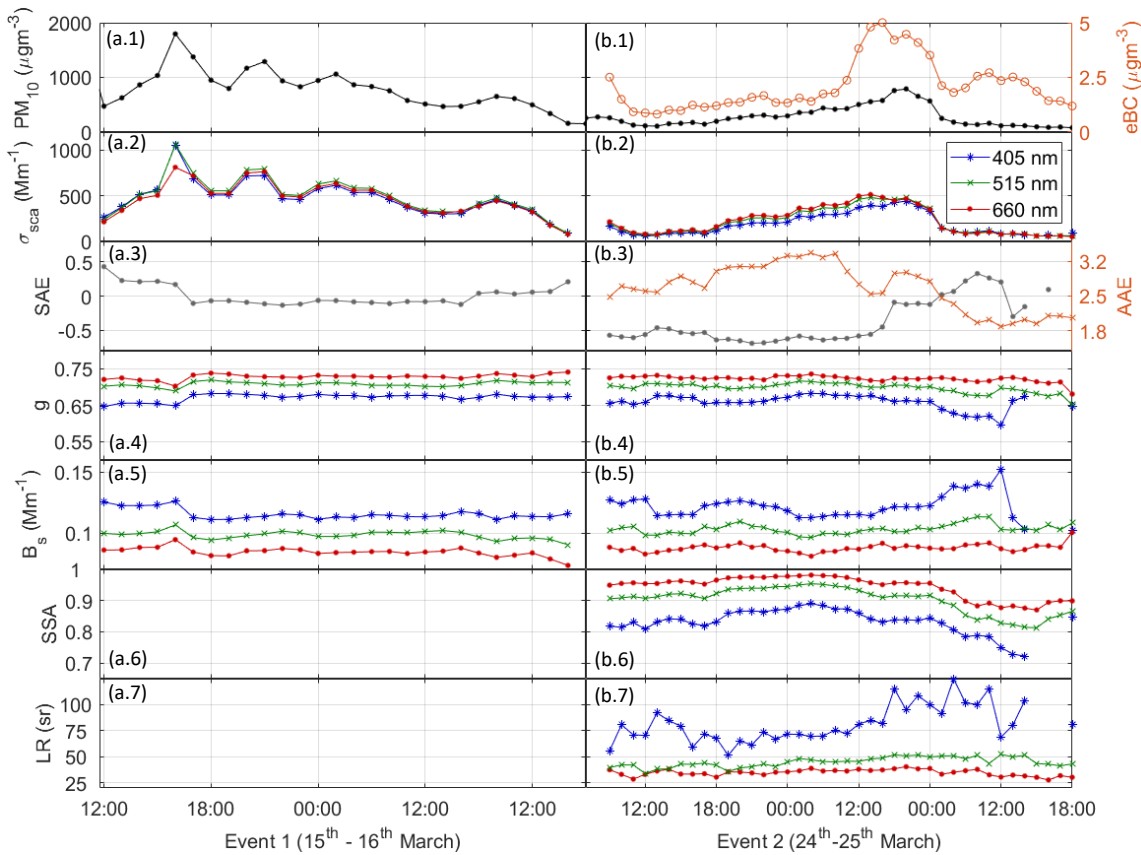

**Figure 5. Time series of the PM$_{10}$ and eBC concentrations (a.1, b.1), σ$_{sca}$ – scattering coefficient (a.2, b.2), SAE – scattering Angström exponent and AAE – absorption Angström exponent (a.3, b.3), g – asymmetry parameter (a.4, b.4), Bs – fraction of backscattered light (a.5, b.5), SSA – single scattering albedo (a.6, b.6) and LR – lidar ratio (a.7, b.7) for the extreme dust events of 15th March (a) and 24th March 2022 (b).**

For the event on 24th - 25th March, the average PM$_{10}$ concentration is 283 µgm$^{-3}$. These PM$_{10}$ concentrations (Fig. 5b.1) increase until 10 UTC, when reach the maximum hourly value of 790 µgm$^{-3}$. These values are lower than those registered on 15th March, but again they are very high and above the daily limit of the European Directive. Scattering coefficients at the time of PM$_{10}$ concentration peak are 468, 476 and 441 Mm$^{-1}$ for the 660, 515 and 405 nm wavelengths, respectively. After this peak, both PM$_{10}$ and σ$_{sca}$ decrease, suggesting the end of the extreme dust outbreak. The SAE main feature is its approximately constant value of –0.5 until early in the morning on 25th March when it slightly starts to increase. This pattern suggests clear predominance of large particles during the dust outbreak, while the increase in the morning can be associated with additional influence of small particles, likely originated from road traffic during morning traffic rush hours (Lyamani et al., 2010a) On the other hand, the AAE shows a different behavior, with high values at the beginning of the time series representative of mineral dust and a later decrease to values around 1.7 coincident with the decrease in PM$_{10}$. These values of AAE are in the range of those reported by Valenzuela et al., (2015) for mixture of dust and black carbon particles, and also agrees with the patterns of eBC and PM$_{10}$ that suggest a larger contribution of eBC to the total PM$_{10}$ concentration at the end of the dust event than at the peak. On the other hand, g mean values are 0.659 ± 0.020, 0.698 ± 0.013 and 0.723 ± 0.009 for 405, 515 and 660 nm, respectively, being very similar to those observed on the other extreme event on 15th – 16th March. For the Bs (Fig.5.5) again is observed a similar behavior to the previous event (Fig. 5a.5), with values around 0.1 for the 515 nm wavelength and below 0.15 in the three channels. The channel at 405 nm seems to be the most sensitive in g and Bs to changes in dust concentration since the time when a sharp change happens (25th March at 14:00 UTC) coincides with the hour when PM$_{10}$ decreases and eBC increases indicating more contribution of eBC to the total ensemble of particles.

Aethalometer measurements during 24th-25th March allowed the study of further aerosol optical properties: the SSA (Fig. 5b.6) shows the highest values for all three wavelengths during the dust outbreaks. However, there is a strong spectral dependency, since the SSA at 405 nm clearly shows lower values (~0.85) compared to the other wavelengths, that present SSA above 0.9. As the PM$_{10}$ concentration decreases (and eBC

concentration contribution to the mixture increases), the SSA shows smaller values with a stronger decrease in the 405 nm wavelength. Lastly, Fig. 5b.7 shows the time series of the LR during the second dust event, on 25th March. The LRs at 515 nm and 660 nm are rather constant, with mean values of 45 ± 5 sr and 35 ± 3 sr, respectively. However, the LR at 405 nm shows higher values and variability, with 81 ± 18 sr. LR at 515 nm is very similar to those measured by lidar systems at 532 nm for transported Saharan dust layers (Groß et al., 2013). Thus, the hypothesis that after 14:00 the presence of pollution particles becomes more relevant implies a decrease in SSA and illustrates variability in LRs, particularly in 405 nm.

Phase matrix elements were exhaustively monitored with the use of the PI-Neph during both extreme events. Given the high concentrations of large particles, the usual configuration of the measurements could lead to saturation of many angles in the forward scattering. Therefore, it was necessary to reduce the gain of the PI-Neph's camera, changing the dynamic range of the camera for obtaining non-saturated measurements at such high concentrations. But these changes were made to also guarantee enough signals in the backward region where the minimums are found. Nevertheless, sporadic pixels might present saturation/low SNR at some angles, but they were filtered out by the data quality criterion for the instrument (Section 2.2.1). For the specific phase function measurements where many data points are rejected, all angles are eliminated because phase matrix measurements were considered non-reliable. Moreover, $-F_{12}/F_{11}$ is computed by subtracting first parallel and perpendicular phase functions, that in many angles are very similar. Dividing this small number by $F_{11}$ can enhance the differences, being particularly critical in the angular regions where the minimum values of scattering are found (typically between 90-150º for dust particles). All these effects, although they are always present, imply larger random noise in the measurements. Actually, noisier patterns in measured $F_{11}$ and $-F_{12}/F_{11}$ affected by large particles when compared with measurements of anthropogenic origin have been already reported by the first versions of PI-Neph in the United States (Espinosa et al., 2018).

Figure 6 shows $F_{11}$ and $-F_{12}/F_{11}$ phase matrix elements for the event on 15th - 16th March 2022 at four different representative stages. Data are 60-minutes averages, where the standard deviations represent the variability of the different parcels of air sampled throughout the hour of measurements. These standard deviations are around 20% of the average values for $F_{11}$ and ranging between 0.1 and 0.2 for $-F_{12}/F_{11}$. Note that the large standard deviation can be explained by the specific issues for the measurements of dust particles commented above. Detailed hourly evolutions can be found in the Supplementary Material (Figs. S1-S2). Mean hourly averages of intensive and extensive aerosol parameters for the time periods shown in Fig. 6 are given in Table 1, particularly $PM_{10}$, $\sigma_{sca}$, SAE, AAE and g – note that SSA and LR were not available due to the lack of Aethalometer data for that day. Error bars in Table 1 are the standard deviations of the hourly mean values. Just before the impact of the extreme dust plume on 15th March 07:00 UTC (Figure 6a) values of $PM_{10}$ (~61 $\mu gm^{-3}$) and SAE (~1.65) can be considered as background in the station and represent a mixture between fine and coarse mode particles. Later, on 15th March 12:00 UTC (Figure 6b) the drastic increase in $PM_{10}$ (~473 $\mu gm^{-3}$) and decrease in SAE (~0.43) highlight a much larger contribution of coarse mode particles. Extreme values of $PM_{10}$ of ~1375 $\mu gm^{-3}$ on 15th March 17:00 UTC (Figure 6c) corresponds to the peak of the event. The lowest SAE (-0.11) was registered at that moment, and therefore a large contribution of coarse particles is expected. Finally, on 16th March 13:00 UTC (Figure 6d) the decrease in PM10 (~338 $\mu gm^{-3}$) plus the increase in SAE (~0.07) seems to indicate that the Saharan dust plume starts to withdraw.

Figure 6 shows a general pattern in $F_{11}$ characterized by strong predominance of forward scattering up to two orders of magnitude greater than backward scattering. However, there are significant changes in both magnitudes and spectral dependence over time, that is, with the intensity of the dust outbreak passage. At the beginning of the dust event (Fig. 6a), the values of $F_{11}$ in the forward scattering region are around 1000 $Mm^{-1}sr^{-1}$ for all three wavelengths, which is even one order of magnitude lower when compared with the cases at the other moments of the event (i.e. 50000 $Mm^{-1}sr^{-1}$ for the three channels during the peak). Also, at the beginning of the event (Fig. 6a) notable spectral separation in $F_{11}$ is observed while such spectral separation is negligible during the rest of the event when coarse mode particles largely predominate. All $F_{11}$ show the minimum in the region 120º-140º but the magnitude of that minimum varies between the different stages. Also, around that minimum is the region where some spectral difference is observed during the cases of strong predominance of coarse mode (Fig. 6c-d). A recovery from that minimum is also observed, being more pronounced in the cases close to the peak of the event.

Figure 6 shows that the differences in $-F_{12}/F_{11}$ patterns and wavelengths dependences with time are more remarkable than those observed for $F_{11}$. At the beginning $-F_{12}/F_{11}$ shows very different spectral patterns with remarkable spectral separation: for 515 and 660 nm, $-F_{12}/F_{11}$ follows bell-shaped patterns with values near to zero at the edges (0° and 180°) and a maximum around 90° of 0.4 and 0.5 for 515 and 660 nm, respectively. However, for 405 nm, the pattern is markedly different, with maximum values of 0.1 occurring at around 80°, followed by a sharp decrease, reaching negative values of -0.4 close to 150°. For the two following cases (Figs. 6b.2 and 6c.2), which are close to the peaks of maximum intensity in the Saharan dust outbreak, $-F_{12}/F_{11}$ shows a distinct pattern characterized by almost negligible differences with wavelength and a very small bell-shaped pattern with maxima around 0.1 at ~90°. Later, after the strong dust passage, Fig. 6d.2 shows for $-F_{12}/F_{11}$ a similar bell-shape pattern at 515 and 660 nm, but there is presence of some negative values at 405 nm.

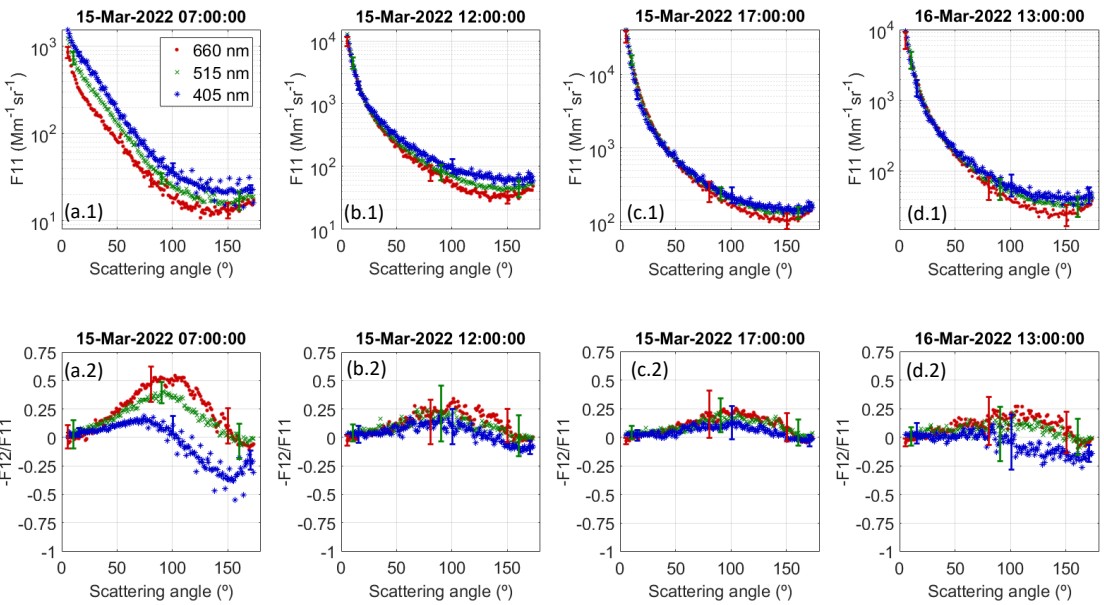

**Figure 6. Hourly averages of phase function ($F_{11}$) and polarized phase function ($-F_{12}/F_{11}$) on 15th - 16th March 2022 for four different stages of the evolution of the extreme Saharan dust outbreak: (a) 15th March 07:00 UTC before the Saharan dust outbreak reached the station, (b) 15th March 12:00 UTC when the Saharan dust begins to reach the station, (c) 15th March 17:00 UTC associated with the peak of the extreme Saharan dust intrusion, and (d) 16th March 13:00 UTC when Saharan dust starts to withdrawn.** *Error bars correspond to the standard deviation of the hourly averages.*

For the event on 24th - 25th March Fig. 7 shows $F_{11}$ and $-F_{12}/F_{11}$ phase matrix elements for four different representative instants during the event. The data correspond again to 60-minute averages, with detailed hourly evolutions shown in the Supplementary Material (Figs. S3-S4), where the standard deviations represent the variability of the samples. These standard deviations are around 20% of the average values for $F_{11}$ and around 0.2 for $-F_{12}/F_{11}$, showing larger variability during this dust event than in the previous one. Again, the specific issues related to the measurements of large particles can explain the deviations. Nevertheless, the larger deviations when compared with the previous events make us think that during this event there were more aerosol variability that is critical for the regions of the minimums in scattering. Mean hourly averages for these periods of intensive and extensive aerosol variables are again in Table 1, but now Aethalometer measurements permitted to add EBC, SSA and LR. Once again, this event exhibits lower values in particulate matter and appears to be less intense when compared to the event on 15th -16th March. However, it can still be considered an extreme event because the maximum $PM_{10}$ concentrations (> 700 $\mu gm^{-3}$) registered are above the typical values (~100 $\mu gm^{-3}$) of Saharan dust transport to the UGR station (Parraga et al., 2021). Table 1 results serve to understand the temporal evolution of the extreme Saharan dust outbreak. On 24th March 13:00 UTC the lowest $PM_{10}$ (~108 $\mu gm^{-3}$) are registered and can be associated with the background conditions before the intense outbreak. On 24th March 21:00 UTC the $PM_{10}$ values (~297 $\mu gm^{-3}$) are almost three times those registered at noon and is associated with the entrance of the extreme event, while at on 25th March 09:00 UTC the largest $PM_{10}$ values (~796 $\mu gm^{-3}$) are registered and that moment can be associated with the peak of the event. Finally, on 25th March 20:00 UTC the lowest

values of $PM_{10}$ (~116 µgm⁻³) and associate that moment with the withdrawal of the extreme event. For the cases of very high $PM_{10}$ (> 300 µgm⁻³), although they present the largest eBC, the values of SAE and AAE suggest large predominance of coarse mode particles. But for the rest of the cases the mixture seems more complicated, and no conclusive claim can be made initially about the predominance of any kind of particles.

Figure 7 shows that $F_{11}$ patterns are very similar to the previous extreme event on 15th - 16th March, with strong predominance of forward scattering (~25000 Mm⁻¹sr⁻¹), being two orders of magnitude above the backscattering (~100 Mm⁻¹sr⁻¹) at the peak of the event on 25th March 9:00 UTC. There are no significant spectral differences, as also happened for the other extreme event on 15th-16th March. These patterns in $F_{11}$ agree with laboratory measurements of dust samples (i.e. Muñoz et al., 2007; Renard et al., 2014; Volten et al., 2001). Nevertheless, there are some features in $F_{11}$ with the different situations: the slope in $F_{11}$ in the forward scattering region becomes sharper when the $PM_{10}$ concentrations are higher (Figs. 7b.1 and 7c.1). For the backward region $F_{11}$ shows a flatter behavior for high $PM_{10}$ concentrations (Figs. 7b.1 and 7.c.1), while for the cases with lower $PM_{10}$ concentrations there is a sharp increase in scattering from 150° to 180°. During the previous extreme dust outbreak, we observed flat patterns for the backward scattering region during the peaks of the dust intrusions.

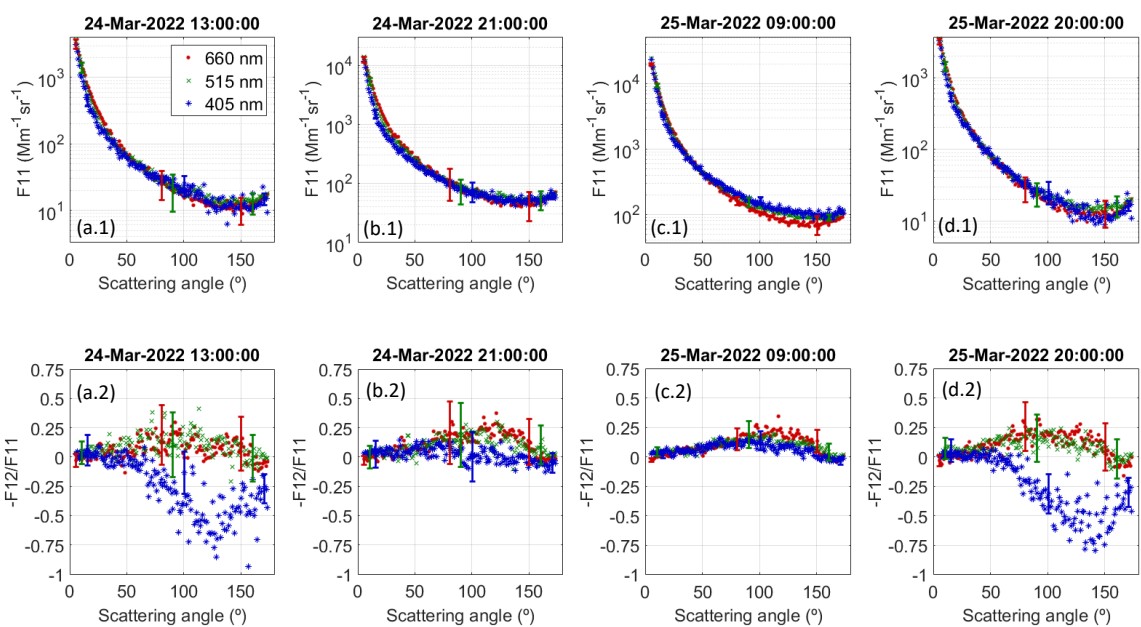

**Figure 7. Hourly averages of phase function ($F_{11}$) and polarized phase function ($-F_{12}/F_{11}$) on 24th - 25th March 2022 for four different stages of the evolution of the extreme Saharan dust outbreak: (a) 24th March 13:00 UTC before the Saharan dust outbreak reached the station, (b) 24th March 21:00 UTC when the Saharan dust starts to reach the station, (c) 25th March 09:00 UTC associated with the peak of the extreme Saharan dust outbreak, and 25th March 20:00 UTC when dust begins to withdrawn. Error bars correspond to the standard deviation of the hourly averages.**

Measurements of $-F_{12}/F_{11}$ in Figure 7 exhibit several behaviors throughout the event. During the peaks of the event on 24th-25th March (Fig. 7b-c), $-F_{12}/F_{11}$ patterns show minimal differences with wavelength and a very small bell-shaped pattern with maximums around 0.1 at 100°. However, there are differences when compared to the instants before the arrival of the dust outbreak (Fig. 7a) and when the dust is withdrawing (Fig. 7d). In these two cases $-F_{12}/F_{11}$ shows bell-shape patterns for 515 and 660 nm, with maximums of approximately 0.2 around 100° and values close to zero in the regions for scattering angles below 50° and above 150°. These patterns in $-F_{12}/F_{11}$ agree with the observed for the other instants of the event. However, the pattern for 405 nm is markedly different from the rest, and it is characterized by almost flat values close to zero of $-F_{12}/F_{11}$ in the range of approximately 0° - 50°, followed by a sharp decrease reaching negative values of -0.6 close to 130°. Then, there is a sharp increase in $-F_{12}/F_{11}$ reaching values close to zero at 180°. Therefore, $-F_{12}/F_{11}$ at 405 nm appears to be highly sensitive to the possible influence of other particles in the mixture. It is also noteworthy that the maximum in eBC coincides with the maximum of $PM_{10}$, but the contribution of eBC to the total aerosol burden is lower due to the high concentrations of dust. This can

 explain the dust-like pattern of $-F_{12}/F_{11}$ at the peak of the event and the general agreement with laboratory measurements of dust samples (Muñoz et al., 2007b; Volten et al., 2001b).

**Table 1. Hourly averaged properties of different stages of the extreme dust outbreaks in March 2022 reported in Figs. 6 and 7. The properties are reported at three wavelengths in the order of 660, 515 and 405 nm from top to bottom. Only the angular range of the PI-Neph (5°-175°) is used as the integration range of $\sigma_{sca}$. Error bars correspond to the standard deviations of the hourly means.**

| | $PM_{10}$ ($\mu gm^{-3}$) | eBC ($\mu gm^{-3}$) | $\sigma_{sca}$ ($Mm^{-1}$) | SAE | AAE | g | Bs ($Mm^{-1}$) | SSA | LR (sr) |
|---|---|---|---|---|---|---|---|---|---|
| 15th Mar 07:00 | 61 ± 5 | - | 55 ± 3 | 1.65 ± 0.13 | - | 0.573 ± 0.015 | 0.135 ± 0.013 | - | - |
| | | | 84 ± 4 | | | 0.604 ± 0.018 | 0.117 ± 0.011 | | |
| | | | 123 ± 7 | | | 0.616 ± 0.008 | 0.116 ± 0.004 | | |
| 15th Mar 12:00 | 473 ± 131 | - | 215 ± 36 | 0.43 ± 0.13 | - | 0.722 ± 0.001 | 0.086 ± 0.005 | - | - |
| | | | 241 ± 38 | | | 0.703 ± 0.001 | 0.100 ± 0.007 | | |
| | | | 265 ± 39 | | | 0.641 ± 0.001 | 0.126 ± 0.005 | | |
| 15th Mar 17:00 | 1376 ± 256 | - | 718 ± 128 | -0.11 ± 0.12 | - | 0.734 ± 0.008 | 0.084 ± 0.005 | - | - |
| | | | 746 ± 112 | | | 0.715 ± 0.009 | 0.096 ± 0.005 | | |
| | | | 682 ± 101 | | | 0.679 ± 0.009 | 0.113 ± 0.006 | | |
| 16th Mar 13:00 | 338 ± 48 | - | 178 ± 47 | 0.07 ± 0.13 | - | 0.739 ± 0.008 | 0.079 ± 0.005 | - | - |
| | | | 191 ± 52 | | | 0.713 ± 0.008 | 0.095 ± 0.005 | | |
| | | | 184 ± 50 | | | 0.673 ± 0.011 | 0.113 ± 0.006 | | |
| 24th Mar 13:00 | 108 ± 14 | 0.81 ± 0.10 | 76 ± 13 | -0.46 ± 0.22 | 2.58 ± 0.10 | 0.733 ± 0.024 | 0.084 ± 0.014 | 0.955 ± 0.004 | 36 ± 10 |
| | | | 70 ± 12 | | | 0.709 ± 0.031 | 0.098 ± 0.020 | 0.912 ± 0.006 | 38 ± 11 |
| | | | 65 ± 8 | | | 0.664 ± 0.035 | 0.117 ± 0.018 | 0.832 ± 0.006 | 92 ± 20 |
| 24th Mar 21:00 | 297 ± 21 | 1.59 ± 0.14 | 284 ± 27 | -0.69 ± 0.18 | 3.09 ± 0.03 | 0.725 ± 0.025 | 0.088 ± 0.016 | 0.975 ± 0.001 | 34 ± 4 |
| | | | 257 ± 25 | | | 0.701 ± 0.025 | 0.105 ± 0.017 | 0.940 ± 0.001 | 40 ± 6 |
| | | | 205 ± 16 | | | 0.659 ± 0.021 | 0.124 ± 0.015 | 0.867 ± 0.003 | 61 ± 16 |
| 25th Mar 09:00 | 760 ± 44 | 4.21 ± 0.21 | 466 ± 15 | -0.09 ± 0.06 | 2.96 ± 0.05 | 0.725 ± 0.007 | 0.087 ± 0.005 | 0.957 ± 0.002 | 39 ± 5 |
| | | | 461 ± 17 | | | 0.705 ± 0.009 | 0.101 ± 0.006 | 0.915 ± 0.003 | 52 ± 6 |
| | | | 427 ± 14 | | | 0.662 ± 0.007 | 0.122 ± 0.005 | 0.838 ± 0.005 | 115 ± 8 |
| 25th Mar 20:00 | 116 ± 6 | 2.3 ± 0.5 | 80 ± 7 | -0.16 ± 0.18 | 2.03 ± 0.08 | 0.722 ± 0.019 | 0.086 ± 0.011 | 0.876 ± 0.015 | 32 ± 7 |
| | | | 78 ± 6 | | | 0.689 ± 0.020 | 0.104 ± 0.014 | 0.815 ± 0.020 | 52 ± 18 |
| | | | 74 ± 3 | | | 0.667 ± 0.021 | 0.113 ± 0.011 | 0.722 ± 0.021 | 104 ± 10 |

**4.2 Moderate dust events during spring/summer 2022**

The PI-Neph also continuously operated from April to September 2022 and other events of Saharan dust transport were registered at the UGR station. However, these outbreaks did not exhibit such an extreme dust transport when compared with the events in March 2022. Actually, hourly averaged $PM_{10}$ levels were

540 below 130 μgm⁻³ and $\sigma_{sca}$ below 130 Mm⁻¹, which are typical values observed at the UGR station during Saharan dust outbreaks (Lyamani et al., 2010). For this entire period of measurements (Apr 14th to Sep 9th) Fig. 8 shows hourly averages of $PM_{10}$ and eBC concentrations, $\sigma_{sca}(\lambda)$, SAE, AAE (both calculated in the 405-660 nm range), g(λ), Bs(λ), SSA(λ) and LR(λ). For the identification of cases with influence of mineral dust particles, data shown in Fig. 8 are filtered out and correspond only to values of SAE< 1, which are

545 used as a proxy for the presence of dust particles in the atmosphere (i.e. Lyamani et al., 2010; Teri et al., 2024).

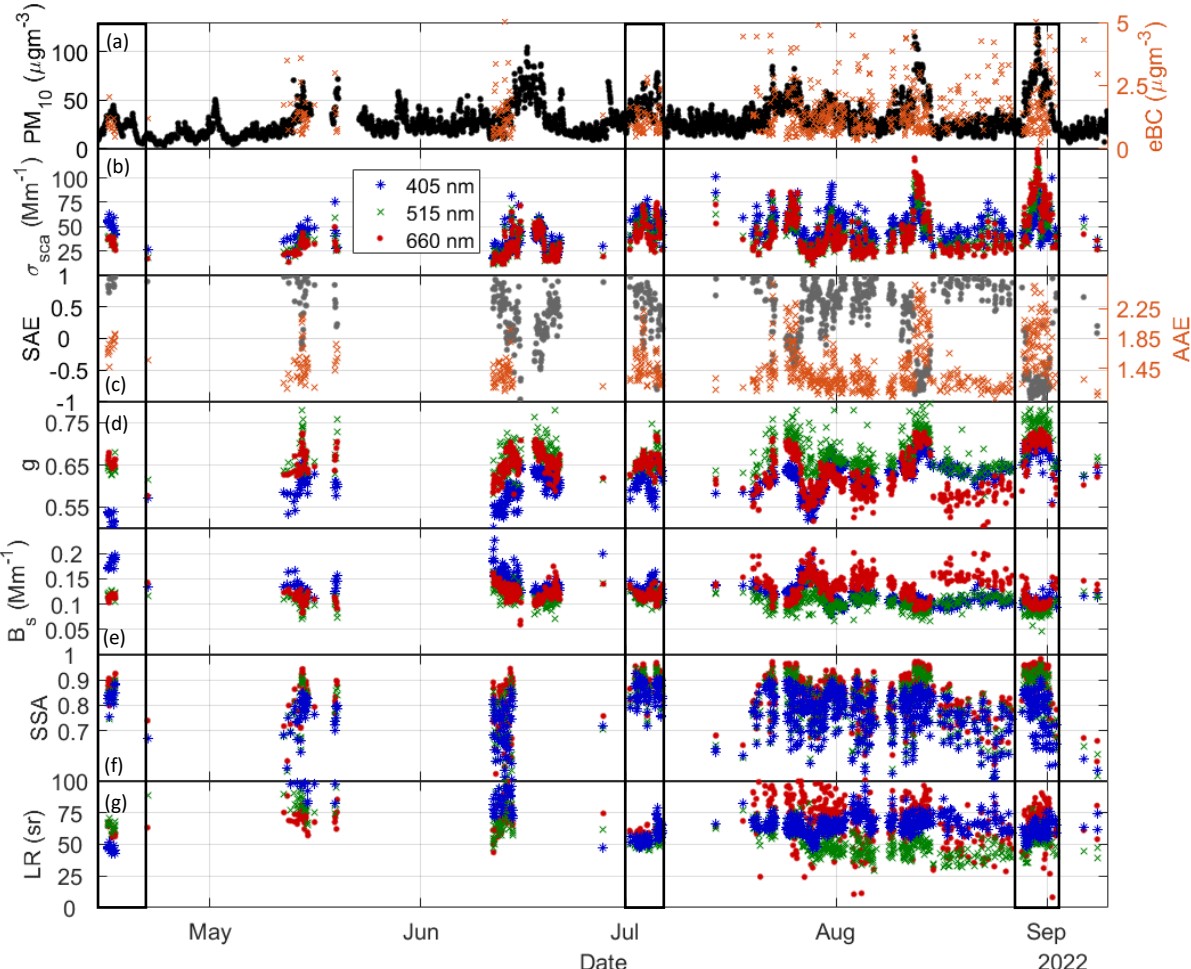

**Figure 8. Time series for the moderate dust events of 2022 of the $PM_{10}$ and EBC concentrations (μgm⁻³) (a), $\sigma_{sca}$ – scattering coefficient (Mm⁻¹) (b), SAE – scattering Angström exponent and AAE – absorption Angström**

**exponent (c), g – asymmetry parameter (d), Bs – fraction of backscattered light (Mm⁻¹) (e), SSA – single scattering albedo (f) and LR – lidar ratio (sr) (g). All optical properties are shown at 405, 515 and 660 nm. Black boxes represent three different events on 16th April, 5th July and 30th August 2022.**

Figure 8 reveals that the period with the least frequency of dust events was April - June, although the few cases detected show high values of $PM_{10}$ concentrations and the scattering coefficients with SAE values

closer to 1, suggesting a high degree of mixture between dust and pollution. Also, this period presents large spectral dependency in g and Bs, with g ranging between 0.5 and 0.7, and Bs ranging from 0.13-0.18, which are typical values more related to non-Saharan dust aerosols (Horvath et al., 2018). Moreover, SSA shows large variability ranging from 0.5 to around 0.9 for all three wavelengths, where the lower limit of the range suggests the presence of absorbing particles in the sample. This is also supported by the AAE with values

close to 1, typical of BC.

During the summer season there is more frequency of Saharan dust intrusions, which is typical in the UGR station (Perez-Ramirez et al., 2016). Generally, there is more variability in all parameters, suggesting more complex mixtures of dust with other particles: the g parameter, which is wavelength dependent, shows very similar values to those obtained during spring, but with important variability ranging between 0.55 and 0.75. SAE and AAE show values between 0 and 1 and between 1 and 2, respectively, which are the typical values when there is influence of dust particles in our station (Valenzuela et al., 2015). The outliers in the summer season with negative SAE close to –0.5 and AAE of up to 2.5 might be associated with cases that have more presence of dust. In summer season the largest values of AAE (close to 2) are observed when compared with the spring season, which can be interpreted as fewer black carbon particles. SSA shows very similar values to those obtained during the spring season.

The LR is a critical variable for backscattered lidar systems and is an intensive aerosol variable that strongly depends on $F_{11}(180°)$ and absorption (Pérez-Ramírez et al., 2019). Because of that, LRs can be very sensitive to the different mixtures of particles in the atmosphere (Burton et al., 2012, 2013; Müller et al., 2007). Results of Fig. 8 serve to illustrate LR variability for dusty conditions but with the influence of other types of particles. Generally, Fig. 8 shows values between 40 sr and 100 sr for the three wavelengths. The lower limits are closer to the values for large predominance of dust (i.e. Müller et al., 2007) while the upper values are typical values registered for predominance of smoke/anthropogenic particles (Alados-Arboledas et al., 2011b; Burton et al., 2012, 2013; Floutsi et al., 2023; Müller et al., 2007). Thus, results of Fig. 8 indicate the large sensitivity of LR to changes in the mixture of particles. A seasonal analysis indicates that in spring – although there is less data – LRs are above 75 sr with little spectral dependence, suggesting more influence of fine particles in the mixture, which are ultimately responsible for LR values. During the summer seasons the lower values around 40-50 sr are more frequent, suggesting more predominance of coarse particles in the mixture.

To gain better understanding of the evolution of the phase matrix elements in different cases of dust mixtures with anthropogenic particles, Fig. 9 shows the phase matrix elements for three different dust events represented by the black boxes in Fig. 8. Particularly, Fig. 9 shows hourly averages of $F_{11}$ and -$F_{12}/F_{11}$ representative of the peak in scattering during each event. Table 2 summarizes hourly mean values of $PM_{10}$ and spectral $\sigma_{sca}$, g, Bs, LR and SSA, plus mean values of SAE and AAE for these selected cases. The standard deviations are larger (30% in in $F_{11}$ and around 0.2 in -$F_{12}/F_{11}$) when compared with the extreme events, and despite the inherent issues in the measurement of phase matrix for large particles, it seems that the sample presents a more complex mixture with more variability during the 1-hour average.

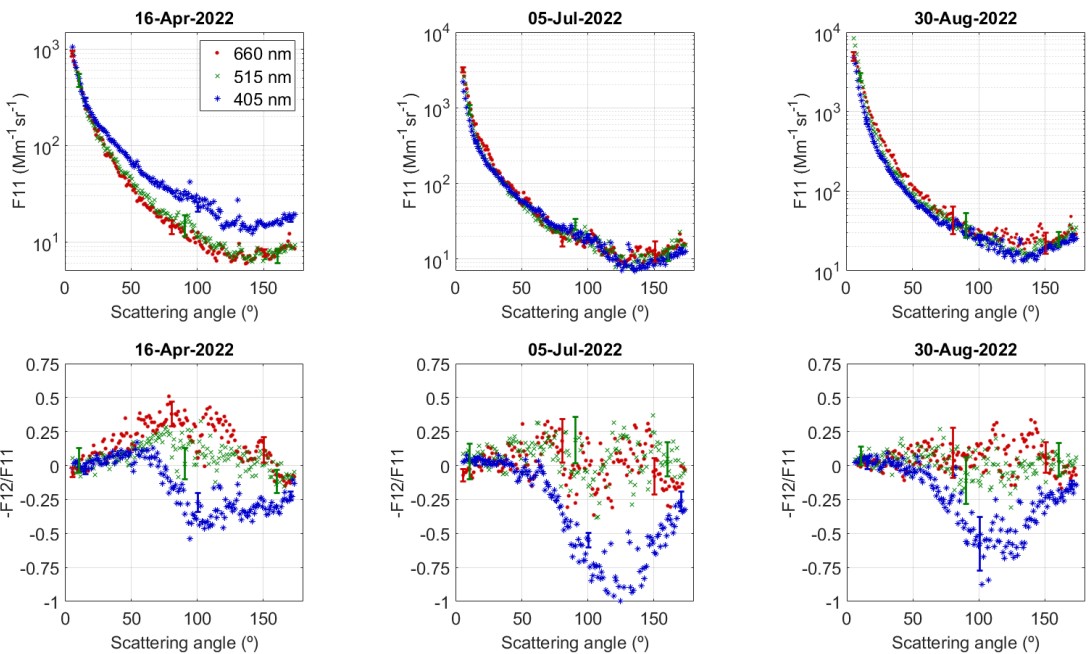

**Figure 9. Phase function ($F_{11}$) and polarized phase function (-$F_{12}/F_{11}$) (for different moderate dust events: 15th April 202, 25th July 2022 and 30th August 2022. Error bars correspond to the standard deviation of the hourly averages.**

The first event on 16[th] April is one of the most complex in terms of mixture of particles, with the lowest PM$_{10}$ but with the largest SAE and AAE. The values of g suggest the lower contribution of dust particles when compared to the other cases (Horvath et al., 2018). The flat spectral pattern in SSA is the typical observed when there is contribution of fine mode particles of anthropogenic origin during a Saharan dust outbreak (Valenzuela et al., 2014). Nevertheless, the high AAE is not typical of black carbon and thus remarks the complexity of the mixture of aerosol particles in that day. For the case on 5[th] July the presence of mineral dust particles seems more relevant. The more pronounced spectral SSA when compared to 16[th] April also relates to the influence of dust in absorption (Dubovik et al., 2002), although the event on 5[th] July shows a lower AAE than for the previous case. Finally, the case on 30[th] August is the one with the largest PM$_{10}$ concentrations but also with the largest eBC, which can make a very complex mixture. However, it should be noted that when affected by high concentrations of dust, the dust particles might interfere with the eBC measurements.

**Table 2. Hourly averaged properties of different dust events in 2022. For properties reported at three wavelengths, the order is 660, 515 and 405 nm (top to bottom). The integration range of σ$_{sca}$ is the angular range of the PI-Neph (5°-175°). Error bars are the standard deviations of the hourly means.**

| | PM$_{10}$ (μgm$^{-3}$) | eBC (μgm$^{-3}$) | σ$_{sca}$ (Mm$^{-1}$) | SAE | AAE | g | Bs (Mm$^{-1}$) | SSA | LR (sr) |
|---|---|---|---|---|---|---|---|---|---|
| 16[th] Apr | 36 ± 1 | 0.92 ± 0.14 | 31 ± 8 | 0.94 ± 0.88 | 1.71 ± 0.14 | 0.641 ± 0.024 | 0.119 ± 0.014 | 0.878 ± 0.017 | 59 ± 14 |
| | | | 31 ± 13 | | | 0.630 ± 0.036 | 0.126 ± 0.017 | 0.837 ± 0.028 | 61 ± 17 |
| | | | 48 ± 12 | | | 0.505 ± 0.005 | 0.193 ± 0.001 | 0.830 ± 0.008 | 43 ± 5 |
| 5[th] Jul | 76 ± 1 | 1.07 ± 0.11 | 61 ± 5 | -0.46 ± 0.07 | 1.37 ± 0.13 | 0.712 ± 0.035 | 0.097 ± 0.024 | 0.930 ± 0.004 | 61 ± 21 |
| | | | 53 ± 5 | | | 0.694 ± 0.060 | 0.109 ± 0.019 | 0.892 ± 0.016 | 58 ± 22 |
| | | | 49 ± 2 | | | 0.656 ± 0.016 | 0.114 ± 0.011 | 0.843 ± 0.007 | 66 ± 20 |
| 30[th] Aug | 124 ± 2 | 2.42 ± 0.71 | 129 ± 5 | -0.86 ± 0.11 | 1.73 ± 0.15 | 0.711 ± 0.019 | 0.096 ± 0.008 | 0.920 ± 0.021 | 67 ± 14 |
| | | | 115 ± 9 | | | 0.761 ± 0.021 | 0.080 ± 0.010 | 0.882 ± 0.028 | 58 ± 7 |
| | | | 85 ± 4 | | | 0.701 ± 0.009 | 0.104 ± 0.007 | 0.789 ± 0.032 | 63 ± 18 |

Figure 9 shows that all F$_{11}$ cases exhibit the typical pattern for large and non-spherical particles characterized by large predominance of scattering in the forward region, although there is no remarkable flat behavior of the curve in the backward scattering which is characteristic of this type of particles from laboratory measurements (i.e Muñoz et al., 2007). However, there are differences between the events shown. In the case of the first dust event on 16[th] April (Fig. 9a), a strong spectral dependency is observed between 405 nm and the other wavelengths from 30° on, which is not observed for the other events. The largest contribution of urban pollution to the mixture can be one of the reasons for these spectral variations on 16[th] April. Another possible reason could be the less influence of absorbing particles as this case also presents the lower eBC and largest SSA when compared with the other two cases. The cases with almost negligible spectral differences could be explained by a more predominance of large particles. Thus, the discussion of these three selected cases illustrates that even though dust predominant cases present a classical scattering pattern characterized by strong forward scattering, the spectral dependences and the shape of the forward scattering depend ultimately on the mixture of particles. Note that all F$_{11}$ are coherent with those obtained for the extreme dust outbreaks in March 2022, being the agreement more remarkable for the cases on 5[th] July and on 30[th] August.

Measurements of -F$_{12}$/F$_{11}$ in Fig. 9 present very similar patterns for the three cases. For 515 and 660 nm the -F$_{12}$/F$_{11}$ patterns are characterized by a bell-shape with large variability that might be associated with the complexity of the mixture of mineral dust and anthropogenic particles, as has been also observed for other dust particles measurements in the United States (Espinosa et al., 2018). However, the -F$_{12}$/F$_{11}$ pattern in

405 nm shows a very different behavior, having -$F_{12}/F_{11}$ positive values until ~ 70° and negative -$F_{12}/F_{11}$
values for the following angles. Minimum -$F_{12}/F_{11}$ values are in the region around 120°. Therefore, -$F_{12}/F_{11}$
measurements can be potentially used for investigating the mixture of particles in the sample.  That pattern
with negative values has been observed in the UGR station for cases with no influence of Saharan dust
particles (Bazo et al., 2024), and also for biomass-burning at 473 nm (Espinosa et al., 2017). Nevertheless,
there are differences between the three different cases that might be associated with the differences in the
mixtures of aerosol particles.

## 5    Discussion
### 5.1 Comprehensive assessment of the different dust events

To gain more insight about aerosol mixtures during the extreme dust event on 24[th] – 25[th] March 2022 and
the rest of dust cases registered in the period April – September 2022 we use the typing methodology
defined in (Cazorla et al., 2013) and modified in (Schmeisser et al., 2017), based on optical properties. To
that end, Fig. 10 shows SAE versus AAE, being both parameters computed in the range 450 -700 nm. For
the SAE, we have used the $\sigma_{sca}$ measured with the TSI integrating nephelometer since it directly provides
measurements at the same wavelengths than those required in Schmeisser et al., (2017). The different types
of aerosols are also illustrated in the Figure, where BC refers to black carbon and BrC to brown carbon in
the definitions given by Schmeisser et al., (2017). Different colors are used to identify different stages in
the temporal evolution for the extreme dust event on 24[th] – 25[th] March (Note that for the event on 15[th] – 16[th]
March there were no aethalometer data and thus no measurements of AAE).

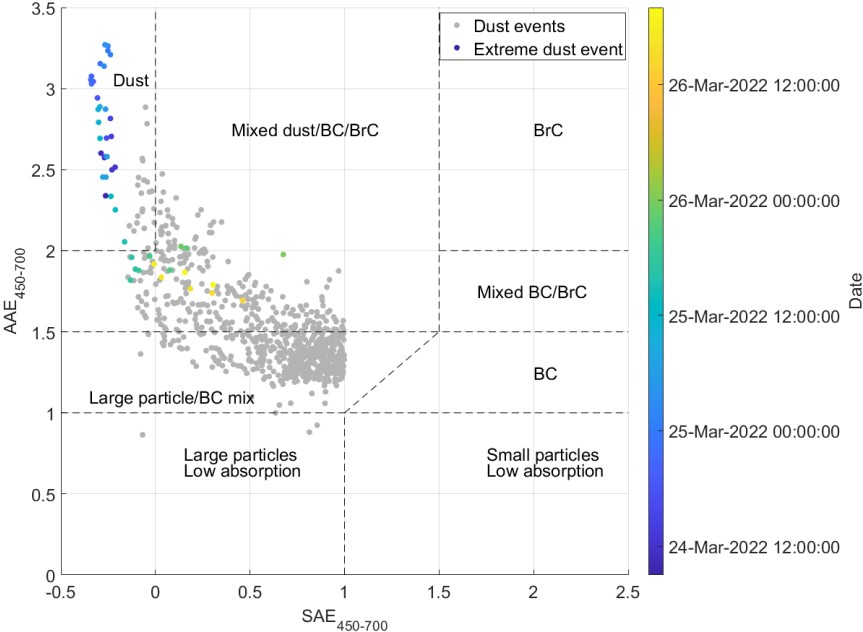

**Figure 10. Absorption Angström exponent (AAE) versus scattering Angström exponent (SAE) for the extreme
dust event on 24[th] - 25[th] March 2022 (colored markers) and for the moderate dust events registered in the UGR
station during the period April-September 2022 (gray markers). Both intensive properties have been calculated
in the range 450-700 nm. Colorbar indicates the temporal evolution of the extreme dust event on 24[th] - 25[th]
March. Classification of different aerosol types following the method proposed by Schmeisser et al., (2017) is
also shown.**

Figure 10 shows that for the extreme dust outbreak on 24[th]- 25[th] March 2022 most of the data fall into the
region of pure dust type, particularly those registered at beginning of the dust event when the PM$_{10}$
concentrations were extremely high implying large predominance of coarse mineral dust particles. As the
dust event evolves, particularly from 26[th] March 2022, the data points start to fall in the region mixed
dust/BC/BrC. This coincides with the drop of PM$_{10}$ concentration and the increase of eBC observed in Fig.
5. Therefore, there could be a more balanced contribution of both urban background pollution and mineral
dust. If we compare these typing features with the scattering matrix elements on 24-25[th] March 2022 (Fig.
7), we observe that for the cases typed as pure dust the -$F_{12}/F_{11}$ follows a bell-shape pattern for all
wavelengths. However, for the cases classified as mixtures, there was a different pattern -$F_{12}/F_{11}$ for the

665 405 nm channel. Thus, the typing classification explains the differences in the phase matrix for the temporal evolution of the extreme dust events during 15[th]-16[th] and 24[th]-25[th] March 2022 in the UGR station and show the potential of ground-based phase matrix measurements to distinguish between different types of aerosol mixtures.

For the rest of the analyzed period from April to September 2022 (classified as moderate events most in
Fig. 8), most of the data falls into the region of large particle/BC mix and mixed dust/BC/BrC in Fig. 10. Since the UGR station is affected by local pollution (mainly road traffic) it is expected that the dust transported from the Saharan Desert gets mixed with the urban background pollution that is already suspended in the atmosphere. Another possibility is that dust already presents anthropogenic particles injected in the ensemble of particles during the transport process (Querol et al., 2019; Valenzuela et al.,
2015). However, going further into the origin of these anthropogenic particles in the mixture is not possible with current data. In this study, the most important feature is that these possible mixtures can explain the differences in $F_{11}$ and $-F_{12}/F_{11}$ between the extreme and moderate dust events, and also the different situations of dust mixtures in the station.

To further understand the behavior of the optical properties during different dust events, we have performed
an average of all moderate dust events of the period April-September 2022 (filtered by SAE < 0.5 to guarantee more dust predominance) and compared it with the peaks of the extreme dust events, i.e. 15[th] March 2022 at 17:00 UTC and 25[th] March 2022 at 09:00 UTC. To this purpose, Fig. 11 shows $F_{11}$ and $-F_{12}/F_{11}$ phase matrix elements for the different situations. For comparisons, we also include laboratory measurements of Saharan dust samples with the polar nephelometer in the Andalusian Institute of
Astrophysics (Muñoz et al., 2010b), which are available in the Granada – Amsterdam Light Scattering Database (Muñoz et al., 2012). This database provides measurements of $F_{11}$ and $-F_{12}/F_{11}$ at 488 and 632 nm - see Gómez Martín et al., (2021) for details. Results of $F_{11}$ have been normalized with respect to $F_{11}(30°)$ to have the same scale for comparison.

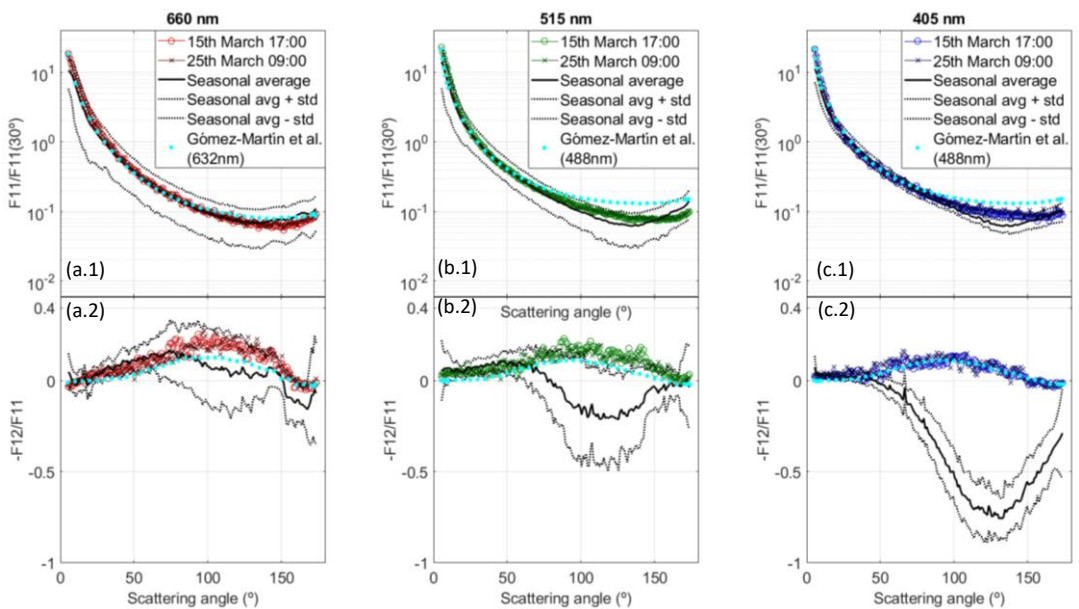

**Figure 11. Phase function ($F_{11}$) (top) and polarized phase function ($-F_{12}/F_{11}$) (bottom) for different situations: Mean values for the cases obtained during moderate dust events (black lines), cases for extreme events on 15[th] March 2022 (red open circles) and on 25[th] March 2022 (red stars), and laboratory measurements at 632 nm and 488 nm (light blue dots) with samples collected in the Sahara and available in the Granada-Amsterdam Light Scattering database (Muñoz et al., 2012).**

Figure 11 reveals that the $F_{11}$ matrix element presents very similar features for the three wavelengths between mean averages for the period April-September 2022 and the extreme dust events, being the difference within the standard deviations. There are only slight differences in the forward region above 160° scattering angles, particularly for 515 nm channel, that might be associated by the complexity of the scattering at these angles for large and non-spherical particles ((Mischenko et al., 2002; Muñoz et al., 2007).
Also, the small standard deviations of $F_{11}$ are remarkable for the mean seasonal values, which suggest that

$F_{11}$ follows very similar patterns when there is predominance of large particles, independently of the mixture with other anthropogenic particles. Moreover, these patterns of $F_{11}$ obtained from ambient aerosol basically coincide with those provided by the Light Scattering Database.

Figure 11 shows that seasonal values of $-F_{12}/F_{11}$ present in all cases larger standard deviations when compared to $F_{11}$. Particularly, for 660 and 515 nm large standard deviations are found in the region between 50º-150º while for 405 nm the standard deviations are considerably lower. Apart of the inherent limitations in the measurements in this scattering range, the results suggests that these $-F_{12}/F_{11}$ values at 660 and 515 nm are very sensitive to changing conditions in the aerosol that is sampled. Moreover, the other region that presents remarkable standard deviations for all wavelengths is the region of scattering angles above 170º. Those regions with large standard deviations are very sensitive to any change in particle type and size, which was demonstrated both from theoretical computations (Mischenko et al., 2002) and in laboratory measurements (Gomez-Martin et al., 2021). However, the lower standard deviations observed for the 405 nm wavelength indicate homogeneity in the response to polarization, even in the presence of other anthropogenic particles in the sample.

The comparison of means $-F_{12}/F_{11}$ between the different situations reveals very important features. Specifically, for 660 nm $-F_{12}/F_{11}$ follows a very similar pattern between the extreme events and the Light Scattering database. The mean seasonal average also follows the same pattern, but with a lower maximum and displaced to lower scattering regions. However, the 405 nm channel presents the most different behavior between the seasonal averages and the phase matrix elements for the dust events with two different patterns: For the extreme dust events and the Light Scattering Database, there is a bell shape pattern with values close to zero and a maximum of 0.1 at 100°, being the differences between both sets of data negligible. A different pattern is observed for the seasonal average, characterized by values around zero up to 50°, decreasing later to a minimum of -0.7 at 120° and increasing again to values close to 0 in the backward scattering region. Finally, the 515 nm channel presents an intermedia situation, being the extreme dust cases and the Light Scattering Database at 488 nm patterns very similar, while for the average of usual dust cases the pattern is like the observed at 405 but with less pronounced negative values.

The overall analysis of $F_{11}$ and $-F_{12}/F_{11}$ phase matrix elements reveal that $F_{11}$ patterns and spectral dependence is strongly affected by the existence of large and non-spherical particles, being the existence of other anthropogenic particles in the mixture affecting mainly the backscattering region. However, the possible existence of anthropogenic particles in the ensemble of particles characterized by predominance of large and non-spherical particles can affect importantly the values of $-F_{12}/F_{11}$ with strong wavelength-dependence. These changes are critical in the 405 nm channel being $-F_{12}/F_{11}$ negative values like that observed for pollution (i.e. Bazo et al., 2024) and biomass-burning at 473 nm (i.e. Espinosa et al., 2017). The 515 and 660 nm channels show $-F_{12}/F_{11}$ patterns more typical of pure dust measurements at laboratory (Muñoz et al., 2007; Renard et al., 2010). Therefore, polarization measurements have great potential for distinguishing different aerosol types in the mixture, which can be either internal or external mixtures of dust with other types of particles. Some model simulations even suggest that non-absorptive coating in mineral dust has a drastic variation in the behavior of $-F_{12}/F_{11}$ (Zhang et al., 2022, 2023), such as coatings of non-absorptive aerosol due to the long-range transport (Dall'Osto et al., 2010). Future works will focus on detailed studies of chemical analyses in combination with polar nephelometry measurements to further exploit the potential of polarization measurements in aerosol studies.

## 5.2 Phase matrix simulations for different aerosol mixture scenarios

To fully understand how different degrees of mixture between anthropogenic particles and mineral dust can affect $F_{11}$ and $-F_{12}/F_{11}$, forward simulations with the Generalized Retrieval of Aerosol and Surface Properties algorithm (GRASP - Dubovik et al., 2014, 2021) have been performed. These simulations need inputs of different size distributions and refractive indexes to generate $F_{11}$ and $-F_{12}/F_{11}$. Particularly, we used bi-lognormal size distribution, one representative of fine mode particles with modal radius of 0.15 µm and 0.25 µm of standard deviation and the other representative of coarse mode particles with modal radius of 2.5 µm and 1 µm of standard deviation. Real refractive indexes were assumed non-spectrally dependent with values of 1.6 for fine mode and 1.55 for coarse mode. Imaginary refractive indexes were 0.0015 for fine mode and with no spectral dependency, while for coarse mode there were of 0.007, 0.005 and 0.005 for 405, 515 and 660 nm. The sphere fraction was also fixed for each mode, being 0.7 for fine mode and 0.05 for coarse mode. The modal radii selected are close to those observed for the particle size distribution

of the deposited particles in the UGR station (not shown for clarity). Moreover, the size distribution and refractive indexes selected can be considered representative of a mixture of anthropogenic pollution and dust (Torres et al., 2017). Three different scenarios were generated giving different weights to each mode: The first is for volume concentrations of 0.3 $\mu m^3/\mu m^3$ for each mode and can be considered as representative of a mixed case where both modes have a similar weight. The second presents more predominance of coarse mode (volume concentration of 0.5 $\mu m^3/\mu m^3$ for the coarse mode) but with non-negligible contribution of anthropogenic particles (volume concentrations of 0.1 $\mu m^3/\mu m^3$ for the fine mode). The last scenario is representative of pure dust mode (volume concentrations of 0.5 $\mu m^3/\mu m^3$) with negligible fine mode contribution (volume concentrations of 0.01 $\mu m^3/\mu m^3$). Results of computed $F_{11}$ and $-F_{12}/F_{11}$ are in Fig. 12.

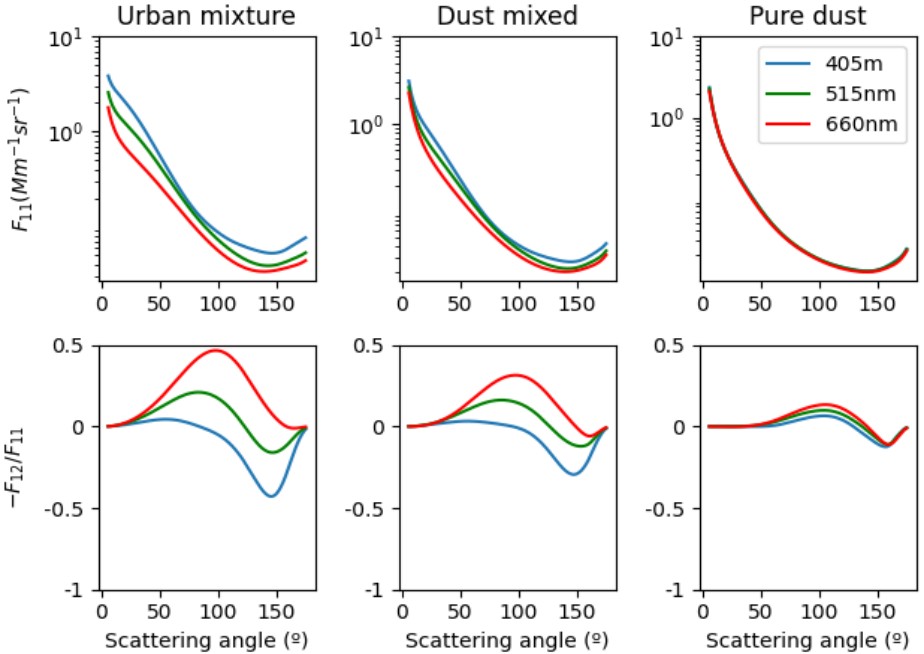

**Figure 12. Simulations of phase function ($F_{11}$) and polarized phase function ($-F_{12}/F_{11}$) using GRASP forward for three different combinations of bi-lognormal size distributions.** *Urban Mixture* **with approximately the same weight of fine and coarse mode,** *Dust mixed* **with predominance of coarse mode but with a non-negligible influence of the fine mode, and** *Pure Dust* **with strong predominance of coarse mode and negligible fine mode. Note that refractive indexes and sphericity of each mode are different.**

Figure 12 reveals important features in phase matrix elements depending on the mixture. For $F_{11}$, the patterns are generally characterized by larger forward scattering with minimums in the region of 120°-150°, independently on the type of aerosol mixture and on wavelength. The largest spectral dependencies are for the *Urban Mixture* case while for the other two cases such spectral dependencies become negligible. However, for $-F_{12}/F_{11}$ the largest variations in spectral dependencies and patterns are observed. In the *Urban Mixture* case the 660 nm channel shows a bell-shape pattern with maximum of ~ 0.4 in the region around 100°, while for 405 nm $-F_{12}/F_{11}$ shows approximately constants values close to zero until ~ 50° when it starts to decrease until the minimum of -0.5 in the region ~140°. Later it recovers reaching zero at 180°. On the other hand, for *Pure Dust* the $-F_{12}/F_{11}$ spectral dependencies are almost negligible, and it is characterized by a bell-shaped pattern with maximum around 0.2 in the region 110°-130°, with small negative values around 180º. Note that this feature is also present in the $-F_{12}/F_{11}$ in Figure 11 for the extreme dust cases, but it is not noticeable due to the scale. For the *Dust mixed* the pattern is in-between the previous ones.

Figure 12 results show how the presence of anthropogenic particles (fine mode) can alter the spectral dependencies in $-F_{12}/F_{11}$ when compared with only dust particles (coarse mode) in the sample, particularly in the blue channels. However, changes in the $F_{11}$ patterns were not so evident. These results help to understand the different phase matrix elements discussed in this manuscript, and their temporal evolutions during the extreme dust events (Supplementary Material). However, studying the relationships between measured $F_{11}$ and $-F_{12}/F_{11}$ with other aerosol optical and microphysical properties requires further analyses because $F_{11}$ and $-F_{12}/F_{11}$ ultimately depend on the size distribution, refractive indexes, and particle shapes. The problem is even more complex if we differentiate optical properties between fine and coarse mode.

Future optimization in GRASP will permit the retrieval of aerosol refractive indexes between fine and coarse mode separately using as inputs $F_{11}$ and $-F_{12}/F_{11}$, and thus permitting further analyses of the different study cases discussed in this work. It is important to mention that the super-coarse mode can also affect the behavior of $F_{11}$ and $-F_{12}/F_{11}$ and the presence of this mode is also observed for long-range transport (i.e. Renard et al., 2010). Future GRASP developments also need the consideration of this super-coarse mode. Another issue to study is the use of irregular-hexahedral for modeling the scattering of large and non-spherical particles that might reproduce better polarization signals (Saito and Yang, 2021, 2023; Saito et al., 2021).

## 6    Conclusions

This work has focused on the analyses of aerosol phase matrix elements and other optical properties during Saharan dust outbreaks that were registered in the UGR station (Southeastern Spain) in the year 2022. The main novelty of the analyses are the measurements by the multiwavelength Polarized Imaging Nephelometer (PI-Neph) developed by GRASP-Earth and capable of providing two aerosol scattering matrix elements ($F_{11}$ and $-F_{12}/F_{11}$) for three different wavelengths (405, 515 and 660 nm). The uniqueness of PI-Neph is that it allows to measure phase matrix elements of ambient aerosol. The optimization of the instrument and the data quality check applied served to obtain ($F_{11}$ and $-F_{12}/F_{11}$) with uncertainties below 10% and 20%, respectively. The multiwavelength $F_{11}$ and $-F_{12}/F_{11}$ measurements for different Saharan dust outbreaks are some of the first carried out for ambient aerosol and serve to complement laboratory measurements of mineral dust particles and of synthetic samples minerals that compose dust particles. The novel measurements of $F_{11}$ and $-F_{12}/F_{11}$ can also complement other optical and microphysical properties of Saharan dust already known from in-situ instrumentation and by active and passive remote sensing instruments, both from the ground and the space. Nevertheless, more $F_{11}$ and $-F_{12}/F_{11}$ measurements are needed at other experimental sites to have a more complete vision of mineral dust role on climate.

The analyses differentiate between two different scenarios: the first is two extreme Saharan dust outbreaks that happened on $15^{th}-16^{th}$ and on $25^{th}-26^{th}$ March 2022. These events were associated with intense low-pressure systems located in southern Algeria that favored the injection of dust in the atmosphere and the posterior transport to the Iberian Peninsula, producing extreme $PM_{10}$ concentrations. Actually, the peaks in $PM_{10}$ were 1800 $\mu gm^{-3}$ and 690 $\mu gm^{-3}$ for the $15^{th}-16^{th}$ and $25^{th}-26^{th}$ March, respectively, being both way over the daily limit value of 50 $\mu gm^{-3}$ delimited by the 2008/50/CE European Directive. The detailed temporal evolution analysis of $F_{11}$ and $-F_{12}/F_{11}$ for these extreme events revealed important features: $F_{11}$ did not show relevant changes with time showing the classical pattern for predominance of big and non-spherical particles characterized by high predominance of forward scattering and almost negligible wavelength differences. However, the patterns $-F_{12}/F_{11}$ showed variability during the different stages of the dust outbreaks: for 515 and 660 nm the $-F_{12}/F_{11}$ patterns were bell-shape centered around 100° and with slightly positive values. However, for 405 nm this bell-shape pattern was present only for the instants of extreme predominance of dust, while for the other instants $-F_{12}/F_{11}$ for 405 nm showed a very different pattern with values close to zero up to 50° - 60° followed by a decrease to values between -0.4 and -0.6 in the region around 120° and a final increase recovering to values close to zero in the backward region. These patterns were further analyzed using additional instrumentation in the UGR station and the typing classification concluded that the bell-shape patterns were typical for cases of only pure dust particles, where the others were associated with mixtures of dust particles with urban background aerosol. That differentiation could be explained with the temporal evolution of $F_{11}$ and $-F_{12}/F_{11}$ when the pure dust particles case corresponded to the peaks of the events while the mixtures happened during the entrance and withdrawn of the events. The variability in the aerosol mixture affected other optical properties such as lidar ratio (LR) that provided typical values for dust (~45 sr) during the peaks of intrusions and more variable values at other instants. Other extensive aerosol optical properties such as the single scattering albedo (SSA) revealed differences between the peaks of the intrusions (0.98 - 0.83, depending on wavelength) when compared to other instants with lower concentration (values between 0.87 - 0.72).

The second analysis scenario was thanks to the continuous operation of the PI-Neph in the period April – September 2022. We differentiated these intrusions from the previous two extreme events because they registered maximum $PM_{10}$ around 100 $\mu gm^{-3}$, which are more typical values for Saharan dust outbreaks in the UGR station. Additional instrumentation permitted measurements of other aerosol properties such as $\sigma_{sca}(\lambda)$, $\sigma_{abs}(\lambda)$, SAE, AAE $g(\lambda)$, $Bs(\lambda)$, $SSA(\lambda)$ and $LR(\lambda)$. The selection of the cases affected by Saharan

dust transport was done by keeping only data with scattering SAE below 1.0. This allowed to include a large variety of situations reflected in the variability of intensive parameters such as SSA or LR. The combination of scattering and absorption measurements classified most of the data for this entire period as mixture of dust particles with other anthropogenic aerosol particles. The analysis of $F_{11}$ and $-F_{12}/F_{11}$ revealed again important features. $F_{11}$ mostly followed the classical pattern characterized by strong forward scattering, although some cases showed some spectral dependence on 405 nm depending on the influence of fine mode particles in the mixture. However, the analysis of $-F_{12}/F_{11}$ revealed big differences among wavelengths, being 405 nm characterized by a well-defined pattern with values close to zero up to 50°-60° and negative values for the rest of scattering angles, while for the other channels at 515 and 660 nm they show the bell-shape pattern but with larger variability compared to the extreme dust cases, probably associated with the complexity of the mixture of fine (anthropogenic) and coarse (dust) mode particles. These patterns in $F_{11}$ and $-F_{12}/F_{11}$ agreed with those observed in the intense Saharan dust outbreaks in March for the instants when the dust was entering and withdrawing.

Laboratory measurements of mineral dust samples available in the Granada-Amsterdam Light Scattering Database allowed a comparative assessment of the different Saharan dust outbreaks that affected the UGR station. To that end, averages of the period April – September 2022 plus the peaks of the extreme events during March 2022 were used. For comparisons, $F_{11}$ were normalized with respect to $F_{11}(30°)$. The results showed that for all $F_{11}$ the differences between temporal averages, peak events and laboratory measurements were minimal, being only notable in the backscattering region close to 180° where according to the T-Matrix theory more sensitive to aerosol particle parameters is found. For $-F_{12}/F_{11}$ laboratory measurements and extreme events measurements (peak of concentration) agree quite well both for 405 and 660 nm, being the differences within the uncertainties. However, when comparing with the seasonal averages for the period April – September some important features were revealed: for 660 nm it seems to reproduce the same pattern than for laboratory/extreme event measurements, although with large standard deviations when compared with other wavelengths. But for 405 nm the seasonal mean has a very distinct pattern characterized again by values close to zero in the region up to 50°-60° and negative values for the rest of angles. Standard deviations are now considerably lower. The channel at 515 nm shows an intermedia situation. Considering that laboratory measurements consist of pure dust samples directly collected in the dessert, we can conclude that the $-F_{12}/F_{11}$ at 405 nm measured in the laboratory is only reproduced when there are extreme concentrations of dust in the atmosphere, while the contribution of anthropogenic particles in the mixture the $-F_{12}/F_{11}$ affects critically to $-F_{12}/F_{11}$. For the other channels, particularly 660 nm, $-F_{12}/F_{11}$ seems to be less critically affected by the contribution of anthropogenic particles.

Simulations performed by the GRASP code for different mixtures of fine mode (anthropogenic particles) and coarse mode (dust particles) revealed that $F_{11}$ and $-F_{12}/F_{11}$ are sensitive to the different contribution of each mode in the mixture, being especially critical for $-F_{12}/F_{11}$ on the 405 nm channel. The negative values for $-F_{12}/F_{11}$ in 405 nm were observed more clearly for the mixture of fine and coarse particles. Thus, these simulations have served to understand the experimental negative values in $-F_{12}/F_{11}$ not observed in laboratory measurements for collected dust. Retrievals of bimodal size distribution with separate refractive indexes for each mode would have shown clarity to this problem. Hower, such retrieval with GRASP using $F_{11}$ and $-F_{12}/F_{11}$ as inputs needs to be optimized. Another additional optimization in GRASP will imply the possibility of implementing the retrieval of super-coarse mode particles. The possibility of implementing the irregular-hexahedral model would be also ideal to better understand polarization patterns. Nevertheless, the possibility of explaining the spectral differences in $F_{11}$ and $-F_{12}/F_{11}$ with wavelength has served to understand the temporal evolution of the extreme dust events and the difference and similitudes when comparing versus laboratory measurements and versus other more moderate events of Saharan dust transport. However, going further in understanding the interaction of dust with these anthropogenic particles requires further analyses that provide the chemical composition and size distribution of the ensemble of particles and the final composition and shape of the particles after interacting. This is planned in future studies that will allow a more complete comprehensive analysis. We therefore believe that multiwavelength polarized polar nephelometry opens new possibilities in the studies of mineral dust role in the climate system.

**Author's contributions**

EB analyzed the data and wrote the manuscript. DPR defined the structure of the paper, conceptualized the
895 investigation and supervised the writing of the manuscript. ADZ analyzed the meteorological conditions
during the extreme Saharan dust outbreaks. FR performed the GRASP simulations. FJO, AV and LAA are
the principal investigators of the projects that funded the research and put the guidelines of the research.
GT, AC, DP, FJGI assisted in the conceptualization. JVM and DF contributed to the development of the
instrumentation. All authors contributed to the discussion of the results and provided comments on the
900 paper.

**Acknowledgments**

This work was supported mainly by the Horizon Europe program under the Marie Sklodowska-Curie Staff
Exchange Actions with the project GRASP-SYNERGY (grant agreement No 101131631). The work was
905 also funded by the European Union's Horizon 2020 research and innovation program through projects
ACTRIS.IMP (grant agreement No 871115) and ATMO_ACCESS (grant agreement No 101008004), and
by the Spanish Ministry of Science and Innovation through projects ELPIS (PID2020-12001-5RB-I00),
MULHACEN (PID2021-128008OB-I00), NUCLEUS (PID2021-128757OB-I00) funded by
MICIU/AEI/10.13039/501100011033 and by ERDF A way of making Europe, and ACTRIS-España
910 (RED2022-134824-E), and by University of Granada Plan Propio through Excellence Research Unit Earth
Science and Singular Laboratory AGORA (LS2022-1) programs. E. Bazo received funding by
MICIU/AEI/10.13039/501100011033 and the ESF + through FPI fellowship PRE2022-101272. F.J.
García-Izquierdo acknowledges financial support from the grant PID2021-123370OB-I00 (CATS) funded
by MCIN/AEI/10.13039/501100011033. We are very thankful to Air Quality Service of Junta de Andalucía
915 for supplying the PM10 data.

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
