# Peer review of "Phase matrix characterization of long-range transported Saharan dust using multiwavelength polarized polar imaging nephelometry"

_EGUsphere, 2024_

## Author Comment (AC3)

Response to Reviewers

The authors greatly acknowledge the anonymous reviewer for carefully reading the manuscript and providing constructive comments. This document contains the author's responses. Each comment is discussed separately with the following typesetting:

**Reviewer's comments**

Authors response

Changes in the manuscript

**The manuscript presents ambient measurements of Saharan dust intrusions at ground level in Southern Spain. The polarized imaging nephelometer enables the measurement of the phase function and polarized phase function and at several wavelengths at scattering angles between 5° and 175°. Especially the polarized phase function is an important quantity in passive remote sensing. They could show with their spectral measurements, that the polarized phase function behaves similar in the observed wavelength range (405 till 660 nm) for intense dust events. Whereas dust which was mixed with local pollution shows a different spectral behavior, especially at the shortest wavelength of 405 nm. Additionally, simulated (polarized) phase functions are shown. However, I see that for a publication in ACP, the research findings should be set into a broader context and the novelty of the study should be worked out stronger. After addressing my comments below, and please take my major comments serious, a publication in ACP can be recommended.**

We thank the reviewer for the comments addressed that helped to improve the quality of the manuscript. Also, following the suggestion of other referees, we have re-written the abstract to highlight the objective and findings of our manuscript

[revised manuscript text omitted]

**Major comments:**

**The paper would benefit a lot, if you include lidar observations to your dust cases. I know that the University of Granada operates a polarization lidar (otherwise I would not ask for it) which could clearly show the Saharan dust layers above the station (Sect. 3.1). Additionally, it may provide you with directly measured lidar ratios. Then, there is just the gap between the PI Nephelometer observations at the ground and the lidar measurements in the lofted Saharan dust layers. So please include your colleagues from the lidar group with their lidar observations at UGR.**

-We agree with the reviewer that PI-Neph measurements can complement lidar measurements available in the University of Granada station. The lidar system in our station (MULHACEN) is part of the EARLINET/ACTRIS network. We have been continuously in contact with our colleagues that are in charge of the lidar system MULHACEN and there are several reasons why such kind of measurements are not included in the present manuscript.

The main reason is the lack of appropriate lidar signals during the extreme Saharan dust outbreaks. We include in the Figure below (Figure R1) the temporal evolution of lidar Range Corrected Signal (RCS) at 532 nm. The RCS reveals that the main aerosol layers are found below 1.5 km approximately. However, when analyzing these RCS we found out that they were saturated and no retrievals of backscattering and extinction coefficients were possible. Moreover, our lidar system had a large region of incomplete overlap from

the ground up to 1.3km, and thus the intense dust layer could not have been monitored even if signals would not have been saturated. Actually, only one profile fulfilled data quality criterion of EARLINET single calculus chain, and the results are illustrated here in Figure R2.

[Figure]

Figure R1: Temporal evolution of the Range Corrected Signal (RCS) at 532 nm on 15th March 2022 in the EARLINET/ACTRIS lidar system operating in AGORA.

[Figure]

Figure R2: Vertical profiles of (a) aerosol backscattering coefficient at 355 and 532 nm obtained with the MULHACEN system plus that at 1064 nm obtained with a CHM15k Ceilometer – that assumes constant lidar ratio (b) backscattering Angström Exponent (AE) between 532 and 355 nm and (c) particle and volume depolarization.

The profiles of Figure R2 clearly show the lack of capacity for obtaining aerosol properties below 1.3 km approximately. Retrieval of backscattering at 355 nm seems to reach negative values above 5.5 km approximately that suggest the difficulties of finding a good signal in the clean zone, which is needed in the lidar retrieval. These difficulties in the retrieval can be behind the noisy retrievals of AE, but even though AE values above 4 km did not reveal large dependence of coarse particles. Moreover, the values of δp and δv do not correspond with values of large dominance of coarse particles, agreeing with AE values. Therefore, the lidar measurements by the MULHACEN system on 15th March

2022 were not useful to complement the PI-Neph measurements at surface. For the case on 25[th] March we found very similar situation, as the RCS at 532 nm show in Figure R3

[Figure]

Figure R3: Temporal evolution of the Range Corrected Signal (RCS) at 532 nm on 25[th] March 2022 in the EARLINET/ACTRIS lidar system operating in AGORA.

For the rest of PI-Neph data shown in Figure 8 there are certainly some correlative MULHACEN measurements. But the main limitation is in the large zone of incomplete overlap for the MULHACEN system that as commented include always the first kilometer (in some cases up to 1.5 km from the ground). That makes not possible a direct comparison between lidar ratios obtained by the two instruments. Only cases with very well mixed conditions could serve in those comparisons.

The main objective of this study is to discuss similitudes and differences of aerosol phase matrix for two extreme Saharan dust outbreaks versus those obtained during more frequent intrusions in the AGORA stations. The measurements with the PI-Neph are in the range 5-175º which is a very wide range, while lidar measurements only refer to values exactly at 180º. Thus, we believe that the lack of appropriate lidar measurements for intercomparisons does not affect the overall objective of our study. Nevertheless, we are aware of the benefit of lidar measurements with phase matrix measurements with the PI-Neph. In this sense, the research team of the AGORA station is implementing the appropriate approach to make the intercomparisons between LRs obtained by PI-Neph and those by lidar. To do so, a special field campaign called LUMINOUS was carried out in summer 2024. LUMINOUS was partly funded by ACTRIS Transnational Access and is an international cooperation between the University of Granada, the Paul Scherrer Institute (Switzerland) and GRASP-Earth. In LUMINOUS new designs of Inverse Multi-Angular Polarimeter with Polarization (IMAP - https://www.grasp-earth.com/imap/) were also deployed in the AGORA stations. One was in the UGR station (where MULHACEN operates) and the other in the Sierra Nevada station at 2.5 km altitude and at 20 km in straight line to the city of Granada. In LUMINOUS we operated other instruments for measuring particle size distribution and chemical compositions. The approach used in LUMINOUS avoids the problems of incomplete overlap and a direct intercomparisons between lidar and in-situ instruments in Sierra Nevada. At the moment, we are analyzing the first results from LUMINOUS.

In the present manuscript, the idea behind showing LRs in Figure 8 was to illustrate the variability of this parameter for different mixtures of mineral dust and

anthropogenic particles. After the referee questions, we believe that this was not emphasized in the manuscript and have modified the text to highlight this variability, and that further analysis is needed that incorporates also lidar measurements. This has been clarified in the revised manuscript when defining LRs computations using PI-Neph data. Also, we highlight that the exact measurements of $F_{11}(180^{\circ})$ required the use of specific instrumentation. Now between lines 265-270 is written:

The computation of $F_{11}(180°)$ has been made using the interpolation method used for completing the entire angular range in $\sigma_{sca}$. Other more robust methods can be used (i.e. Gomez-Martin et al., 2021), that can imply differences in $F_{11}(180°)$ of up to 20-30%. Therefore, LRs estimations will serve as an illustration of how this parameter varies under different conditions. We highlight that PI-Neph is not designed to accurately measure $F_{11}(180^{\circ})$ and there are other specific instruments that serve for that purpose (Järvinen et al., 2016; Miffre et al., 2023; Sakai et al., 2010).

In the discussion of the extreme event on 25th March 2022 we clarify that the data of LRs serve as an illustration of the mixture of particles (L406-407):

"Thus, the hypothesis that after 14:00 the presence of pollution particles becomes more relevant implies a decrease in SSA and illustrates variability in LRs, particularly in 405 nm."

Also, when discussing LRs for the entire period April-September 2022 we highlight that our results serve only as illustration of LRs variability (L554-566).

[revised manuscript text omitted]

**2.      Do you have additional size distributions for your cases? The spectral slope of your scattering properties might depend on particle size. Having an additional size distribution would add certainly a lot of value to your discussion. Could you estimate the amount of pollution in your polarization measurements? It would be an important information in quantifying your observations. By requesting additional size distributions and lidar (lidar ratio) observations underlines my request to broaden your field and take all available information into account to deliver a comprehensive picture and to place your results in a broader context.**

-We completely agree with the reviewer. Particle size distribution measurements would have solved many of the challenges in the analyses of aerosol phase matrix measurements. The AGORA observatory has an Aerosol Particle Sizer (APS TSI 3321) and an Aerosol Chemical Speciation Monitor (ACSM - Aerodyne). However, during the two extreme Saharan dust outbreaks in March 2022 these two instruments were not available to operate at the station due to maintenance tasks, that got extended until summer 2022. That is why these important measurements were not available for this study.

For the extreme event on 15th-16th March 2022 we measured the size distribution for the deposited dust (Figure R4). There is very low super-coarse mode if compared to the fine and coarse modes. Additionally, the modal radius of the coarse mode agrees with the aerosol size distribution used in GRASP simulations, which served as support to our simulations. This statement was added to the revised manuscript (L730-L731)

"The modal radii selected are close to those observed for the particle size distribution of the deposited particles in the UGR station (not shown for clarity)."

[Figure]

Figure R4. Number size distribution measured for the deposited dust for the extreme dust event on 15th – 16th March 2022.

The use of polarization measurement in GRASP also has a strong potential to differentiate pollution particles in the ambient samples affected by Saharan mineral dust. The challenge is optimizing GRASP to perform a retrieval capable of separating refractive index and other aerosol optical properties between fine and coarse mode. This is already available for the combination of backscattering lidar plus AERONET sun photometry measurements, but not for a configuration using only $F_{11}$ and $-F_{12}/F_{11}$ as inputs. The simulations carried out only illustrate that the patterns observed in section 3.2.4 – now section 5.2 in the revised manuscript. Now, in the revised manuscript is given (L763-L768):

"However, studying the relationships between measured $F_{11}$ and -$F_{12}/F_{11}$ with other aerosol optical and microphysical properties requires further analyses because $F_{11}$ and -$F_{12}/F_{11}$ ultimately depend on the size distribution, refractive indexes, and particle shapes. The problem is even more complex if we differentiate optical properties between fine and coarse mode. Future optimization in GRASP will permit the retrieval of aerosol refractive indexes between fine and coarse mode separately using as inputs $F_{11}$ and -$F_{12}/F_{11}$, and thus permitting further analyses of the different study cases discussed in this work."

Our group is collaborating with GRASP developers to implement such an approach in GRASP, and although preliminary results are promising, we need further analyses. Our goal is to prepare a future work with this new GRASP implementation and analyze our entire PI-Neph dataset – where we will analyze also cases not affected by Saharan dust particles.

Given the importance of the points raised by the referee (the other referees also pointed out very similar issues), we have decided to modify the conclusion section to emphasize the needs of acquiring correlative measurements of PI-Neph with additional instrumentation that provide information on particles size distribution and chemical compositions. Moreover, we emphasize the need for further advancing in GRASP for retrieving bimodal size distribution differentiating optical properties (especially refractive index) between fine and coarse modes using PI-Neph data (L849-864).

"Simulations performed by the GRASP code for different mixtures of fine mode (anthropogenic particles) and coarse mode (dust particles) revealed that $F_{11}$ and -$F_{12}/F_{11}$ are sensitive to the different contribution of each mode in the mixture, being especially critical for -$F_{12}/F_{11}$ on the 405 nm channel. The negative values for -$F_{12}/F_{11}$ in 405 nm were observed more clearly for the mixture of fine and coarse particles. Thus, these simulations have served to understand the experimental negative values in -$F_{12}/F_{11}$ not observed in laboratory measurements for collected dust. Retrievals of bimodal size distribution with separate refractive indexes for each mode would have shown clarity to this problem. Hower, such retrieval with GRASP using $F_{11}$ and -$F_{12}/F_{11}$ as inputs need to be optimized. Another additional optimization in GRASP will imply the possibility of implementing the retrieval of super-coarse mode particles. Nevertheless, the possibility of explaining the spectral differences in $F_{11}$ and -$F_{12}/F_{11}$ with wavelength has served to understand the temporal evolution of the extreme dust events and the difference and similitudes when comparing versus laboratory measurements and versus other more moderate events of Saharan dust transport. However, going further in understanding the interaction of dust with these anthropogenic particles requires further analyses that provide the chemical composition and size distribution of the ensemble of particles and the final composition and shape of the particles after interacting. This is planned in future studies that will allow a more complete comprehensive analysis."

**3.** **You have not written a lot concerning your uncertainty estimates. In Tab. 1 you provide uncertainties without mentioning what you are reporting. I guess, it is the standard deviation of the hourly mean, but it is stated nowhere. If it is the case, I am wondering about the systematic uncertainties of your measurements. Probably, it is reported elsewhere. But we need an assessment of the systematic uncertainties. Otherwise, the results are not comparable to other measurements.**

-The reviewer is correct, the uncertainties in Table 1 correspond to the standard deviation of the hourly averages. We have added this information to the text (L422-423).

"Error bars in Table 1 are the standard deviations of the hourly mean values."

And modified Table 1 caption that now is given as (L513-516):

"Table 1. Hourly averaged properties of different stages of the extreme dust outbreaks in March 2022 reported in Figs. 6 and 7. The properties are reported at three wavelengths in the order of 660, 515 and 405 nm from top to bottom. Only the angular range of the PI-Neph (5°-175°) is used as the integration range of $\sigma_{sca}$. Error bars correspond to the standard deviations of the hourly means."

The issues about the uncertainties of the measurements have been also pointed out by the previous referee and by the comments of Jean-Baptiste Renard. In our previous study (Bazo et al., 2024), we performed an extensive characterization of the PI-Neph and error analyses. To avoid duplicate answer, we encourage the referee to read the overview of the study in Bazo et al., (2024) given to referee 1 – it should be published in the section public discussion. We highlight that in the new description of the PI-Neph we have given an overview of the error sources and uncertainties, plus the data quality check procedure. Now, between L199 and L225:

"An extensive analysis of the error sources in the PI-Neph was performed in Bazo et al., (2024) but an overview is given here: an exhaustive calibration of the instrument is performed consisting of two different steps. The first is a geometric correction that corrects from the different light paths to the different pixels in the CMOS camera. Later the absolute calibration permits to obtain phase matrix elements in physical units. In each step we used known scatterers ($CO_2$ and particle free air) whose parallel and perpendicular signals can be computed analytically using the Rayleigh theory (Anderson et al., 1996). Evaluation of the calibration with time did reveal great stability (variations around 3%). Instrument stability was evaluated with $CO_2$ measurements at a constant flow rate of 10 Lmin$^{-1}$ during 15 min. These measurements revealed constant values of scattering coefficients with differences below 1% versus theoretical values from (Bodhaine et al., 1991). Finally, inherent aspects of the imaging technique were evaluated such as the impact of the exposure time. The largest noise is found for exposure times below 5 s, while the smoother values are obtained for exposure times of 10-20 s. However, large exposure times can yield to more angles that are saturated, and the software must find a compromise between noise and saturation. Thus, the typical exposure time is of 10 s and with that we estimate that uncertainties in measured parallel and perpendicular signals are around 5% in laboratory conditions. The evaluation of the instrument versus known scattered (monodisperse polystyrene spheres - PSL) showed good agreements with RMSE around 0.10 for both $F_{11}$ and $-F_{12}/F_{11}$.

The uncertainties in direct measurements of the instrument (parallel and perpendicular signals) under laboratory conditions imply uncertainties below 10% in $F_{11}$ and below 20% in $-F_{12}/F_{11}$. However, in-situ measurements present natural variability of the aerosol sampled and the differences can be enhanced because of the short exposure times ($\sim$ 10s). Effects during the measurements such as saturation or low signal to noise ratios (SNR) of some pixels can happen. Other issues such as the passage of an individual

super-coarse particle can have an impact on certain angles of the phase matrix. Therefore, we apply a data-quality check procedure that accounts for all these issues and provide an effective phase matrix representative of an average time of 30 min or 1 hour, depending on the specific conditions of natural aerosol variability. Note that standard deviations during these periods might be larger than the uncertainties of the instrument. Details of this quality check procedure are in Bazo et al., (2024).”

**In Tab. 2 you don't provide any uncertainties at all. Please add them.**

-Thanks for pointing this out. In the revised manuscript we include now the standard deviations of the hourly means in Table 2, as a representation of the variability of the aerosol particles being sampled. We have also added in the Table caption that error bars are the standard deviations of the hourly means (L589):

“Error bars are the standard deviations of the hourly means.”

**Your phase matrix elements (Fig. 6,7 & 9) do not contain any uncertainties. Putting an error bar to every point would certainly overload the figure, but having at least 3 points with error bars in each plot would help to assess the range of uncertainty.**

We thank the referee for the suggestion. In Figure R5 we show some examples of the hourly phase function and polarized phase function with the standard deviation as error bars in several angles, including angles in the forward (~5-10º) and backward (~150-175º) scattering as well as in the middle (~90-100º) of the angular range. We have chosen to represent the standard deviation of the hourly averages because these values are larger than the uncertainty of the instrument. However, one must keep in mind that these standard deviations represent the variability of the different parcels of air that are being sampled throughout the hour, and they are not in any way uncertainty of the PI-Neph's measurements. We observe that for less variable conditions, such as a clean atmosphere or a highly polluted atmosphere, the standard deviations are smaller than those obtained during situations when the air is changing. In any case, the standard deviation in the phase functions is around 20% of the hourly mean, whereas for the $-F_{12}/F_{11}$ element the values of the standard deviations are more variable, usually ranging from 0.05-0.1 in the forward region and 0.1-0.2 in the other regions.

[Figure]

Figure R5. Examples of hourly phase functions (F11, top) and polarized phase functions (-F12/F11, bottom) with the standard deviation as error bars.

As already mentioned, the standard deviations in the scattering matrix elements are only a representation of the variability of the atmosphere that surrounds the sampling line connected to the instruments. Therefore, since it does not represent the uncertainty of the measurements, we have not added any error bars to Figures 6,7 and 9. We refer to the reader to our previous study (Bazo et al., 2024) where we evaluated the performance of the instrument and studied its sources of error (see modified text in previous comment) in laboratory under controlled conditions.

And we have also indicated the usual range of variability of the scattering matrix elements shown in this work (L415-L419):

"Computed standard deviations were larger than instrument uncertainties and they are associated with the variability of the different parcels of air sampled throughout the hour of measurements. These standard deviations were of ~ 20% for $F_{11}$ and ranging between 0.1 and 0.2 for $-F_{12}/F_{11}$ (minimums for the forward region and maximums in the middle region around 90º)."

L463-L465: "Again, the computed standard deviations are larger than the uncertainties of the instruments and they represent the variability of the samples. As for the previous extreme event, these standard deviations are ~20% for $F_{11}$ and between 0.1- 0.2 for $-F_{12}/F_{11}$. "

L568-L570: "The standard deviations were 20-30% for $F_{11}$ and around 0.2 in $-F_{12}/F_{11}$, which are larger than the uncertainties of the instruments for all cases and explained by the large variability of aerosol samples during the measurement process"

**4. After finalizing my review, I am now reading the comments of Jean-Baptiste Renard and I want to enforce his point, that the (spectral) scattering properties depend on particle size (see my major comment #2). And there is much more literature on the size effect of the polarized scattering properties than mentioned in his comment. The size effect is almost not discussed in the manuscript. Please try seriously to get more size information than just a PM10 concentration.**

We agree with the referee and with Dr. Jean-Baptiste Renard. Without information of size distribution, we have to be cautious in the interpretation of our results. As commented above, correlative measurements of aerosol size distributions were not available specially during the two extreme Saharan dust outbreaks. In the LUMINOUS field campaign, we are acquiring such kind of correlative data polarized polar nephelometry + size distributions + chemical information.

After reading all the issues raised by the referees, we believe that we have been naive in the interpretation of the phase matrix results. The patterns observed can be associated with many causes – differences in the size distribution, size of particles and refractive indexes – that were not measured correlatively. This is even more complicated when we study mixtures of different particles. In the revised manuscript we have avoided any hypothesis and remark that further correlative measurements are needed.

In the conclusion section, we insist that our novelty is on the measurement of phase matrix elements of ambient aerosols, and that we present some of the first measurements for ambient Saharan dust (L772-L785).

"This work has focused on the analyses of aerosol phase matrix elements and other optical properties during Saharan dust outbreaks that were registered in the UGR station (Southeastern Spain) in the year 2022. The main novelty of the analyses are the measurements by the multiwavelength Polarized Imaging Nephelometer (PI-Neph) developed by GRASP-Earth and capable of providing two aerosol scattering matrix elements ($F_{11}$ and $-F_{12}/F_{11}$) for three different wavelengths (405, 515 and 660 nm). The uniqueness of PI-Neph is that it allows to measure phase matrix elements of ambient aerosol. The instrument can provide $F_{11}$ and $-F_{12}/F_{11}$ with 10% and 20% uncertainty, respectively, under laboratory conditions. The optimization of the instrument and the use of appropriate data quality check approach served to continuously measure $F_{11}$ and $-F_{12}/F_{11}$ for ambient air, but in these cases the natural variability of the air sampled typically imply large uncertainties, being the typical standard deviations of ~20% for $F_{11}$ and between 0.1- 0.2 for $-F_{12}/F_{11}$ and therefore larger than the uncertainties of the instrument. The multiwavelength $F_{11}$ and $-F_{12}/F_{11}$ measurements for different Saharan dust outbreaks are some of the first carried out for ambient aerosols and serve to complement laboratory measurements of mineral dust particles and of synthetic samples minerals that compose dust particles"

We also highlight that our measurements serve to complement other already measurements by in-situ and remote sensing instruments (L785-L789).

"The novel measurements of $F_{11}$ and $-F_{12}/F_{11}$ can also complement other optical and microphysical properties of Saharan dust already known from in-situ instrumentation and by active and passive remote sensing instruments, both from the ground and the space.

Nevertheless, more $F_{11}$ and -$F_{12}/F_{11}$ measurements are needed at other experimental sites to have a more complete vision of mineral dust role on climate."

And we remark that further experiments are needed combining PI-Neph with other instruments that provide chemical information and size distribution of aerosol particles (L860-L864).

"However, going further in understanding the interaction of dust with these anthropogenic particles requires further analyses that provide the chemical composition and size distribution of the ensemble of particles and the final composition and shape of the particles after interacting. This is planned in future studies that will allow a more complete comprehensive analysis."

**Minor comments:**

• **Title: "polarized polar imaging nephelometry" – in the manuscript, you always state Polarized Imaging Nephelometer (PI-Neph). Why do you use the term "polar" only in the title?**

-Thanks for the suggestion. The name of the instrument is as written in the manuscript (Polarized Imaging Nephelometer), but we wanted to highlight the angular resolution of the instrument in the title, so we added the term polar to it.

• **L20 Please mention already in the introduction the angular range of the instrument.**

-Done.

• **L56 a strange unit.**

-We refer to the inverse of the climate sensitivity parameter, which expresses the change in temperature per unit of radiative forcing. We have changed the notation of the units in the main text to make it clearer.

• **L67 complex refractive index**

-Thanks. It has been corrected.

• **Your introduction (L70-85) is focused on passive remote sensing. I am missing the active remote sensing with e.g., CALIPSO or EarthCARE and the linked laboratory studies which focus on the backscattered light close to 180° (e.g., Sakai et al., 2010, Järvinen et al., 2016 or Miffre et al., 2023).**

We thank you for this suggestion. We have modified the introduction to better contextualize our work (see major comment above). For space measurements, polarization only affects the retrieval of aerosol properties for space polarimetry and that has been stated in the revised introduction. Including references to EarthCare or CALIPSO would have made the introduction more confusing because PI-Neph does not provide measurements at 180º that is the angle required for lidar measurements. In Section 2.2.2 where we explain how to compute the LRs we mention that PI-Neph is not the ideal instrument to measure LRs. Indeed, other instruments are more appropriate (L268-L270):

"We highlight that PI-Neph is not designed to accurately measure $F_{11}(180º)$ and there are other specific instruments that serve for that purpose (Järvinen et al., 2016; Miffre et al., 2023; Sakai et al., 2010)."

We have also highlighted in the conclusion section that PI-Neph measurements serve to complement other measurements from in-situ instrumentation and by active and passive remote sensing instruments (L785-L788):

"The novel measurements of $F_{11}$ and $-F_{12}/F_{11}$ can also complement other optical and microphysical properties of Saharan dust already known from in-situ instrumentation and by active and passive remote sensing instruments, both from the ground and the space."

- **Eq 5 + 6: Do you use the decadic logarithm (log) or the natural logarithm (ln) for your definition?**

-Thanks for pointing this out. We use the natural logarithm; we have changed the notation in equations 5 and 6.

- **According to eq. (7), the units of the LR should be sr and not sr-1. Please change it throughout the manuscript (including figures and tables).**

-We thank the reviewer for identifying this mistake, we have changed it throughout the manuscript and in the figures and tables.

- **How do you extrapolate to the scattering angle of 180°. Please add some description and assessment of the related uncertainties.**

-The objective is not to compute $F_{11}(180°)$. Indeed, we want to estimate $F_{11}$ for the range 175º - 180º. To do so we used methodology based on linear extrapolation using the neighbors measurements. According to studies of Horvath et al., (2015), these linear extrapolations only imply 5% uncertainty in the computation of $\sigma_{sca}$, g and $B_s$. We have modified the text to clarify this point (L234-L235):

"… where data from 0 to 5° and from 175 to 180° have been linearly extrapolated to obtain the complete phase function which according to Horvath, (2015) only implies uncertainties up to 5% in the computations of the $\sigma_{sca}$, g and $B_s$. "

- **From the reader's perspective, I would suggest a different section numbering to avoid the 4th level of subsections (e.g., 3.2.1.1). My suggestions are: Give the meteorological conditions (currently Sect. 3.1) an own section (Sect. 3) before you go to your results of the PI-Neph in Sect. 4 (Results or Aerosol phase matrix from different dust scenarios). Section 3.2.3 and 3.2.4 can be combined to new section: Sect. 5 – Discussion with 5.1 and 5.2 for the two respective subsections.**

We thank the referee for this feedback. We have re-shaped the manuscript in the following way:

Section 3 'Overview of extreme dust events during March 2022' that initially was sub-section 3.2.1.1. This section gives an overview of the meteorological conditions associated with these two extreme events, plus some satellite observations that give an overview of the intensity of the dust plume.

Section 4 'Results of aerosol phase matrix from different dust scenarios': Here we include now sub-section 4.1 'Extreme events' that was initially subsection 3.2.1.2, and sub-section 4.2 'Moderate dust events during spring/summer 2022' that was initially subsection 3.2.2

Section 5 'Discussion': Here we include the sub-section 5.1 'Comprehensive assessment of the different dust events' that initially was sub-section 3.2.3 and sub-section 5.2 'Phase matrix simulations for different aerosol mixture scenarios' that initially was sub-section 3.2.4

• **The meteorological conditions (Sect. 3.1) are given in great level of detail. I am wondering, if the 4 CAMS model outputs for each case are necessary, because this information is not used in the next sections. To my opinion, it can be combined to one CAMS output per dust case. In this section some lidar observations of the Granada station would be helpful to demonstrate the vertical layering of the dust above your station. It must not be a difference in the strength of the dust outbreak between the two cases in March, but on 15 March, the dust was mixed to a larger extend towards the ground and therefore to your PI-Nephelometer. Lidar observations would reveal the vertical layering of the dust.**

-We plotted the evolution of each event with the CAMS model in two separate graphs with the objective of illustrating the temporal evolution of the event and highlighting the moment of maximum intensity and spatial impact. We also analyzed the impact of the different events in terms of Aerosol Optical Depth measured by AERONET network.

The referee is right that lidar measurements would have given a great compliment. But as commented above, lidar measurements in the AGORA observatory were saturated in the first 1-2 km and did not fulfill the data-quality criterion of EARLINET/ACTRIS single calculus chain. Nevertheless, these measurements served as illustration that most transport of mineral dust during these events happened in the first 1-2 km, and that was added in the text (L334-L337):

"Lidar measurements at EARLINET\ACTRIS Granada station were saturated in the first 1-2 km and avoided any kind of retrieval of aerosol optical properties. Nevertheless, these measurements served to illustrate that most of the transport occurred in the first two kilometers above the ground."

• **L200 Libya**

-Done.

• **L250 Please provide the values for usual dust outbreaks and not just a reference. Similarly, in line 373**

Thanks for the suggestion. This value is of ~100 μgm$^{-3}$ and it is now specified in the revised manuscript.

• **Fig. 5 Please add more detailed steps to the time axis. The scale for SAE is not optimal.**

-Thanks for pointing this out. The axis and scale have been modified to make the Figure clearer.

- **L329-334 This text might be moved to the figure caption of Fig. 6. At least Fig. 6 needs some more explanations in the caption. The same holds for L 374-377 and Fig. 7.**

-We thank the referee for the suggestions. Figure 6 and Figure 7 captions have been modified in the revised manuscript:

L455-459: "Figure 6. Hourly averages of phase function ($F_{11}$) and polarized phase function (-$F_{12}$/$F_{11}$) on 15th - 16th March 2022 for four different stages of the evolution of the extreme Saharan dust outbreak: (a) 15th March 07:00 UTC before the Saharan dust outbreak reached the station, (b) 15th March 12:00 UTC when the Saharan dust begin to reach the station, (c) 15th March 17:00 UTC associated with the peak of the extreme Saharan dust intrusion, and (d) 16th March 13:00 UTC when Saharan dust start to withdrawn."

L493-497: "Figure 7. Hourly averages of phase function ($F_{11}$) and polarized phase function (-$F_{12}$/$F_{11}$) on 24th - 25th March 2022 for four different stages of the evolution of the extreme Saharan dust outbreak: (a) 24th March 13:00 UTC before the Saharan dust outbreak reached the station, (b) 24th March 21:00 UTC when the Saharan dust start to reach the station, (c) 25th March 09:00 UTC associated with the peak of the extreme Saharan dust outbreak, and 25th March 20:00 UTC when dust concentrations begin to withdrawn"

- **L340 "notable spectral separation" – I am not sure if it just a manner of scaling the y-axis. Fig. 6a1 has a maximum value of 10^3 whereas the other subfigures extend to 10^4. Therefore the spectral separation is better visible in Fig. 6a1.**

-The referee is right, the most remarkable spectral separation in $F_{11}$ values is in Fig 6.a1. This point is even clearer if we make a zoom in Figure 6, as it is shown in Figure R6 showed below.

[Figure]

Figure R6. Zoom of the phase functions ($F_{11}$) for four different stages of the dust event on 15th-16th March 2022.

We therefore believe that our initial paragraph was not clear. That paragraph has been re-written to emphasize this point and avoid redundancies (L431-L442)

"Figure 6 shows a general pattern in $F_{11}$ characterized by strong predominance of forward scattering up to two orders of magnitude greater than backward scattering. However, there are significant changes in both magnitudes and spectral dependence over time, that is, with the intensity of the dust outbreak passage. At the beginning of the dust event (Fig. 6a), the values of $F_{11}$ in the forward scattering region are around 1000 Mm$^{-1}$sr$^{-1}$ for all

three wavelengths, which is even one order of magnitude lower when compared with the cases at the other moments of the event (i.e. 50000 Mm$^{-1}$sr$^{-1}$ for the three channels during the peak). Also, at the beginning of the event (Fig. 6a) notable spectral separation in $F_{11}$ is observed while such spectral separation is negligible during the rest of the event when coarse mode particles largely predominate. All $F_{11}$ show the minimum in the region 120º-140º but the magnitude of that minimum varies between the different stages. Also, around that minimum is the region where some spectral difference is observed during the cases of strong predominance of coarse mode (Fig. 6c-d). A recovery from that minimum is also observed, being more pronounced in cases close to the peak of the event."

• **Figs 6,7,9 lower row: the y-scaling is quite coarse, please add some ticks in between.**

-Done.

**You may add the value of the PM10 concentration for the different time steps because you are discussing it a lot and it is not always visible from Fig. 5+8.**

-We thank referee suggestion. This information can be found in Tables 1 and 2 and adding them could be repetitive.

**The term "moment" seems not the best choice, maybe better "instances" or "time steps" or something else. Please consider changing it here and in the text.**

-Done.

• **L381-383 Sentence unclear.**

-We have modified this sentence (L481-484).

"Figure 7 shows that $F_{11}$ patterns are very similar to the previous extreme event on 15$^{th}$ - 16$^{th}$ March, with strong predominance of forward scattering (~25000 Mm$^{-1}$sr$^{-1}$), being two orders of magnitude above the backscattering (~100 Mm$^{-1}$sr$^{-1}$) at the peak of the event on 25$^{th}$ March 9:00 UTC. There are no significant spectral differences, as also happened for the other extreme event on 15$^{th}$-16$^{th}$ March."

• **L387-389 Be more precise. The whole paragraph on page 11 needs some rephrasing to be more precise.**

-That paragraph has been re-written, and now is given by (L481-L491):

"Figure 7 shows that $F_{11}$ patterns are very similar to the previous extreme event on 15$^{th}$ - 16$^{th}$ March, with strong predominance of forward scattering (~25000 Mm$^{-1}$sr$^{-1}$), being two orders of magnitude above the backscattering (~100 Mm$^{-1}$sr$^{-1}$) at the peak of the event on 25$^{th}$ March 9:00 UTC. There are no significant spectral differences, as also happened for the other extreme event on 15$^{th}$-16$^{th}$ March. These patterns in $F_{11}$ agree with laboratory measurements of dust samples (i.e. Muñoz et al., 2007; Renard et al., 2014; Volten et al., 2001). Nevertheless, there are some features in $F_{11}$ with different situations: the slope in $F_{11}$ in the forward scattering region becomes sharper when the PM$_{10}$ concentrations are higher (Figs. 7b.1 and 7c.1). For the backward region $F_{11}$ shows a flatter behavior for high PM$_{10}$ concentrations (Figs. 7b.1 and 7.c.1), while for the cases with lower PM$_{10}$ concentrations there is a sharp increase in scattering from 150° to 180°. During the

previous extreme dust outbreak, we observed flat patterns for the backward scattering region during the peaks of the dust intrusions."

• **Tab 1+2 "The angular range of the PI-Neph is used as the integration range of sigma_sca." Please provide the angular range otherwise this information is not very helpful.**

-We have added this information to the caption of the tables.

**Why some values at 405 nm are missing on 25 March?**

-After re-calculating using the data quality criterion in Bazo et al., (2024) we figure out that some data were eliminated incorrectly. Consequently, we have modified the values in Table 1 and Figure 5.

• **L450: A newer and more comprehensive overview of lidar ratios was reported in Floutsi et al., AMT 2023.**

-We thank the reviewer for this useful reference. We have added the citation in the text and included Floutsi et al., (2023) in the list of references.

• **Fig 8. Sorry, but it is really challenging to get anything out of Fig. 8. Please use more space to show your results, half of a page minimum. Or remove the figure from the manuscript. I see only dots and can hardly infer any value from the y-axis. You probably want to show that there are more dust outbreaks during summer.**

- We agree with the referee. We have used more space to show Figure 8.

• **Tab 2 and Fig. 9: Do you show the measurements for the whole day? Or for one hour? Or for the periods marked in Fig. 8?**

-We show hourly-averaged measurements that correspond to the peak in scattering during the dust event. We have added this information to the text (L567-568):

"Particularly, Fig. 9 shows hourly averages of $F_{11}$ and $-F_{12}/F_{11}$ representative of the peak in scattering during each event."

• **Fig 10 and surrounding text: BC and BrC are not defined.**

-We have added the definitions of both acronyms in L625-626 following the definitions in Schmeisser et al. (2017).

"….where BC refers to black carbon and BrC to brown carbon in the definitions given by Schmeisser et al., (2017).

• **L539-541: Polarization measurements are very valuable for separating dust and non-dust contributions. This potential is used in the active remote sensing community for two decades now, starting with Shimizu et al., 2004, continuing to Tesche et al., 2009 and Mamouri and Ansmann 2017. You now adding PI-Nephelometer measurements to this separation.**

-We agree with the reviewer on this and we have accordingly modified the sentence (L646-647):

" …. show the potential of ground-based phase matrix measurements to distinguish between different types of aerosol mixtures."

- **Fig 11 Do you expect that the seasonal average of -F12/F11 should go back to zero for 180° or could it stay negative?**

-In Figure 11 we show $F_{11}$ and $-F_{12}/F_{11}$ only in the range 5º-175º that are the ranges where the polar nephelometers operate. The value at 180º can be computed by linear extrapolation of the closest points (Horvath et al., (2015) or by other more complex methods such as the proposed in Gomez-Martin et al., (2021). However, the objective of this work is not to provide at the exact angle of 180º accurate values of $F_{11}$ and $-F_{12}/F_{11}$, even though it is critical for the lidar community. What Figure 11 reveals is the larger standard deviations in the seasonal averages for scattering angles larger than 170º approximately when compared with the range 150-160º. Nevertheless, the standard deviations for scattering angles larger than 170º are lower than those standard deviations observed in the region of minimum $-F_{12}/F_{12}$ values. The standard deviations are explained because of the variability of the aerosol sampled during the entire season.

To clarify the issues related to the standard deviations in $F_{11}$ and $-F_{12}/F_{11}$ patterns we have re-phrased the paragraph and now is given by (L683-692):

"Figure 11 shows that seasonal values of $-F_{12}/F_{11}$ present in all cases larger standard deviations when compared to $F_{11}$. Particularly, for 660 and 515 nm large standard deviations are found in the region between 50º-150º while for 405 nm the standard deviations are considerably lower. This suggests that these $-F_{12}/F_{11}$ values at 660 and 515 nm are very sensitive to changing conditions in the aerosol that is sampled. Moreover, the other region that presents remarkable standard deviations for all wavelengths is the region of scattering angles above 170º. That region is very sensitive to any change in particle type and size, what was demonstrated both from theoretical computations (Mischenko et al., 2002) and in laboratory measurements (Gomez-Martin et al., 2021). However, the lower standard deviations observed for the 405 nm wavelength indicate homogeneity in the response to polarization, even in the presence of other anthropogenic particles in the sample."

**Please indicate the wavelengths for the Granada Amsterdam Light Scattering data base in the caption or even in the figure. By the way, 488 nm are much closer to 515 nm than to 405 nm. Why did you choose to show it together with the 405 nm?**

-We have included the wavelengths for the Granada Amsterdam Light Scattering database in the caption and in the figure. We had compared the 488 nm wavelength with the 405nm to reinforce the fact that during the extreme dust events the 405nm in the PI-Neph shows similar results as other instruments in the laboratory. However, the reviewer is right about the wavelength difference. Therefore, we have also shown a comparison of the 488nm wavelength along with our measurements for 515 nm.

**Yellow is probably not the best choice – could you choose a different color? And overall, the final publication should get the figure in a higher resolution. It is hard for me to distinguish the open circles from the stars.**

-Thanks for the suggestion. We have changed the color of this dataset from yellow to light blue. Also, we have tried increasing the size of the markers in the figure. However, since the scattering matrix elements from the two extreme events are very similar, there is superposition of datasets, and we decided to keep the original sizes.

**Caption: "top" instead of "up"**

-Done.

• **Do you have a pure pollution case for comparison? It would certainly enhance the message to have a contrasting pollution case presented or repeated from Bazo et al., 2024 in this paper as well.**

-Unfortunately, for pure pollution we only have the case from Bazo et al., (2024).

• **Paragraph (L598-612): You already mentioned Teri et al., 2024. Here some more discussion to the polluted dust cases in the Eastern Mediterranean would be helpful. Overall, I would recommend placing your work in a broader context besides of previous measurements at Granada station.**

-The referee is right and a comparison with other places would have enriched the manuscript. But our paper focusses only on $F_{11}$ and $-F_{12}/F_{11}$ from ambient air, and unfortunately, we are not aware of the availability of other studies in the Eastern Mediterranean that deal with this type of measurements. Referee 1 pointed out that conclusions cannot be extrapolated to all kinds of mineral dust particles, with is very interesting point as well. Therefore, we have modified the conclusion section remarking that more measurements at other experimental sites are needed (L785-L789).

"…The novel measurements of $F_{11}$ and $-F_{12}/F_{11}$ can also complement other optical and microphysical properties of Saharan dust already known from in-situ instrumentation and by active and passive remote sensing instruments, both from the ground and the space. Nevertheless, more $F_{11}$ and $-F_{12}/F_{11}$ measurements are needed at other experimental sites to have a more complete vision of mineral dust role on climate."

• **L654-657: Sentence unclear. Please rephrase to be more precise.**

-Thanks for the feedback. In the revised manuscript it is now written (L766-L768):

"Future optimization in GRASP will permit the retrieval of aerosol refractive indexes between fine and coarse mode separately using as inputs $F_{11}$ and $-F_{12}/F_{11}$, and thus permitting further analyses of the different study cases discussed in this work"

• **Fig. 12 The negative values for the $-F_{12}/F_{11}$ close to 180° for the pure dust case are not seen in the observations of the extreme dust events. Is it a modelling artefact or is it missing in the observations? What is your explanation? Please discuss in your paper.**

-We are not sure we understand this point. Figure 12, which are GRASP simulations, shows negative values of $-F_{12}/F_{11}$ very close to zero in the backward scattering region. If the reviewer refers to the observations from Figure 11, we include below a zoom close to 180º region (Figure R7). We can observe values very close to zero in $-F_{12}/F_{11}$ for scattering angles close to 180º - any difference falls within the uncertainties. Note that theoretically

-$F_{12}/F_{11}$ is zero only under the assumption of mirror symmetry and randomly oriented particles in the sample, that is not always fulfilled in the nature (Mischenko et al., 2002). We also observe a small negative branch of -$F_{12}/F_{11}$ near 180°, as the simulations from Figure 12 showed.

[Figure]

Figure R7. Zoom of the degree of linear polarizations (-$F_{12}/F_{11}$) for the extreme events in March 2022.

We have included the following statement (L755-757):

"…with small negative values around 180°. Note that this feature is also present in the -$F_{12}/F_{11}$ in Figure 11 for the extreme dust cases, but it is not noticeable due to the scale."

• **L668: "To our knowledge, these are the first measurements of this type for ambient mineral dust transported to southern Europe" – only in Southern Europe. Where else ambient mineral dust measurements have been performed? What is the difference to your observations?**

-We are referring to $F_{11}$ and -$F_{12}/F_{11}$ measurements of ambient mineral dust. Previous study of Horvath et al., (2018) measured $F_{11}$ with a single-wavelength polar nephelometer of ambient transported Saharan dust aerosol in the Sierra Nevada National Park, which is around 20 km in horizontal distance from the UGR station where the measurements in this work took place. However, this study did not provide multiwavelength -$F_{12}/F_{11}$ measurements.

On the other hand, previous designs of PI-Neph by the University of Maryland, Baltimore County (UMBC) provide the $F_{11}$ and -$F_{12}/F_{11}$ measurements of ambient aerosol. In the DC3 field campaign the first design of PI-Neph operated only at 532 nm, while in SEAC4RS it operated with 473, 532 and 671 (Espinosa et al., 2017, 2018, 2019). Another PI-Neph version (also developed in UMBC) can acquire $F_{11}$ and -$F_{12}/F_{11}$ measurements at 660 and 405 and is operated by NOAA (Ahern et al., 2022). However, these instruments have not acquired multiwavelength measurements of Saharan dust yet. There are other developments in laser imaging nephelometry (i.e Moallemi et al., 2023) capable of measuring $F_{11}$ and -$F_{12}/F_{11}$ but up to now these instruments only operate for laboratory conditions. All these issues are now included in the new instruction section (L129-L146):

[revised manuscript text omitted]

Shimizu, A.; Sugimoto, N.; Matsui, I.; Arao, K.; Uno, I.; Murayama, T.; Kagawa, N.; Aoki, K.; Uchiyama, A. & Yamazaki, A.: Continuous observations of Asian dust and other aerosols by polarization lidars in China and Japan during ACE-Asia, Journal of Geophysical Research: Atmospheres, 2004, 109, D19S17.

Tesche, M.; Ansmann, A.; Müller, D.; Althausen, D.; Engelmann, R.; Freudenthaler, V. & Gross, S.: Vertically resolved separation of dust and smoke over Cape Verde using multiwavelength Raman and polarization lidars during Saharan Mineral Dust Experiment 2008, Journal of Geophysical Research: Atmospheres, 2009, 114, D13202.

---

## Author Comment (AC4)

The authors greatly acknowledge the anonymous reviewer for carefully reading the manuscript and providing constructive comments. This document contains the author's responses. Each comment is discussed separately with the following typesetting:

**Reviewer's comments**

Authors response

Changes in the manuscript

Reviewer #1

**The manuscript provides a summary of ambient measurements of F11 (phase function) and F12 made in Granada, Spain by the Airphoton/GRASP-Earth PIN100 as well as measurements of PM10 and a suite of other optical quantities. Two cases with very high concentrations of advected Saharan dust are explored in detailed followed by a summary of several cases with more moderate dust loading. Results are contextualized by single scattering calculations for spheres and spheroids showing theoretical F11 and F12 for hypothetical urban, mixed, and pure dust aerosols.**

**The complex shape of desert dust particles limits our ability to accurately predict the optical properties required in aerosol modeling and remote sensing but direct, ambient measurements of these quantities, especially phase function and other scattering matrix elements has been very limited. This paper presents some of the only ambient dust scattering matrix measurements to date and the results have great potential to help improve on these past limitations. The identification of the strong dependence in F12 to the fine mode, particular at shorter visible wavelengths is also a valuable contribution. I do however feel the manuscript fails to properly contextualize results, particularly in its accounting for potential measurement errors. At times it also seems to jump to very strong conclusions based on what seems to be limited or inconclusive evidence. To be suitable for publication in ACP, I feel that these issues, outlined in detail below, must be addressed.**

We thank the reviewer for his/her time and efforts to review our work, and we believe that the feedback received is very positive to improve the manuscript.

The question about the uncertainties of the Airphoton/GRASP-Earth PIN100 uncertainties were also raised by the other referees. We did not give many details about the uncertainties because in Bazo et al., (2024) we performed an exhaustive characterization of the PI- Neph and of its uncertainties. Here, we want to give an overview that will help in this public discussion.

**PUBLIC DISCUSSION OF Airphoton/GRASP-Earth PIN100 UNCERTAINTIES FROM BAZO ET AL., (2024) MANUSCRIPT**

Appropriate calibration of the instrument

The instrument follows an exhaustive calibration procedure that consists of two different steps: The first is a geometric correction that corrects from the different light paths to the different pixels in the CMOS camera used as detector. To do so we used gases of known phase functions that follow Rayleigh theory (e.g. $CO_2$ or particle free air), particularly the perpendicular signal that follows the shape of $\cos^2\theta$ (see Eq. 13 in Bazo et al., for details). Measurements are fit to the theoretical shape by using an $8^{th}$ order polynomial. The following Figure (Fig. 5 in Bazo et al., (2024)) illustrate how the application of the computed geometric correction serves to accurately reproduce theoretical phase functions of $CO_2$ and clean air

[Figure]

Figure 5 in Bazo et al., (2024). (a) Geometric correction procedure: raw perpendicular signal for 515 nm (green dots); red asterisks represent the edge values used to calculate the correction function (gray circles); the correction function is fitted to an 8th degree polynomial (dashed line), that corresponds to the geometric correction. (b) Application of the geometric correction to parallel (b.1) and perpendicular (b.2) particle free air signals.

The second step is to perform an absolute calibration that permits to obtain phase matrix in physical units. We did it by measuring the parallel and perpendicular signals of pure $CO_2$ and particle free air, since their phase functions are well-known. The sum of both components corresponds to the phase function in digital counts $C_{11}(\theta) = C_{output,||}(\theta) + C_{output,\perp}(\theta)$. Therefore, it is possible to obtain a linear relationship between the counts recorded by the CMOS and the scattered light pattern: $F_{11}(\theta) = K \cdot C_{11}(\theta) + W$ where $K$ and $W$ are the coefficients of a linear fitting between theoretical and experimental data. These coefficients of the linear relationship between the phase functions in physical units $F_{11}(\theta)$ and in digital counts can be computed following the same procedure used in common integrating nephelometers by fitting the theoretical scattering coefficients of pure $CO_2$ and particle free air to the integral of the phase function in digital counts ($\tilde{\sigma}_{counts}$):

$$\tilde{\sigma}_{counts} = \frac{1}{2} \cdot \int_{0°}^{180°} C_{11}(\theta) \cdot \sin\theta \cdot d\theta$$

where the values of $C_{11}(\theta)$ between 0 and 5° and 175 and 180° are extrapolated by the closest neighbor method, since the theoretical Rayleigh $F_{11}(\theta)$ follows the $(1 + \cos^2\theta)$ shape. For the computation of the theoretical scattering coefficients the methodology

proposed in Bodhaine et al. (1991) was used, applying corrections of pressure and temperature from PI-Neph sensor measurements. We can then compute $\sigma_{\text{theoretical}} = K \cdot \tilde{\sigma}_{\text{counts}} + W$. Note that the geometric correction is first applied to the measurements. The evaluation of the calibration constants with time was also done and minimum differences (below 5%) were obtained.

Our evaluation of the different ways of calibrating the PI-Neph instrument was even further by performing an angle-to-angle calibration that consists of associating each angle with a different calibration constant. To do so, we use again well-known scatters that follow the Rayleigh scattering matrix (again $CO_2$ and particle free air). Then, an angle-to-angle comparison of measurements with the theoretical scattering matrix elements is done, obtaining a calibration constant for each data point in the measurements. That approach can serve to mitigate measurements affected by stray light. Figure below (Fig. S1 in Bazo et al., (2024)) shows the results in $F_{11}$ and $-F_{12}/F_{11}$ using the above calibration method and the angle-to-angle calibration. Differences in $F_{11}$ and $-F_{12}/F_{11}$ are negligible.

[Figure]

Figure S1 in Bazo et al., (2024): $F_{11}$ (up) and $-F_{12}/F_{11}$ (bottom) for 405 (left), 515 (middle) and 660 nm (right) calculated with total gas calibration (dashed line) and angle to angle gas calibration (colored dots).

Evaluation of Instrument Stability

Figure below - Figure S2 in Bazo et al., (2024) - shows the evolution of the total scattering coefficient during 15 min both for particle free air and $CO_2$ measured at a constant flow rate of 10 Lmin$^{-1}$ versus theoretical values (corrected of temperature and pressure). In general, no significant changes in the scattering coefficients are observed with time, which reveals very good stability of the instrument. Mean differences between the theoretical and calculated particle free air scattering coefficient are 0.0608 Mm$^{-1}$ for the 660 nm channel (1.07% difference), 0.0341 Mm$^{-1}$ for 515 nm (0.25% difference), and 0.3152 Mm$^{-1}$ for 405 nm (0.73% difference). Similar values are found for the $CO_2$ scattering coefficients, with mean differences of 0.0567 Mm$^{-1}$ for 660 nm (0.38% difference), 0.0952 Mm$^{-1}$ for 515 nm (0.17% difference), and 0.4258 Mm$^{-1}$ for 405 nm (0.40% difference)

[Figure]

Figure S2 in Bazo et al., (2024): Changes in the scattering coefficients of particle free air (left) and $CO_2$ (right) with time, over a 15 min measurement period.

Errors Associated with the Imaging Technique

The first one is the different background pixel counts due to changes in the temperature of the CMOS detector, but this error is minimized because background images are taken at the beginning of each sequence when differences in temperature are negligible. The other source of error in the imaging technique comes from the selection of the pixels associated with the scattering of the laser beam. The PI-Neph software is capable of appropriately detecting saturated pixels and those with low SNR.

Natural fluctuations of the sample can lead to noisy measurements being particularly important in conditions of high aerosol variability and enhanced with very short exposure times. To characterize these effects, we carried out measurements with very low aerosol loads and different exposure times, ranging from 1s to 20s. Figure below - Figure S4 in Bazo et al., (2024) - shows instantaneous $F_{11}$ and $-F_{12}/F_{11}$ (in arbitrary units) for 515 nm for the different exposure times, revealing an improvement in both parameters as the exposure time increases. Selecting the appropriate exposure time for every measurement makes the continuous operation of the instrument difficult, and although exposure times must be revised by the user it is fixed during long periods. That is why noisy measurements plus saturated and low SNR measurements are present and must be adequately treated.

[Figure]

Figure S4 in Bazo et al., (2024): For 515 nm, P11 and -P12/P11 with different exposure times.

Instrument Validation versus Monodisperse Aerosol

For the validation of PI-Neph data, monodisperse aerosol measurements were carried out. In particular, measurements of PSL of 0.75 μm and 1 μm diameter have been performed in the laboratory. For the generation of the aerosol sample, an Aerosol Generator (TSI 3076) was used, followed by a Diffusion Dryer (TSI 3062) that was connected to the PI-Neph's inlet. The set of measurements was done for 30 minutes, with a total of 30 matrix elements measurements. Exposure times were adjusted depending on the wavelength and polarization state. Results presented here correspond to temporal averages of the entire period. Figure below (Figure 10 in Bazo et al., (2024)) shows experimental data and fits with the GRASP algorithm. Generally, the agreements in $F_{11}$ are excellent, reproducing all resonances, while for $-F_{12}/F_{11}$ there are some deviations. If we assume that errors in the instruments are of 5-10% and 10-20% for $F_{11}$ and $-F_{12}/F_{11}$, respectively, and associated with issues associated with calibration and data acquisition of the CMOS, plus other errors associated with the imperfections and preparations of the PSL sample, then the differences observed in the Figure can be explained by errors propagation. Nevertheless, all resonances are well reproduced confirming that the pixel to angle relation in the PI-Neph is accurate and therefore serves as validation of PI-Neph measurements. The agreement between measured and retrieved value is supported by the high correlation coefficients and low residuals. GRASP retrievals were able to reproduce particle diameters and refractive index. Similar results were obtained when using a Mie code instead of GRASP algorithm.

[Figure]

Figure 10 in Bazo et al., (2024). PSL PI-Neph measurements of $F_{11}$ (left) and -$F_{12}/F_{11}$ (right) and GRASP fits for 0.75 μm (top) and 1 μm (bottom).

Data Quality Check

For analyzing a large dataset where the atmospheric conditions are variable, we developed a data quality procedure illustrated in the Figure below – Fig. 6 in Bazo et al., (2024). The idea behind this data quality check is to minimize errors associated with the measurement process. The complete procedure is applied to each channel and polarization individually. The first step is to select reliable data from parallel and perpendicular signals using the quality check parameter that serves to mark the data point as non-reliable if the averaged counts reach the saturation threshold. Since the number of counts at a specific angle is averaged through a group of pixels, the value of the quality check parameter of that angular measurement will depend on the number of pixels within the group that reached saturation. Specifically, it can vary from 0 to 1, where 1 means that all CMOS counts per pixel that have been average do not reach the saturation threshold. On the other hand, a value of 0 implies that all pixels in the region were saturated. Any value between 0 and 1 gives information about the ratio of CMOS counts per pixel that did or did not reach the saturation threshold. More details about the quality check parameter can be found in Section 2.2 of Bazo et al., (2024).

[Figure]

Figure 6 in Bazo et al., (2024). Flow-chart for the filtering procedure for the PI-Neph data.

In the data quality check procedure data with quality check parameter below 0.8 are discarded. Later, the procedure evaluates any possible peaks in individual measurements caused by a particle or by stray light. The filtering is based on the fit to a Legendre polynomial of the natural logarithm of the phase functions, which can be expressed as an expansion of Legendre polynomials. This method serves to detect outliers and assumes that if individual (angle dependent) values of the natural logarithm of the phase functions differ from the values of the fit to Legendre polynomial by more than a 10%, these points are discarded, since it is most likely that they come from a large particle that crossed the sample chamber or from stray light. The 10% threshold was selected after appropriate optimizations (see Bazo et al., 2024 for details).

After applying the filtering process, temporal averages are carried out. Temporal resolution of phase functions typically depends on the concentration of aerosols being sampled. Low aerosol concentrations lead to noisier averaged phase functions, hence larger averaging times are needed to have smoother patterns of the matrix elements, especially of $-F_{12}/F_{11}$. After performing averages, if more than 75% data points are qualified as non-reliable, all measurements are discarded to avoid any biases caused by the filtering process. If after performing the time averages there are more than 4 values in consecutive angles that are considered unreliable by the filtering procedure, the entire phase function is discarded as well. For cases with fewer values that do not comply with the filtering, a linear interpolation is performed (see Bazo et al., 2024 for details).

The Figures below – Figure 7 and 8 in Bazo et al., (2024) - show examples of the final $F_{11}$ and $-F_{12}/F_{11}$, respectively, after applying the data-quality check algorithm. Data corresponding to rows a, b, c to instantaneous, 30-minute average and 60-minute average, respectively; whereas columns 1, 2, 3 correspond to non-filtered, quality filtered, and Legendre filtered data (previously quality filtered), respectively. Data was acquired for ambient aerosol samples. In general, quality filter mostly affects the 405 nm channel in the angular range of 70 to 100°, since it is the most sensitive wavelength to low SNR data. Legendre filter removes data in the three channels for instantaneous data. Note that peaks in the three phase functions at 120° get filtered. However, there is lack of data in many regions in the angular range, so temporal averages (rows b, c) must be done to have a continuous phase function. It can be observed that temporal averages reduce noise in the phase functions but there are some outliers that remain. Therefore, applying a Legendre filter to the instantaneous phase function components (parallel and perpendicular signals)

and then performing temporal averages is the best option to have reliable data, especially for the $-F_{12}/-F_{11}$ element, which is affected by more noise because of the subtraction of the parallel and perpendicular signals measured by the PI-Neph.

[Figure]

Figure 7 in Bazo et al., (2024). Phase function $F_{11}$ for different stages of the filtering process. Rows (a), (b), (c) show instantaneous, 30 min average and 60 min average phase functions, respectively. Columns (1), (2), (3) show non-filtered, quality filtered, and Legendre filtered data, respectively.

The discussion presented serves to illustrate the different sources of errors that can affect PI-Neph measurements. We note that for laboratory measurements under controlled conditions uncertainties are in the range of 5-10% for $F_{11}$ and 10-20% for $F_{12}$. For ambient aerosol samples, the situation is more complex because of the natural variability of the aerosol. The large variability of particles that can be present in ambient aerosols make natural fluctuations in the phase matrix to appear. For example, a very coarse particle can be in the air sample and depending on the orientation, it can affect critically the shape of the phase functions. The data quality approach tries to avoid these fluctuations, although they are present in nature. We therefore believe that PI-Neph uncertainties are typically 5% for $F_{11}$ and 10% for $F_{12}$ but for real aerosol samples the fluctuations in aerosol particles plus the complexity of their mixture make the standard deviations higher than the claimed uncertainties in most cases.

[Figure]

Figure 8 in Bazo et al., 2024. Polarized phase function $-F_{12}/F_{11}$ for different stages of the filtering process. Rows (a), (b), (c) show instantaneous, 30 min average and 60 min average polarized phase functions, respectively. Columns (1), (2), (3) show non-filtered, quality filtered, and Legendre filtered data, respectively.

Validation versus TSI Integrating Nephelometer

The PI-Neph was measuring continuously in April-September 2022 and a with a collocated integrated nephelometer (TSI 3563). That allowed to evaluate the scattering coefficient defined as:

$$\sigma_{sca}(\lambda) = \frac{1}{2}\int_0^{180} F_{11}(\theta, \lambda) sin\,\theta \cdot d\theta\,, \qquad (1)$$

The integrating nephelometer measured in the range 7º-170º while the PI-Neph measured $F_{11}$ in the range 5º-175º. Therefore, to avoid any hypothesis about corrections we evaluated the truncated scattering coefficient in the range 7-170º that was measured directly by both instruments. The results of the general fits were very good with slopes around 0.93 and $R^2$ above 0.81. Differences could be associated with the uncertainties of each instrument. Computed RMSE were of 3.79, 4.57 and 8.51 for 660, 515 and 405 nm channels. More details are in Bazo et al., (2024)

\*\*\*\*\*\*\*\*\*\*\*\*\*\*\*\*\*\*\*\*\*\*\*\*\*\*\*\*\*\*\*\*\*\*\*\*\*\*\*\*\*\*\*\*\*\*\*\*\*\*\*\*\*\*\*\*\*\*\*\*\*\*\*\*\*\*\*\*\*\*\*\*\*\*\*\*\*\*\*\*\*

We understand referee concerns about the uncertainties of the instrument, and therefore we have included an overview of instrument uncertainties and data quality check procedure in Section 2.2.1 (L199-225):

[revised manuscript text omitted]

And we have also modified part of the introduction to contextualize better the study and objectives of our manuscript. The main modifications are in Lines 82-158:

[revised manuscript text omitted]

**GENERAL COMMENTS**

 **The exact method for obtaining several key optical quantities is not specified. Four aspects of the data were particularly unclear:**

**(1) The PI-NEPH does not measure F$_{11}$ at exact backscattering, but LR is provided throughout. How was F$_{11}$(180°) estimated given the available data?**

For the computation of the scattering coefficient, the objective is not to compute an exact value for F$_{11}$(180°). Indeed, we want to estimate F$_{11}$ for the entire range 175º - 180º. To do so we used the methodology based on linear extrapolation using the neighbors measurements. According to studies of Horvath et al., (2015), these linear extrapolations only imply 5% uncertainty in the computation of σ$_{scat}$, g and B$_s$. We have modified the text to clarify this point (L233-235).

"…where data from 0 to 5° and from 175 to 180° have been linearly extrapolated to obtain the complete phase function which according to Horvath (2015) only implies uncertainties up to 5% in the computations of the $\sigma_{scat}$, g and B$_s$."

Once computed the scattering coefficients, we use measurements of the absorption coefficient to obtain single scattering albedo – note that the absorption coefficient at the exact wavelengths of the PI-Neph is computed using absorption Angström exponent (see reply to comments below). However, the computation of extinction-to-backscattering ratio (LRs) requires exact values of $F_{11}(180°)$, and we use values of $F_{11}(180°)$ computed following the linear extrapolation method. More robust methods could have been used being the differences between up to 20-30 % (i.e. Gomez-Martin et al., 2021). These differences are close to the uncertainties of the PI-Neph and therefore LRs estimations will serve only as an illustration of how this parameter varies under different conditions. The text has been modified accordingly (L265-268):

"The computation of $F_{11}(180°)$ has been made using the interpolation method used for completing the entire angular range in $\sigma_{sca}$. Other more robust methods can be used (i.e. Gomez-Martin et al., 2021), that can imply differences in $F_{11}(180°)$ of up to 20-30%. Therefore, LRs estimations will serve as an illustration of how this parameter varies under different conditions"

We have also added the reference Horvath (2015) to the reference list:

"Horvath, H.: Extrapolation of a truncated aerosol volume scattering function to the far forward and back region, J. Aerosol Sci., https://doi.org/10.1016/j.jaerosci.2015.08.001, 2015."

**(2) The TSI integrating nephelometer is described and it is stated that "$\sigma_{sca}$ can be also obtained with the integrating nephelometer" but it's not clear if this data is ever used, or all $\sigma_{sca}$ values are obtained via the PI-Neph. If the latter is the case, then there is no need to introduce the TSI instrument.**

We have used the scattering coefficients from the TSI integrating nephelometer to obtain the Angstrom matrix in section 5.1. We could have done this with the scattering coefficients by the PI-Neph, but we wanted to maintain the same wavelength range for the calculation of the scattering Angstrom exponent to apply the classification proposed by Schmeisser et al. (2017). We have added the following statements to the manuscript (L623-625):

"For the SAE, we have used the $\sigma_{sca}$ measured with the TSI integrating nephelometer since it directly provides measurements at the same wavelengths than those required in the Schmeisser et al., (2017)."

**(3) Equations 5 and 6 show general definitions for SAE and AAE but I don't think the exact wavelengths used are ever stated. Please specify the wavelengths used in these calculations.**

-Thanks for this point. We included this information in the results section, whenever both variables were introduced. But to make this clear, we specified the wavelengths used for SAE and AAE when defining these parameters in Equations 5 and 6 (L257-L258).

"… The wavelengths used in this work to calculate both SAE and AAE have been 405-660 nm and 450-700 nm, respectively."

**(4) How were the particles sampled and did any aspect of the process have the potential to produced size dependent biases relative to the ambient aerosol. This is particularly import for the dust dominated scenes since a significant portion of dust particle volume concentration occurs in particles several microns in diameter that are easily lost to inlets and tubing.**

-We agree with the reviewer's comment on the importance of sampling losses during dust events. For this reason, the experimental set-up used in this study was designed to minimize any potential losses of large particles. Measurements were performed with a total inlet (no size-cut) consisting of 5 m long stainless-steel tube displayed vertically to minimize deposition losses, and a flow splitter that provided the appropriate flow to each instrument. The final connection to all instruments, including the PI-Neph, was performed with conductive tubing (TSI), avoiding bends. No aerosol dryer, that might lead to additional particle losses, was used due to the low ambient relative humidity at the measurement site. Further details of the inlet system can be found in Lyamani et al. (2008). We have added the following statement to the manuscript (L177-L181):

"In this study, particles were sampled using a total inlet (no size cut) that consist of a 5 m long stainless-steel tube with a 20 cm diameter (Lyamani et al., 2008). Inside the stainless-steel tube there are several pipes that split the aerosol flow into the different instruments. The inlet system is completely vertical to minimize deposition losses. The final connection to the instruments is performed with conductive tubing avoiding bends. Additionally, all the measurements were at ambient conditions (no aerosol dryer was used)."

And we have added the reference Lyamani et al. (2008) to the reference list between

"Lyamani, H., Olmo, F.J., Alados-Arboledas, L. Light scattering and absorption properties of aerosol particles in the urban environment of Granada, Spain. Atmos. Environ. 42, 2630–2642. https://doi.org/10.1016/j.atmosenv.2007.10.070, 2008"

**◇ At times measurements seem give nonphysical results. For example, -F$_{12}$/F$_{11}$ is unrealistically low in Figure 9b.2 and trends in AAE do not always match spectral trends in SSA (see specific comment – LN 467). Measurement errors are unavoidable, but I think these issues further emphasize the need to include well characterized uncertainties on the measurements, as the other reviewers have noted.**

We agree with the referee, and we have added a more detailed description of the instrument including uncertainties of the instrument (see above comment). About the specific case in Figure 9.b, we highlight the variability in -F$_{12}$/F$_{11}$ due to the variability of aerosol samples in this period. Particularly, standards deviations at that moment were around 0.2, above the uncertainty of the instrument. This is now clarified in the revised manuscript (Lines 568-570):

"The standard deviations were 20-30% for $F_{11}$ and around 0.2 in $-F_{12}/F_{11}$, which are larger than the uncertainties of the instruments for all cases and explained by the large variability of aerosol samples during the measurement process."

**<> In my view, some conclusions are not well supported, especially regarding relationships between optical features and the underlying microphysical properties of the aerosol (see specific comments for examples). The inclusion of additional measurements like those of size distribution or chemical composition, if available, could go a long way here towards proving or disproving some of these claims. Additionally, GRASP could be a very powerful tool for interrogating some of the relationships between optics and microphysics hypothesized, at least in the context of spheroidal particles. This seems to have been started in Section 3.2.4 but it was not clear to me why this analysis was not extended to actual inversions of the phase matrix (and possibly absorption) measurements.**

The referee is right in all the issues raised to complement our study, but there were many shortcomings that avoided adding those points to our analyses.

The AGORA observatory has an Aerosol Particle Sizer (APS TSI 3321) and an Aerosol Chemical Speciation Monitor (ACSM - Aerodyne). However, during the two extreme Saharan dust outbreaks in March 2022 these two instruments were not available to operate at the station due to maintenance issues, that got extended until summer 2022. That is why these important measurements were not available for this study.

Concerning the issue pointed out of using GRASP, we would like to mention that currently GRASP is not optimized to perform inversion of bimodal size distributions representative of mixture of mineral dust (coarse particles) and anthropogenic pollution (fine mode particles) using $F_{11}$ and $-F_{12}/F_{11}$ as inputs. The simulations carried out only illustrate that the patterns observed in section 3.2.4 can be obtained for a mixture of particles– now section 5.2 in the revised manuscript. We have highlighted these issues in the revised manuscript (Lines 763-7678):

"However, studying the relationships between measured $F_{11}$ and $-F_{12}/F_{11}$ with other aerosol optical and microphysical properties requires further analyses because $F_{11}$ and $-F_{12}/F_{11}$ ultimately depend on the size distribution, refractive indexes, and particle shapes. The problem is even more complex if we differentiate optical properties between fine and coarse mode. Future optimization in GRASP will permit the retrieval of aerosol refractive indexes between fine and coarse mode separately using as inputs $F_{11}$ and $-F_{12}/F_{11}$, and thus permitting further analyses of the different study cases discussed in this work."

Moreover, given the importance of the points raised by the referee, we have decided to modify the conclusion section to emphasize the need to acquire correlative measurements of PI-Neph with additional instrumentation that provide information of particles size distribution and of particles chemical compositions (Lines 860-864).

"However, going further in understanding the interaction of dust with these anthropogenic particles requires further analyses that provide the chemical composition and size distribution of the ensemble of particles and the final composition and shape of the particles after interacting. This is planned in future studies that will allow a more complete comprehensive analysis."

**<> I found the structure of the text to be a bit difficult to follow. Perhaps moving some sections towards to beginning may help provide a clearer "landscape" for the reader as they progress through the text. Moving section 3.2.3 higher up, or at least providing a clear and consistent definitions of the aerosol types observed/hypothesized earlier on would help a lot.**

Due to comments from Reviewer 2, we have modified the numbering of the results and have divided them into three different sections. We have divided section 3 'Results and Discussion' in three different sections:

Section 3 'Overview of extreme dust events during March 2022' that initially was sub-section 3.2.1.1. This section gives an overview of the meteorological conditions associated with these two extreme events, plus some satellite observations that give an overview of the intensity of the dust plume.

Section 4 'Results of aerosol phase matrix from different dust scenarios': Here we include now sub-section 4.1 'Extreme events' that was initially subsection 3.2.1.2, and sub-section 4.2 'Moderate dust events during spring/summer 2022' that was initially subsection 3.2.2

Section 5 'Discussions': Here we include the sub-section 5.1 'Comprehensive assessment of the different dust events' that initially was sub-section 3.2.3 and sub-section 5.2 'Phase matrix simulations for different aerosol mixture scenarios' that initially was sub-section 3.2.4

**SPECIFIC COMMENTS**

**Eq 1: I see the reason $2\pi$ and $4\pi$ were kept separate, but I wonder if reducing it to 1/2 would make this equation a little easier to read, especially given the improved consistency with Eq 4.**

Thank you for pointing this out. We have changed the expression of Eq. 1 so it is consistent with Eq. 4.

**LN 140: This sentence is confusing. It is not clear that the single beam novelty they are referring to is the PIN-100. Also, it should be "University of Maryland, Baltimore County".**

-The referee is right, it should be noted University of Maryland, Baltimore County. We have modified this throughout the text. Concerning the text about the PIN-100, we have modified this part of the introduction (L189-191):

"The instrument uses previous heritage in PI-Neph developments in the University of Maryland, Baltimore County (Dolgos and Martins, 2014), where the novelty in the PIN-100 is the use of one beam instead of a mirror system to fold the laser beam, as was in previous models."

**Eq 7: How is $F_{11}(180°)$ obtained? F11 has a lot of structure in the backscattering direction and linear extrapolation is probably not very accurate, especially for nonspherical particle like dust. The method for extrapolating should be some discussion of the potential resulting uncertainties should be provided.**

As explained discussed in the general comment above, $F_{11}(180°)$ is only used in the computations of lidar ratios, that is a variable of interest for the lidar community. We used the linear extrapolation method, and we highlight the possible differences versus other more robust methods. We have corrected the text highlighting that the values given in the text serve to illustrate the LRs variability (L265-L268):

"The computation of $F_{11}(180°)$ has been made using the interpolation method used for completing the entire angular range in $\sigma_{sca}$. Other more robust methods can be used (i.e. Gomez-Martin et al., 2021), that can imply differences in $F_{11}(180°)$ of up to 20-30%. Therefore LRs estimations will serve as an illustration of how this parameter varies under different conditions."

**LN 197: "pression" -> "pressure"**

-Done.

**Figure 2/3: If all four subplots needed the color bars should be made consistent for all subplots and some discussion should be added to the text explaining what features differentiating the four subplots should be provided in the text. As it is, there is no significant difference that stands out to me between the four model times shown.**

-We have modified the color bars of Figures 2 and 3 and now the same scale is shown for all cases. We plotted the evolution of each event with the CAMS model with the objective of illustrating the temporal evolution of the event and what regions are affected. The following discussions have been added to the text (L290-L292):

"The wind field at the surface is also represented in these Figures, and different times selected serve to understand how the dust was transported and affected different regions…"

And Lines 296-298:

The wind pattern reveals how dust enters through the southeast of the Iberian Peninsula reaching later the northwest latitudes reaching even southern France.

**Figure 4: It would be good to include the name of the specific satellite/sensor that obtained these images.**

-Done, now the caption reads (L346-L347):

"Figure 4. (a) Satellite image from 15th March 2022. (b) Satellite image from 25th March 2022. Images from https://wvs.earthdata.nasa.gov, obtained with MODIS (Moderate Resolution Imaging Spectroradiometer)."

**LN 273: I assume this is a regulatory limit? This should be made clearer in the text.**

-The reviewer is right, it is a regulatory limit. We have added the following information in the text to make it clearer (L358):

"…regulatory daily limit value of 50 $\mu gm^{-3}$ established by the Ambient Air Quality Directive (2008/50/CE European Directive)"

**LN 284: It is well established that nonspherical particles can have weaker scattering near scattering angles of 180° but that only contributes a very small amount to the**

**total Bs which integrates from 90° to 180°. If there is a specific reference stating Bs decreases with particle nonsphericity that should be included here, otherwise I would suggest this sentence be removed or rephrased.**

The text has been modified as follows (L367-L369):

"The low values of the Bs observed at the three wavelengths are associated with small values of $F_{11}$ at the backward scattering angles, which is common for non-spherical particles (Nousiainen & Kandler, 2015)."

And we have added the reference Nousiainen and Kandler (2015) to the reference list between

Nousiainen, T., & Kandler, K. (2015). Light scattering by atmospheric mineral dust particles. In *Light Scattering Reviews 9: Light Scattering and Radiative Transfer* (pp. 3–52). Springer Berlin Heidelberg. https://doi.org/10.1007/978-3-642-37985-7_1

**Figure 5: The time series for Event 1 should be extended back to at least 7:00 on March 15th to be consistent with F11 and F12 shown in Figure 6 and to give better context to the background conditions prior to the dust intrusion.**

-We agree with the reviewer on this. However, most of the data between 7:00 and 12:00 did not pass the quality check procedure explained above, due to the intensity of the episode. We tried to monitor the dust event by adjusting the exposure time of the images to prevent saturation of the detector, but it was still impossible to obtain quality data so we had to lower the gain of the camera. That is why we do have a continuous quality-checked data set from 12:00 until the end of the episode. In Figure R1, the reviewer can find the extended time series where the is clearly a lack of data in the PI-Neph products. For this reason, we did not include this period in the figure but included Table 1 which contains the properties of Figure 5 at 7:00.

[Figure]

Figure R1. Time series of the PM$_{10}$ (1), $\sigma_{sca}$ – scattering coefficient (2), SAE – scattering Angström exponent (3), g – asymmetry parameter (4), Bs – fraction of backscattered light (5), SSA – single scattering albedo (6) and LR – lidar ratio (7) for the extreme dust events of 15th March.

**Section 3.2.1.2: Because data at each angle in a PI-Neph is based on the sum of many individual pixels, saturation tends to happen gradually and result in a steadily increasing bias rather than a sharp, obvious upper limit in F11. The challenge of avoiding such saturation given an aerosol with a combination of a strong forward peak and very high concentration is not discussed in the text. Was each image checked to ensure that the counts of all pixels were below the upper limit of the detector's dynamic range? The text should include at least a brief explanation of the steps taken to ensure saturation did not bias the measurements.**

-This is a very interesting point. As stated in Bazo et al. (2024) the software of the PI-Neph PIN-100 has a quality check parameter that marks the point as non-reliable if saturation happens. Whenever this happens, we discard the affected point and if it happens for many angles, we directly discard the complete measurement. Therefore, since we have the quality check parameter to account for saturation, we have not checked the images every image. Moreover, we had to change the gain of the camera due to the extremely high scattering that happens in the extreme dust events. For these reasons, and to also clarify why there were some gaps of measurements early in the morning on 15$^{th}$ March, we add the following text to the manuscript (L408-L413):

"Phase matrix elements were exhaustively monitored with the use of the PI-Neph during both extreme events. Given the high aerosol concentrations, the usual configuration of the measurements could lead to saturation of many angles in the forward scattering.

Therefore, it was necessary to reduce the gain of the PI-Neph's camera, changing the dynamic range of the camera for obtaining non-saturated measurements at such high concentrations. After this was done, some instantaneous measurements might still present saturation at some angles that were filtered out by the data quality criterion of the instrument (Section 2.2.1)."

**LN 341: Again, it is not clear to me how the values of g and Bs on their own–which are generally accepted to have the strongest dependence to particle size–are able to clearly suggest the predominance of spherical vs nonspherical particles.**

-We agree with the reviewer, and we are sorry for the misunderstanding in the original manuscript. What we wanted to remark is that the obtained values of g and Bs are typical of that observed for fine mode particles predominance. After carefully reading the comments of the three referees, we agree that we must be very cautious about making conclusions on particle size, shape and chemical composition from our phase matrix measurements. They have encouraged us to re-read the manuscript and make corrections to avoid any ambiguity. Specifically, for the line mentioned here we have simplified and avoided any mention to spherical vs non-spherical particles.

**LN 343: It's worth noting that holding intensive properties constant but increasing the aerosol concentration will increase F11 at all angles. Given that the authors seem to be using the PI-Neph to interrogate changes in microphysics here, I wonder if P11 would be a better variable to directly analyze? Total scattering and spectral dependencies between the three wavelengths can easily be captured in σscat.**

We agree with the reviewer, and we have re-written these sentences including a better discussion of $F_{11}$. Now, between lines 431 – 442 is given by:

"Figure 6 shows a general pattern in $F_{11}$ characterized by strong predominance of forward scattering up to two orders of magnitude greater than backward scattering. However, there are significant changes in both magnitudes and spectral dependence over time, that is, with the intensity of the dust outbreak passage. At the beginning of the dust event (Fig. 6a), the values of $F_{11}$ in the forward scattering region are around 1000 $Mm^{-1}sr^{-1}$ for all three wavelengths, which is even one order of magnitude lower when compared with the cases at the other moments of the event (i.e. 50000 $Mm^{-1}sr^{-1}$ for the three channels during the peak). Also, at the beginning of the event (Fig. 6a) notable spectral separation in $F_{11}$ is observed while such spectral separation is negligible during the rest of the event when coarse mode particles largely predominate. All $F_{11}$ show the minimum in the region 120º-140º but the magnitude of that minimum varies between the different stages. Also, around that minimum is the region where some spectral difference is observed during the cases of strong predominance of coarse mode (Fig. 6c-d). A recovery from that minimum is also observed, being more pronounced in cases close to the peak of the event."

The hourly temporal evolutions of $F_{11}$ and $-F_{12}/F_{11}$ are given in the supplementary material for more details.

**LN 348: "constant tendency" needs to be made more precise. It looks to me like F11(~180°) is an order of magnitude higher in panel c.1 relative to a.1 in Figure 6.**

-The reviewer is right about the values of $F_{11}$ from panels a.1 and c.1. By constant tendency we only refer that in the backward region the different $F_{11}$ show similar pattern in the backward scattering. In the revised manuscript we avoid these misunderstandings (see previous comment with the revised paragraph).

**LN 349: While the spectral dependence is less, the red channel, and maybe the green too, appears lower than blue to me in Figure 6a.1. The language here should also be made more precise.**

-The reviewer is right. We have avoided ambiguities in the revised manuscript and re-write the whole paragraph – see previous comment.

**LN 381: Should this say, "increase in F11" rather than "decrease"?**

-Following the suggestion of all referees, we have avoided ambiguities in our interpretation of the results. The whole paragraph where we discussed $F_{11}$ on 15th-16th March was rewritten, and now is given as (L481-491):

"Figure 7 shows that $F_{11}$ patterns are very similar to the previous extreme event on 15th - 16th March, with strong predominance of forward scattering (~25000 $Mm^{-1}sr^{-1}$), being two orders of magnitude above the backscattering (~100 $Mm^{-1}sr^{-1}$) at the peak of the event on 25th March 9:00 UTC.  There are no significant spectral differences, as also happened for the other extreme event on 15th-16th March. These patterns in $F_{11}$ agree with laboratory measurements of dust samples (i.e. Muñoz et al., 2007; Renard et al., 2014; Volten et al., 2001). Nevertheless, there are some features in $F_{11}$ with different situations:  the slope in $F_{11}$ in the forward scattering region becomes sharper when the $PM_{10}$ concentrations are higher (Figs. 7b.1 and 7c.1). For the backward region $F_{11}$ shows a flatter behavior for high $PM_{10}$ concentrations (Figs. 7b.1 and 7.c.1), while for the cases with lower $PM_{10}$ concentrations there is a sharp increase in scattering from 150° to 180°. During the previous extreme dust outbreak, we observed flat patterns for the backward scattering region during the peaks of the dust intrusions."

**LN 383: Many things can increase the forward peak of $F_{11}$ and it is known to be particularly sensitive to aerosol size. However, it seems to be implied here that change in shape is the most probable cause. This should be explained.**

-The reviewer is right. As we have commented in the previous referee´s question, we have re-written the entire paragraph to avoid ambiguities. In the conclusion section we mention that the patterns in $F_{11}$ and $-F_{12}/F_{11}$ depend on size distribution, particle shapes and refractive indexes. More measurements are needed to draw further conclusions. Also, optimization of GRASP inversion will help in the future.

**Figure 7: I think the caption should say "four different moments".**

-Thank you for identifying this typo, we have corrected it.

**LN 425: I found this a bit hard to follow. Is the statement that some studies suggest that SAE is a useful discriminator between the pure mineral dust and dusty mixtures with urban background pollution?**

-We use SAE as a proxy that helps to identify possible presence of mineral dust, which agrees with what's stated in the bibliography (i.e. Lyamani et al., 2008; Teri et al., 2024).

Measurements of PM10 plus the Angstrom matrix classification proposed by Cazorla et al. (2013) and Schmeisser et al. (2017) are later used to confirm the presence of mineral dust.

To clarify that SAE is used as a proxy based on bibliography, we have modified the manuscript that now reads as (L524-L527):

"For the identification of cases with influence of mineral dust particles, data shown in Fig. 8 are filtered out and correspond only to values of SAE< 1, which are used as a proxy to identify possible presence of dust particles in the atmosphere (i.e. Lyamani et al., 2010; Teri et al., 2024).

**Table 1: Aside from the ± variability windows, is there any information in this table that is not already provided in Figure 5? If not, I wonder if those could be represented by bars in Figure 5 and the table could be moved to the supplement.**

-We thank the referee for this suggestion. We consider that adding error bars to Figure 5 would make the graph more difficult to read, since there are several subplots that show many variables. Also, the objective of Figure 5 is to show a detailed temporal evolution. Table 1 is devoted to showing the mean values of the different variables at the exact times of PI-Neph data in Figures 6 and 7. We therefore believe that the current configuration of the Figures and Tables is appropriate.

**LN 443: "absorbing particles" should be made more precise. Is this referring specifically to black carbon?**

-We agree with the referee. We have changed 'absorbing particles' to 'black carbon particles'

**LN 463: AAE=1.71 is significantly higher than pure black carbon so it may be too strong to say that "absorption is mostly explained by black carbon".**

That value is for the case on 16th April that is the case with more complex aerosol mixture. Black carbon is a possible contributor in the mixture. We have modified the text to remark that flat dependences of SSA might suggest the presence of black carbon, but the high AAE suggests the presence of other particles such as mineral dust (L575-L581):

"The first event on 16th April is the one of the most complex in terms of mixture of particles, with the lowest $PM_{10}$ but with the largest SAE and AAE. The values of g suggest the lower contribution of dust particles when compared to the other cases (Horvath et al., 2018). The flat spectral pattern in SSA is the typical observed when there is contribution of fine mode particles of anthropogenic origin during a Saharan dust outbreak (Valenzuela et al., 2014). Nevertheless, the high AAE is not typical of black carbon and thus remarks the complexity of the mixture of aerosol particles in that day."

The reference Valenzuela et al., (2014) was added:

Valenzuela, A., Olmo, F. J., Lyamani, H., Granados-Muñoz, M. J., Antón, M., Guerrero-Rascado, J. L., Quirantes, A., Toledano, C., Perez-Ramírez, D., & Alados-Arboledas, L. (2014). Aerosol transport over the western mediterranean basin: Evidence of the contribution of fine particles to desert dust plumes over alborán island. *Journal of Geophysical Research*, *119*(22), 14,028-14,044. https://doi.org/10.1002/2014JD022044

**LN 465: Again, it is not clear to me that g can be used as a proxy for non-sphericity.**

-We have changed this sentence to (L576-577):

"The values of g suggest the lower contribution of dust particles when compared to the other cases (Horvath et al., 2018)"

**LN 467: It should be noted that while the SSA measurements on July 5 may show stronger spectral dependence, the AAE measurement is actually lower on July 5 than Apr 16.**

-The referee is right, we have added this information in L581-583.

"The more pronounced spectral SSA when compared to $16^{th}$ April also relates to the influence of dust in absorption (Dubovik et al., 2002), although the event on $5^{th}$ July shows a lower AAE than for the previous case."

**LN 469: It should probably be emphasized here that eBC is only equivalent BC for a given amount of absorption. Even at the longer wavelengths at which eBC is derived, very high levels of dust could produce significant absorption.**

-We agree with the reviewer, so we have added the following statement to the manuscript (L583-586):

"Finally, the case on $30^{th}$ August is the one with the largest $PM_{10}$ concentrations but also with the largest eBC, which can make a very complex mixture. However, it should be noted that when affected by high concentrations of dust, the dust particles might interfere with the eBC measurements."

**Figure 9: Blue -$F_{12}$/$F_{11}$ is much lower than any prior theoretical calculations (including those presented here in Figure 12) or measurements have shown. There are also some unnatural looking "kinks" in $F_{11}$ around the same angular region. This emphasizes need for good uncertainty estimates and careful data screening.**

-We agree with the referee. Uncertainties of the instrument are ~ 10% for F11 and ~20% for -F12/F11 but in laboratory conditions. For ambient samples, such as those presented in this work, natural variability of the samples during the 1 hour of data acquisition to fulfill the data quality requirements can lead to standard deviations larger than the uncertainties of the instrument. This is explicitly mentioned in the revised manuscript:

(L415-L419): "Computed standard deviations were larger than instrument uncertainties and they are associated with the variability of the different parcels of air sampled throughout the hour of measurements. These standard deviations were of ~ 20% for $F_{11}$ and ranging between 0.1 and 0.2 for -$F_{12}$/$F_{11}$ (minimums for the forward region and maximums in the middle region around 90º)."

L463-L465: "Again, the computed standard deviations are larger than the uncertainties of the instruments and they represent the variability of the samples. As for the previous extreme event, these standard deviations are ~20% for $F_{11}$ and between 0.1- 0.2 for -$F_{12}$/$F_{11}$. "

L568-L570: "The standard deviations were 20-30% for $F_{11}$ and around 0.2 in $-F_{12}/F_{11}$, which are larger than the uncertainties of the instruments for all cases and explained by the large variability of aerosol samples during the measurement process"

**LN 504: I'm not following this sentence. Why would high levels of noise mean that the measurement is "very sensitivity to aerosol mixtures"?**

-When comparing to Figure 7, phase functions in red and green channels for the stage of high dust concentrations (panel c) show a clear bell-shape typical of mineral dust. However, when the concentration of dust is not so high, this shape is not that clear, which is more like what we find in the phase functions of Figure 9. This is what we wanted to emphasize with this sentence, however the use of 'noisy' might not be very accurate. Therefore, we have re-phrased the sentence as (L605-L607):

"For 515 and 660 nm the $-F_{12}/F_{11}$ patterns are characterized by a bell-shape with large variability that might be associated with the complexity of the mixture of mineral dust and anthropogenic particles."

**LN 506: This also left me confused on first read. Is the idea here that the blue channel of $-F_{12}/F_{11}$ is depressed in anthropogenic (or absorbing?) aerosol and the feature shows up at low dust concentrations when the dust is not optically dominate?**

-Yes, the reviewer is right and that is the point we want to highlight. We have modified the text accordingly (L608-616):

"However, the $-F_{12}/F_{11}$ pattern in 405 nm shows a very different behavior, having $-F_{12}/F_{11}$ positive values until ~ 70° and negative $-F_{12}/F_{11}$ values for the following angles. Minimum $-F_{12}/F_{11}$ values are in the region around 120°. Therefore, $-F_{12}/F_{11}$ measurements can be potentially used for investigating the mixture of particles in the sample. That pattern with negative values has been observed in the UGR station for cases with no influence of Saharan dust particles (Bazo et al., 2024), and also for biomass-burning at 473 nm (Espinosa et al., 2017). Nevertheless, there are differences between the three different cases that might be associated with the differences in the mixtures of aerosol particles."

The reference Espinosa et al., (2017) was added:

Espinosa, W. R., Remer, L. A., Dubovik, O., Ziemba, L., Beyersdorf, A., Orozco, D., Schuster, G., Lapyonok, T., Fuertes, D., & Martins, J. V. (2017). Retrievals of aerosol optical and microphysical properties from Imaging Polar Nephelometer scattering measurements. *Atmospheric Measurement Techniques*, *10*(3), 811–824. https://doi.org/10.5194/amt-10-811-2017

**LN 517: I think this sentence is redundant with the first sentence of this paragraph.**

-We have removed the last part of the sentence.

**LN 540: I believe it is important to emphasize here that utility of 405 nm $P_{12}$ as a marker for smaller, likely anthropogenic particles may be unique to the region/circumstances studied here. For example, because of the size invariance rule for single scattering properties, an aerosol that is identical but with diameters only ~27% (1 - 515nm/405nm) smaller would produce scattering patterns in the blue that**

**are identical to what was measured in the 515 nm channel, which did not show this unique anthropogenic "fingerprint". This is not meant to diminish the finding, but instead to recommend that it be clearly caveated by the statement that this fingerprint may not be generalizable to all regions/seasons with dust intrusions.**

-The reviewer is right. We wanted only to highlight the potential of polarization measurements to distinguish different mixtures of mineral dust with other types of particles (L643-L646):

"However, for the cases classified as mixtures, there was a different pattern $-F_{12}/F_{11}$ for the 405 nm channel. Thus, the typing classification explains the differences in the phase matrix for the temporal evolution of the extreme dust events during $15^{th}$-$16^{th}$ and $24^{th}$-$25^{th}$ March 2022 in the UGR station…."

In the conclusion section we have highlighted that more measurements (i.e. chemical composition, size distribution) and optimized GRASP retrievals are needed to further understand the interaction of dust with anthropogenic particles (L860-L864):

"However, going further in understanding the interaction of dust with these anthropogenic particles requires further analyses that provide the chemical composition and size distribution of the ensemble of particles and the final composition and shape of the particles after interacting. This is planned in future studies that will allow a more complete comprehensive analysis."

**LN 603: Should this say, "blue wavelength"? I generally consider 400 nm to be the upper limit of the UV. Either way, the 473 nm measurements of Espinosa et al. (2017) were certainly not in the UV.**

-The referee is right, we have replaced UV with 405 nm channel.

**LN 655: I'm not sure what is meant by "optical properties" here. Either way, GRASP is certainly capable of simulating separate fine and coarse single scattering properties and refractive indices, as is evident in the forward calculation results show in Figure 12. I strongly suggest the authors include such inversions here.**

-As the reviewer says, GRASP can do the forward calculations, which were done for studying how different mixtures can produce different patterns in $F_{11}$ and $-F_{12}/F_{11}$. However, at the moment of writing the manuscript GRASP configuration using $F_{11}$ and $-F_{12}/F_{11}$ as inputs is not optimized for the retrieval that provides different optical properties between fine and coarse mode. Indeed, GRASP retrieves effective optical parameters (i.e. refractive index) that are the same for fine and coarse mode. It is true that GRASP retrieval can separate between fine and coarse mode for the configuration lidar + sun photometry sky radiances and based on that experience we are working on GRASP optimization for retrieving two modes with different optical properties using $F_{11}$ and $-F_{12}/F_{11}$. But this will be the scope of a future publication where we plan to analyze with GRASP the entire PI-Neph database at AGORA, with the use of additional in-situ instruments that provide measurements of absorption and size distribution. In that study we plan to add other cases such as biomass-burning that were measured in AGORA.

We have re-phrased this sentence to highlight that the forward simulation with GRASP helps us to understand that the negative values in $-F_{12}/F_{11}$ can be explained by a mixture

of dust with anthropogenic particles, but that going further needs future optimizations with GRASP. We would like also to note that future GRASP optimizations also need to implement super-coarse mode as suggested by Dr. Jean-Baptiste Renard.

L758-770: "Figure 12 results show how the presence of anthropogenic particles (fine mode) can alter the spectral dependencies in $-F_{12}/F_{11}$ when compared with only dust particles (coarse mode) in the sample, particularly in the blue channels. However, changes in the $F_{11}$ patterns were not so evident. These results help to understand the different phase matrix elements discussed in this manuscript, and their temporal evolutions during the extreme dust events (Supplementary Material). However, studying the relationships between measured $F_{11}$ and $-F_{12}/F_{11}$ with other aerosol optical and microphysical properties requires further analyses because $F_{11}$ and $-F_{12}/F_{11}$ ultimately depends on the size distribution, refractive indexes, and particle shapes. The problem is even more complex if we differentiate optical properties between fine and coarse mode. Future optimization in GRASP will permit the retrieval of aerosol refractive indexes between fine and coarse mode separately using as inputs $F_{11}$ and $-F_{12}/F_{11}$, and thus permitting further analyses of the different study cases discussed in this work. It is important to mention that the super-coarse mode can also affect the behavior of $F_{11}$ and $-F_{12}/F_{11}$ and the presence of this mode is also observed for long-range transport (i.e. Renard et al., 2010). Future GRASP developments also need the consideration of this super-coarse mode."

**LN 707: I still am not following how instrument noise can be linked with particle properties.**

We thank the referee for pointing this out. We wanted to remark that the large variability in $-F_{12}/F_{11}$ at 515 and 660 nm can be associated with the different mixtures of mineral dust and anthropogenic particles. We have re-phrased this sentence to make this point clearer (L826-830):

" …. while for the other channels at 515 and 660 nm they show the bell-shape pattern but with larger variability if compared to the extreme dust cases, probably associated with the complexity of the mixture of fine (anthropogenic) and coarse (dust) mode particles. These patterns in $F_{11}$ and $-F_{12}/F_{11}$ agreed with those observed in the intense Saharan dust outbreaks in March for the instants when the dust was entering and withdrawing"

---

## Author Comment (AC5)

Response to Reviewers

The authors greatly acknowledge the anonymous reviews for carefully reading the manuscript and providing constructive comments. Each comment is discussed separately with the following typesetting:

**Reviewer's comments**

Authors response

Changes in the manuscript

**I congratulate the authors for this nice work. Nevertheless, I am a bit surprised that the authors has not considered our previous works on this subject that could help them to improve their paper and to correct some flaws.**

We thank Dr. Jean-Baptiste Renard for his time reviewing our manuscript. It is great to have comments in the discussion phase because it is the way to improve our manuscript. We feel grateful for his comments and the acknowledge of our work. At the same time, we are sorry for not being cautious in acknowledging Dr. Renard's team experience in Saharan dust measurements. We have learnt a lot after reading some of their works that have served to enrich our manuscript. We believe that more interactions are needed between atmospheric scientists and the teams that perform laboratory measurements of particle phase matrixes.

**Line 63: For the size distribution, the authors could consider the measurements from balloon-borne aerosol counter inside a Saharan dust plume : Renard, J.-B.; Dulac, F.; Durand, P.; Bourgeois, Q.; Denjean, C.; Vignelles, D.; Couté, B.; Jeannot, M.; Verdier, N.; Mallet, M. In situ measurements of desert dust particles above the western Mediterranean Sea with the balloon-borne Light Optical Aerosol Counter/sizer (LOAC) during the ChArMEx campaign of summer 2013. Atmos. Chem. Phys. 2018, 18, 3677-3699.**

-We are grateful for this reference. After reading that manuscript, we believe that it is very important to remark on the possible presence of the super-coarse mode in the transported Saharan dust. This has been included in the new introduction section (L66-67):

"Mineral dust particles are typically considered as large particles in the coarse (1-10 µm) and super-coarse (> 10 µm) modes (Renard et al., 2018)"

And the reference Renard et al., 2018 has been added to the reference list

Renard, J. B., Dulac, F., Durand, P., Bourgeois, Q., Denjean, C., Vignelles, D., Couté, B., Jeannot, M., Verdier, N., & Mallet, M. (2018). In situ measurements of desert dust particles above the western Mediterranean Sea with the balloon-borne Light Optical Aerosol Counter/sizer (LOAC) during the ChArMEx campaign of summer 2013.

*Atmospheric Chemistry and Physics*, *18*(5), 3677–3699. https://doi.org/10.5194/acp-18-3677-2018

**Line 74: The authors have forgotten to consider the effect of the size distribution of the particles. Scattering properties (including polarization) are sensitive to the size of the particles, even for mineral dust, as shown by our team :**

-We agree with the referee, and we have re-written this part of the introduction section. Now it is given as (L119-122):

"… the usual study of scattering matrix elements of dust particles is done in the laboratory for synthetic samples minerals that compose dust particles (Curtis et al., 2008; Huang et al., 2020; Meland et al., 2010; Muñoz et al., 2010a; J. B. Renard et al., 2014; J.-B. Renard et al., 2010) or with collected dust samples (Muñoz et al., 2007a; J. B. Renard et al., 2014; J.-B. Renard et al., 2010, 2024)"

We have added these references to the list of references.

Renard, J. B., Hadamcik, E., Couté, B., Jeannot, M., & Levasseur-Regourd, A. C. (2014). Wavelength dependence of linear polarization in the visible and near infrared domain for large levitating grains (PROGRA2 instruments). *Journal of Quantitative Spectroscopy and Radiative Transfer*, *146*, 424–430. https://doi.org/10.1016/j.jqsrt.2014.02.024

Renard, J.-B., Francis, M., Hadamcik, E., Daugeron, D., Couté, B., Gaubicher, B., & Jeannot, M. (2010). Scattering properties of sands. 2. Results for sands from different origins. *Applied Optics*, *40*(18), 3352–3559. http://www.icare.univ-lille1.fr/progra2/

Renard, J.-B., Hadamcik, E., Worms, J.-C., & Hadamcik, E. (2024). The laboratory PROGRA2 database to interpret the linear polarization and brightness phase curves of light scattered by solid particles in clouds and layers. *Journal of Quantitative Spectroscopy and Radiative Transfer*, *320*, 108980. https://doi.org/10.1016/j.jqsrt.2024.108980ï

After receiving comments from all referees, we figured out that we might have written some inconsistent conclusions in our analysis. In the revised manuscript we have insisted that $F_{11}$ and $-F_{12}/F_{11}$ depend on several factors, particularly particle size distribution, shape and refractive index. It was not possible to perform inversions to retrieve size distribution and refractive index, and also unfortunately we did not have correlative measurements of size distributions with other instruments. Because of these reasons, we avoid speculations in the interpretation of the results, and we just highlight that the different patterns can be explained by different reasons.

We plan to continuously acquire measurements but with correlative measurements of aerosol size distributions and chemical compositions. We are also planning several field campaigns to do closure studies with lidar systems (see comments to referee 2). Moreover, we are collaborating with the GRASP team in the optimization of the inversion to retrieve several modes of aerosol optical and microphysical properties from our PI-Neph measurements. These issues are now clearer in the new conclusions section

**Lines 78-82: Why the large database of PROGRA2 (https://www.icare.univ-lille.fr/progra2-en/?noredirect=en_US) that contains such laboratory measurements is not cited?**

-We thank the referee for remarking on this point. We are impressed by the work done in PROGRA2 and it has been cited. Particularly, we have added reference Renard et al., (2024) to this part in the introduction section.

**Part 2.2.1. If I well understand, the instrument retrieves the light scattering parameter from a cloud of particles, without size selection. Thus, if the authors want to make comparison between different sessions of measurement in different conditions, they must be sure that the size distributions are the same. Otherwise, they will observe a combination of size distributions, refractive indexes, and even porosities of the grains and the agglomerates. The authors must explain how these different parameters influence their measurements and their analysis.**

-Yes, the instrument measured light scattered by an ensemble of particles that was sampled from a total air inlet, so there is no size selection or cutoff. We agree with the referee that for comparisons size distribution needs to be monitored and ideally controlled. But the main difference between PI-Neph measurements and those at laboratory is that the PI-Neph is acquiring ambient air and thus it implies changeable aerosol conditions. Our objective is to study phase matrix parameters, particularly $F_{11}$ and -$F_{12}/F_{11}$, in these changing conditions. This has been highlighted in the introduction section of the revised manuscript (L129-146):

"The latest developments use imaging techniques (Bian et al., 2017; Curtis et al., 2007; Dolgos & Martins, 2014a) to determine phase matrix with single detector and relatively compact design that does not require moveable parts. The Polarized Imaging Nephelometer (PI-Neph) was one of the first designs of a polar nephelometer that used imaging techniques, developed by the University of Maryland, Baltimore County (UMBC). This first prototype of the PI-Neph could acquire aerosol phase matrix at 473, 532 and 671 nm with 0.5º resolution. The instrument was deployed on the NASA DC8 aircraft and operated during special field (Espinosa et al., 2018; Reed Espinosa et al., 2017). Other PI-Neph instruments based on the first UMBC design are operated by NOAA (Ahern et al., 2022; Manfred et al., 2018). The main novelty of these prototypes is that they measure phase matrix elements of ambient air, when conditions can be very different to laboratory measurements. However, to date none of these instruments have been operating continuously and report any multiwavelength measurements of Saharan dust. The imaging technique is being expanded worldwide with further designs although limited to laboratory operation yet (Moallemi et al., 2023). All designs in polar nephelometry present physical limitations that limit the measurements to the range 3º-178º, but synthetic tests have revealed that multi-wavelength polarimetric PI-Neph measurements improve the information content for the retrieval of aerosol optical and microphysical properties (Moallemi et al., 2022) . Therefore, measurements of dust phase matrix elements for ambient aerosol samples in the atmosphere will serve to advance in the understanding of mineral dust absorption properties and chemical composition (Di Biagio et al., 2017, 2019)."

More specifically, our study focuses on how phase matrix elements for transported Saharan dust varies between two extreme events and other more moderate that reach our station in the southeast of the Iberian Peninsula. Polarization measurements seem to be very valuable when the aerosol sampled is equivalent to a mixture of particles of anthropogenic origin and those of mineral dust. We have clarified this point in the conclusions section, but adding that more measurements are required particularly at other locations (see comments to referee 2).

L772-789: "This work has focused on the analyses of aerosol phase matrix elements and other optical properties during Saharan dust outbreaks that were registered in the UGR station (Southeastern Spain) in the year 2022. The main novelty of the analyses are the measurements by the multiwavelength Polarized Imaging Nephelometer (PI-Neph) developed by GRASP-Earth and capable of providing two aerosol scattering matrix elements ($F_{11}$ and $-F_{12}/F_{11}$) for three different wavelengths (405, 515 and 660 nm). The uniqueness of PI-Neph is that it allows to measure phase matrix elements of ambient aerosol. The instrument can provide $F_{11}$ and $-F_{12}/F_{11}$ with 10% and 20% uncertainty, respectively, under laboratory conditions. The optimization of the instrument and the use of appropriate data quality check approach served to continuously measure $F_{11}$ and $-F_{12}/F_{11}$ for ambient air, but in these cases the natural variability of the air sampled typically imply large uncertainties, being the typical standard deviations of ~20% for $F_{11}$ and between 0.1- 0.2 for $-F_{12}/F_{11}$ and therefore larger than the uncertainties of the instrument. The multiwavelength $F_{11}$ and $-F_{12}/F_{11}$ measurements for different Saharan dust outbreaks are some of the first carried out for ambient aerosols and serve to complement laboratory measurements of mineral dust particles and of synthetic samples minerals that compose dust particles. The novel measurements of $F_{11}$ and $-F_{12}/F_{11}$ can also complement other optical and microphysical properties of Saharan dust already known from in-situ instrumentation and by active and passive remote sensing instruments, both from the ground and the space. Nevertheless, more $F_{11}$ and $-F_{12}/F_{11}$ measurements are needed at other experimental sites to have a more complete vision of mineral dust role on climate."

**The authors must consider some aerodynamic effects that can orient the particles during their motion in the instrument, and also the particles' speed. Such parameter can change the polarization results, as shown by our team : Daugeron, D.; Renard, J.-B.; Gaubicher, B.; Couté, B.; Hadamcik, E.; Gensdarmes, F.; Basso, G.; Fournier, C. Scattering properties of sands, 1. Comparison between different techniques of measurements, Applied Optics, 45, 8331-8337, 2006.**

-This is a very interesting effect that we have not considered in our measurements. However, considering the flowrate at which we measure, the speed of the particles is in the lower limit of the speeds explored in Daugeron et al., (2006), which led to less changes in the polarization results. Also, the particles we sample are not as large as the one measured in this work, so the effect will be even less evident. Thus, in our results we assume that particles are randomly oriented. Nevertheless, we think it is appropriate to clarify this point in the revised manuscript (L183-185):

"Given the flowrate used in the measurements we can assume that particles are randomly oriented, avoiding the limitations in polarization results of super-coarse particles with

particle speed that can orient the particle in particular orientations (Daugeron et al., 2006)."

And the reference Daugeron et al., (2006) have been added to the reference list

Daugeron, D., Renard, J.-B., Gaubicher, B., Couté, B., Hadamcik, E., Gensdarmes, F., Basso, G., & Fournier, C. (2006). Scattering properties of sands. 1. Comparison between different techniques of measurements. *Applied Optics*, *45*(32), 8331–8337.

**Figures 6, 7 and 9, and in text : The negative polarization at the large angles for the blue domain only is strange, and perhaps suspicious. Such high negative values were never observed in laboratory (and even in space) for randomly oriented compact irregular particles. More strangely, such phenomenon does not occur at the other wavelengths. The authors must explain the origin off such negative values (that in fact are almost impossible for such large dust particles). I suspect an instrumental error like a stray light contamination not accurately removed, a too log signal to noise ratio ....**

We understand the referee concerns about the pattern observed in 405 nm. That was exhaustively checked by our team, and we are confident in our measurements. The possible influence of straight light was evaluated through the validation of the instrument using monodisperse aerosol (PSL) measurements in laboratory. Results are in Bazo et al., (2024). In particular, we used measurements of PSL of 0.75 μm and 1 μm. Figures below (Figure 10 in Bazo et al., (2024)) show experimental data and fits with the GRASP algorithm. Very good agreements between measured and theoretical/retrieved scattering matrix were obtained. Generally, the agreements in $F_{11}$ are excellent, reproducing all resonances, while for $-F_{12}/F_{11}$ there are some deviations. If we assume that errors in the instruments under controlled laboratory conditions are of 5% and 10% for $F_{11}$ and $-F_{12}/F_{11}$ (associated with issues of calibration and data acquisition of the CMOS plus other errors associated with the imperfections and preparations of the PSL sample), then the differences observed in the Figure can be explained by errors propagation. The agreement between measured and retrieved values is supported by high correlation coefficients and low residuals. GRASP retrievals were able to reproduce particle diameters and refractive index. Similar results were obtained when using Mie code instead of GRASP algorithm.

All these issues were remarked in the revised section 2.1 related to instrument description (L213-215):

"The evaluation of the instrument versus known scattered (monodisperse polystyrene spheres - PSL) showed good agreements with RMSE around 0.10 for both $F_{11}$ and $-F_{12}/F_{11}$."

The other point that makes us believe that our measurements are right is the temporal evolutions observe during the extreme Saharan dust outbreaks – Figures are given in the supplementary material. During the peaks of these intrusions the 405 nm pattern in $-F_{12}/F_{11}$ is the same as observed for the other two channels. However, negative values in $-F_{12}/F_{11}$ are observed in the moments prior and posterior to the Saharan dust advection. If the negative value is an artifact, then it should be present at any moment.

[Figure]

Figure 10 of Bazo et al., (2024). PSL PI-Neph measurements of $F_{11}$ (left) and $-F_{12}/F_{11}$ (right) and GRASP fits for 0.75 μm (top) and 1 μm (bottom).

Finally, we performed computations of $F_{11}$ and $-F_{12}/F_{11}$ using GRASP for a bimodal size distribution with different complex refractive indexes between the two modes. Particularly, we used bi-lognormal size distribution, one representative of fine mode particles with modal radius of 0.15 μm and 0.25 μm of standard deviation and the other representative of coarse mode particles with modal radius of 2.5 μm and 1 μm of standard deviation. Real refractive indexes were assumed non-spectrally dependent with values of 1.6 for fine mode and 1.55 for coarse mode. Imaginary refractive indexes were 0.0015 for fine mode and with no spectral dependency, while for coarse mode they were of 0.007, 0.005 and 0.005 for 405, 515 and 660 nm. The sphere fraction was also fixed for each mode, being 0.7 for fine mode and 0.05 for coarse mode. Three different scenarios were generated giving different weights to each mode: The first is for volume concentrations of 0.3 μm³/μm³ for each mode and can be considered as representative of a mixed case where both modes have a similar weight. The second presents more predominance of coarse mode (volume concentrations of 0.5 μm³/μm³) but with non-negligible contribution of anthropogenic particles (volume concentrations of 0.1 μm³/μm³) in the fine mode. The last scenario is representative of pure dust mode (volume concentrations of 0.5 μm³/μm³) with negligible fine mode contribution (volume concentrations of 0.01 μm³/μm³).

[Figure]

Figure R1. Volume size distribution used for the GRASP simulations of the 'Urban mixture' case.

[Figure]

Figure 12 in the manuscript. Simulations of phase function ($F_{11}$) and polarized phase function (-$F_{12}/F_{11}$) using GRASP forward for three different combinations of bi-lognormal size distributions. *Urban Mixture* with approximately the same weight of fine and coarse mode, *Dust mixed* with predominance of coarse mode but with a non-negligible influence of the fine mode, and *Pure Dust* with strong predominance of coarse mode and negligible fine mode. Note that refractive indexes and sphericity of each mode are different.

The feature of negative values in -$F_{12}/F_{11}$ for the 405 nm channel is also observed in the GRASP simulations. We are aware that the negative polarization is not as high as in our measurements, but further optimizations in GRASP are still needed (i.e. implementation of super-coarse mode). Nevertheless, GRASP simulations support the possible existence of negative values in -$F_{12}/F_{11}$ in the 405 nm for a mixture of dust and anthropogenic particles. These points have been added to the conclusions section (L849-860):

"Simulations performed by the GRASP code for different mixtures of fine mode (anthropogenic particles) and coarse mode (dust particles) revealed that $F_{11}$ and -$F_{12}/F_{11}$ are sensitive to the different contribution of each mode in the mixture, being especially critical for -$F_{12}/F_{11}$ on the 405 nm channel. The negative values for -$F_{12}/F_{11}$ in 405 nm were observed more clearly for the mixture of fine and coarse particles. Thus, these simulations have served to understand the experimental negative values in -$F_{12}/F_{11}$ not observed in laboratory measurements for collected dust. Retrievals of bimodal size distribution with separate refractive indexes for each mode would have shown clarity to this problem. Hower, such retrieval with GRASP using $F_{11}$ and -$F_{12}/F_{11}$ as inputs need to be optimized. Another additional optimization in GRASP will imply the possibility of implementing the retrieval of super-coarse mode particles. Nevertheless, the possibility of explaining the spectral differences in $F_{11}$ and -$F_{12}/F_{11}$ with wavelength has served to understand the temporal evolution of the extreme dust events and the difference and similitudes when comparing versus laboratory measurements and versus other more moderate events of Saharan dust transport."

**Line 389: The wavelength effect was largely studied in the PROGRA2 data base and largely published by our team.**

-Thanks for the suggestion. We have added the reference Renard et al., (2014) in this part of the result discussion.

**Line 465: Aging can produce more compact particles, but it is difficult to call them "spherical".**

-Thanks for pointing this out. Referee 2 raised several issues in this paragraph, and thus we have re-written the paragraph. The sentence '*The values of g suggest the higher predominance of spherical particles when compared to the other cases, so it is likely that the urban background particles suffered from aging*' has been removed. Now, we only remark that the values of g suggest the lower contribution of dust particles.

**Line 485: This analysis could be inaccurate. Other effects than the sphericity of the particles must be considered (size, refractive index, porosity). The presence of "spherical particles" do not change significantly the shape of the light scattering curves (of course, only a medium composed of perfect spheres can change the shape of the curve).**

-We agree with the referee and we have removed the second part of this sentence.

**Figure 11: Yellow dots are difficult to see.**

-Thanks for the suggestion. We have changed the colors of the dots to light blue for better visualization.

**Line 620: Do they authors have also considered the super-coarse mode of particles ? Keep in mind that the largest particles are the most luminous, and thus can dominate the scattered (polarized) intensities.**

-We have not considered super-coarse particles for the simulations performed with GRASP. To our knowledge, the super-coarse model is not available in the latest version

of GRASP. The objective of the simulations is to show the variability in $F_{11}$ and $-F_{12}/F_{11}$ with the mixtures of dust and anthropogenic pollution.

We agree with the referee that the super-coarse mode can be presented in the long-range Saharan dust transport. For the extreme event on 15th-16th March 2022 we measured the size distribution for the deposited dust (Figure R2) and the presence of super-coarse particles in the sample is low compared to the fine and coarse modes. Additionally, the modal radius of the coarse mode agrees with the aerosol size distribution used in GRASP simulations. This has been added to the discussion in the revised manuscript (L730-731).

"The modal radii selected are close to those observed for the particle size distribution of the deposited particles (not shown for clarity)."

However, as the referee suggests, the residual super-coarse mode can have an impact on the phase matrix. This has been remarked in the revised manuscript. This super-coarse mode needs to be also implemented in GRASP. We have added the following sentence to the manuscript (L768-770):

"It is important to mention that the super-coarse mode can also affect the behavior of $F_{11}$ and $-F_{12}/F_{11}$ and the presence of this mode is also observed for long-range transport (i.e. Renard et al., 2010). Future GRASP developments also need the consideration of this super-coarse mode."

[Figure]

Figure R2. Number size distribution measured for the deposited dust for the extreme dust event on 15th – 16th March 2022.

**Figure 12: Negative polarization down to -0.4 are unrealistic for dust sample, unless the size distribution is dominated by submicron particles without large ones.**

-Figure 12 shows the simulations with GRASP using a bi-modal size distribution with different refractive indexes (see previous comments). The objective was to show $F_{11}$ and $-F_{12}/F_{11}$ for different mixtures of dust with anthropogenic pollution. In our station, background conditions are dominated by anthropogenic particles. During a Saharan dust intrusion, the dust particles mixture with these background particles lead to different mixtures of particles that ultimately were measured by the PI-Neph. The simulations show that the presence of anthropogenic particles affect $F_{11}$ and $-F_{12}/F_{11}$ when compared with

the case of only dust particles. This last point is more relevant for $-F_{12}/F_{11}$ in the 405 nm channel and serves to understand the behaviors observed in Figure 11 and in the temporal evolutions during the extreme dust event given in the Supplementary Material. The text has been accordingly modified (L758-762):

"Figure 12 results show how the presence of anthropogenic particles (fine mode) can alter the spectral dependencies in $-F_{12}/F_{11}$ when compared with only dust particles (coarse mode) in the sample, particularly in the blue channels. However, changes in the $F_{11}$ patterns were not so evident. These results help to understand the different phase matrix elements discussed in this manuscript, and their temporal evolutions during the extreme dust events (Supplementary Material)."

---

## Author Response (AR2)

**Reviewer 1**

**In their revision the authors have addressed the majority of my prior concerns. The claims that in my view were not well justified in the initial submission have mostly been softened or better supported. While uncertainties tailored to ambient measurements would have been ideal, I appreciate the challenge in doing so and approximate error estimates have now been included based on the Bazo et al. (2024) paper studying PIN100 performance in the lab which is satisfactory. There are however several points below for which I am still not following and/or believe need correction. In my opinion, the manuscript would be suitable for publication once these have been addressed.**

We thank the reviewer for acknowledging the first review process by the authors, it has indeed helped to improve the quality of the manuscript. We proceed to answer each comment raised by the reviewer below.

**GENERAL COMMENTS**

**<Regarding response to my initial comment on sources of key optical quantities>**

**Horvath (2015) proposes a stepwise extrapolation technique that relies on fits of log(P11) vs scattering angle and shows, that when this particular technique is used, errors in integrals over the full range of scattering angles (0-180°) are less than 5% for size parameters x≤15. However, the present manuscript simply says that regular linear extrapolation is used to fill in the truncated regions and also does not discuss the possibility of particles with size parameters greater than 15. This should be addressed.**

- The reviewer is right, the extrapolation technique used in this work is not exactly the same as the one described in Horvath, (2015). Indeed, we have also done the stepwise extrapolation and compared it to our initial extrapolations. We have obtained differences in total scattering coefficients below 1% between both approaches, so we believe that we can assume that the errors of both extrapolation techniques are equivalent. According to Horvath, (2015): "*Using the stepwise extrapolation, the total scattering coefficient, the asymmetry parameter and the fraction of backscattered light can be attained with an accuracy far below 1%.*". Thus, for integrated values such as extinction coefficient, BS, SSA and asymmetry parameter errors are below 5%.

Horvath, (2015) also states "*for a particle sizes above 4 µm the deviations increase (although still in a range of <5%).*" These particles of around 4 µm in diameter would have size parameters between 19 and 31 in our wavelengths. According to Horvath, (2015), the extrapolation technique when these particles are present would still have errors below 5%.

To clarify these points, modifications in the text have been made (Lines 235-237)

"Stepwise extrapolations might be more consistent (i.e. Horvath, 2015), but our additional computations remarked that differences between linear and stepwise extrapolations were below 1% for $\sigma_{sca}$, g and $B_s$. For particle sizes above 4 µm the uncertainties can yield 5% (Horvath, 2015)."

**I'm also wondering if an "illustration" of lidar ratio values is needed here. In my view, it would be much better to leave lidar ratios entirely out of the manuscript since they are determined primarily by extrapolation over an extremely variable region of F11 and are thus potentially subject to large very errors.**

We appreciate referee suggestions. We think that our current analysis serves to illustrate the variability of lidar ratio, and also to highlight the difficulty of obtaining this parameter from polar nephelometry measurements. We have highlighted these issues in the revised manuscript

**<Regarding response to my initial comment on unrealistic/nonphysical -F12/F11>**

**If I understand correctly, the authors are saying that the unrealistically low values of P12 are due to variability in the sample. I'm not completely following this explanation though. Each individual sample should still have physically plausible values of F12, even if the underlying aerosol was changing from one measurement sample to the next. And averaging many samples inside the physically plausible range could never produces means falling outside of that range.**

We understand referee concerns and to clarify these issues we need to discuss details of how the instrument performs during dust measurements: When dealing with big particles forward scattering strongly predominates. That implies a challenge for PI-Neph measurements because it might saturate some angles at forward scattering and undersaturated others in the backward scattering. During the extreme events in March 2022, we had even to low camera gain to avoid supersaturations – which later implied specific calibrations. For the rest of the data series it didn´t happen and the camera gain was higher and kept constant. Calibration was specifically checked – see Bazo et al., (2023) for details. These specific modifications in the gain´s camera help to get appropriate measurements in all angles. Nevertheless, during the measurement process some saturations/undersaturation still happened but the data quality process that we followed was able to reject all saturated/undersaturated data. For the specific phase function measurements where many data points are rejected, all angles are eliminated because phase matrix measurements were considered non-reliable.

The determination of $F_{12}$ is done by subtracting parallel and perpendicular phase function measurements and later dividing by $F_{11}$, which is computed as the sum of both components. The backward scattering regions are more sensitive to variations in the phase functions, and typically the lower signals are registered (although still they are not undersaturated). Also $F_{11}$ minimums are founds in these regions. This can create a mathematical artifact that enhances variations and some outliers that even do not have physical meaning.

Nevertheless, we do not show hourly averages that fall outside of the physical range between 1 and -1. It is true that blue channel shows a non-typical pattern with low values of -$F_{12}/F_{11}$ in the range 100-150º for 405 nm, but studying the evolution of the extreme dust event a smooth variation is observed. In fact, the non-typical pattern in -$F_{12}/F_{11}$ at 405 nm is observed just before the event arrives and when it is withdrawing, while during the peaks of the events -$F_{12}/F_{11}$ at 405 nm follows the classical pattern measured in laboratory with pure dust samples. The PI-Neph is validated versus laboratory measurements using PSL (see Bazo et al., 2023 for details) plus the fact that the pattern at 405 nm is not always present make us trust our measurements, despite the noise.

To clarify all these points, we have modified the manuscript (Lines 413-428):

"Phase matrix elements were exhaustively monitored with the use of the PI-Neph during both extreme events. Given the high concentrations of large particles, the usual configuration of the measurements could lead to saturation of many angles in the forward scattering. Therefore, it was necessary to reduce the gain of the PI-Neph's camera, changing the dynamic range of the camera for obtaining non-saturated measurements at such high concentrations. But these

changes were made to also guarantee enough signals in the backward region where the minimums are found. Nevertheless, sporadic pixels might present saturation/low SNR at some angles, but they were filtered out by the data quality criterion for the instrument (Section 2.2.1). For the specific phase function measurements where many data points are rejected, all angles are eliminated because phase matrix measurements were considered non-reliable. Moreover, -$F_{12}$/$F_{11}$ is computed by subtracting first parallel and horizontal phase function, that in many angles are very similar. Dividing this small number by $F_{11}$ can enhance the differences, being particularly critical in the angular regions where the minimum values of scattering are found (typically between 90-150º for dust particles). All these effects, although they are always present, imply larger random noise in the measurements. Actually, noisier patterns in measured $F_{11}$ and -$F_{12}$/$F_{11}$ affected by large particles when compared with measurements of anthropogenic origin has been already reported by the first versions of PI-Neph in the United States (Espinosa et al., 2018).”

**I'd also be curious to hear the authors thoughts on why the aerosol was varying so drastically from one measurement to the next. I would expect the properties of an aerosol that has traveled many 100's of km (Sahara to Grenada) to be relatively homogenous.**

The supplementary material shows the hourly evolution of the $F_{11}$ and -$F_{12}$/$F_{11}$ during the two extreme Saharan dust events in March 2022. Data temporal resolution is of 1 hour. In the supplementary material we observe a smooth variation of all phase matrix elements. The main changes are observed in -$F_{12}$/$F_{11}$ at 405 nm but in our opinion the figure reveals a smooth change. We associated the variations in -$F_{12}$/$F_{11}$ at 405 nm with the possible influence of other anthropogenic particles. During the peaks of extreme events such possible background anthropogenic particles seem to not have an impact on the mixture, while that does not happen before and after the peaks of the event.

For the analyses of the entire dataset where more common Saharan dust events reach the stations – none of them is classified as extreme – the non-typical pattern in -$F_{12}$/$F_{11}$ at 405 nm is mostly present. This is supported by measurements of additional instrumentation that classified the aerosol sample as a mixture of dust with anthropogenic particles (Figure 10). Also, from the seasonal analyses the pattern observed for -$F_{12}$/$F_{11}$ at 405 nm presents the lowest standard deviations (Figure 11). Simulations with GRASP help to support how those mixtures of dust with anthropogenic particles can alter the shape and spectral dependences of -$F_{12}$/$F_{11}$.

We believe that further studies are needed. On the one hand, more measurements at other locations will help to understand the different patterns of -$F_{12}$/$F_{11}$ when influenced by Saharan dust particles. Also, further developments of GRASP will help to understand aerosol optical and microphysical properties differentiating between fine and coarse mode. Further studies with other modelling approaches for the scattering of big and non-spherical particles are also required (i.e. spheres, spheroids and irregular-hexahedral). We have slightly modified the text accordingly (Lines 795-797):

“Another issue to study is the use of irregular-hexahedral for modeling the scattering of large and non-spherical particles that might reproduce better polarization signals (Saito and Yang, 2021, 2023; Saito et al., 2021).”

And in the Conclusions section (Lines 883-884)

The possibility of implementing the irregular-hexahedral model would be also ideal to better understand polarization patterns.

The following references have been added:

Saito, M., and Yang, P. (2021): Advanced bulk optical models linking the backscattering and microphysical properties of mineral dust, *Geophysical Research Letters*, *48*, e2021GL095121. https://doi.org/10.1029/2021GL095121

Saito, M., Yang, P., Ding, J., and Liu, X. (2021): A comprehensive database of the optical properties of irregular aerosol particles for radiative transfer simulations. Journal of the Atmospheric Sciences, 78, 2089-2111.

Saito, M., & Yang, P. (2023). Quantifying the impact of the surface roughness of hexagonal ice crystals on backscattering properties for lidar-based remote sensing applications. *Geophysical Research Letters*, *50*, e2023GL104175. https://doi. org/10.1029/2023GL104175.

**SPECIFIC COMMENTS (LN # links to the first submission of the manuscript)**

**LN 284: While Nousiainen & Kandler (2015) state several times that P11(180°) is lower in nonspherical particles, I could not find anything in that review supporting the idea that the total integral of P11 over the range 90° to 180° is lower in nonspherical particles. P11 generally increases with nonsphericity in the angular range of 90° to ~150° and this can cancel out the higher P11 values found in spheres at the more extreme backscattering angles (θ>150°). See for example the P11 panel of Figure 3 in Zhou et al. (2020).**

- According to Horvath et al., (2018): *"The scattering function of the desert aerosol has a low back scattering, which is typical for non-spherical particles. Therefore it is to be expected that a characterization of desert aerosol particles could be achieved by considering the fraction of backscattered light. It is defined as the ratio of the integral of the volume scattering function between 90 and 180° divided by the integral over the full angle and is readily available once the volume scattering function is known.* […] *the backscattered fraction obtained from polar nephelometer measurements is lower for the aerosol dominated by desert particles, as expected."*

Therefore, we believe that our statement is supported. We have changed the reference Nousiainen and Kandler, (2015) for the reference Horvath et al., (2018) in this statement.

**Figure 5: To avoid confusion, if the Figure R1 version is not to be used in the final manuscript, I would at least suggest including a mention of the missing data to avoid confusion.**

- The reviewer is right. We have added the following sentence between lines 360-361: "Note that Panel (a) in Figure 5 does not cover the beginning of the Saharan dust event due to lack of data related to supersaturation of the PI-Neph's measurements."

**LN 504/707: I'm still not following what is meant by "large variability". Is this referring to the changes in P12/P11 on the order of ±0.2 that vary randomly and show little correlation over lengths greater than a few degrees in the red and green channels? Those variations are clearly nonphysical instrument artifact, likely stemming from lower aerosol concentration which result in low signal–to-noise. The dust measurements shown in Espinosa et al. (2018) – for which a citation was added to the text here – are also the result of instrument artifacts and do not represent a physical characteristic of the aerosol. This should be made clear.**

We understand referee concerns. Apart from the natural variability of the sampled aerosol, measurements of phase matrix for large and non-spherical particles face additional challenges due to instrumental and physical limitations that ultimately might imply larger variability. On the one hand, these large and non-spherical particles present maximum scattering in the forward regions and minimum in the backward region, particularly in the region 90-150º. Thus, there are strong differences in the measured signals between both regions. As the referee comments, these issues were already raised in Espinosa et al., (2018). In our study, for the extreme Saharan dust events we had even to change the camera gain to avoid saturations in the forward scattering region. Nevertheless, we guarantee that the measurements presented are in the lineal dynamic range of the detector, filtering out saturations and under-saturations. On the other hand, we face limitations that the computation of $-F_{12}/F_{11}$ imposes. First, $F_{12}$ is computed by subtracting horizontal and vertical phase function measurements that can be very similar for the case of dust. For the scattering region where minimums are found (90-150º), signals are weaker and thus are even more sensitive to errors in the measurements. This can be even enhanced when divided by $F_{11}$. Although these difficulties in the measurement of phase matrix for dust particles are systematic, they are ultimately traduced in random errors. In the revised manuscript, we have made all these points clearer (Lines 413-428):

"Phase matrix elements were exhaustively monitored with the use of the PI-Neph during both extreme events. Given the high concentrations of large particles, the usual configuration of the measurements could lead to saturation of many angles in the forward scattering. Therefore, it was necessary to reduce the gain of the PI-Neph's camera, changing the dynamic range of the camera for obtaining non-saturated measurements at such high concentrations. But these changes were made to also guarantee enough signals in the backward region where the minimums are found. Nevertheless, sporadic pixels might present saturation/low SNR at some angles, but they were filtered out by the data quality criterion for the instrument (Section 2.2.1). For the specific phase function measurements where many data points are rejected, all angles are eliminated because phase matrix measurements were considered non-reliable. Moreover, - $F_{12}/F_{11}$ is computed by subtracting first parallel and perpendicular phase functions, that in many angles are very similar. Dividing this small number by $F_{11}$ can enhance the differences, being particularly critical in the angular regions where the minimum values of scattering are found (typically between 90-150º for dust particles). All these effects, although they are always present, imply larger random noise in the measurements. Actually, noisier patterns in measured $F_{11}$ and $-F_{12}/F_{11}$ affected by large particles when compared with measurements of anthropogenic origin have been already reported by the first versions of PI-Neph in the United States (Espinosa et al., 2018)."

And we have included slight modifications in the discussions of the uncertainties. Now, between Lines 432-434 in the discussion of Figure 6 uncertainties.

"Note that the large standard deviation can be explained by the specific issues for the measurements of dust particles commented above".

In the discussion of Figure 7 uncertainties (Lines 482-484)

"Again, the specific issues related to the measurements of large particles can explain the deviations. Nevertheless, the larger deviations when compared with the previous events make us think that during this event there were more aerosol variability that is critical for the regions of the minimums in scattering."

In the discussion of uncertainties for Figure 9 (Lines 590-592)

"The standard deviations are larger (30% in in $F_{11}$ and around 0.2 in -$F_{12}/F_{11}$) when compared with the extreme events, and despite the inherent issues in the measurement of phase matrix for big particles, it seems that the sample presents a more complex mixture with more variability during the 1-hour average."

And finally in the discussion of Figure 11 (Lines 706-713)

"Particularly, for 660 and 515 nm large standard deviations are found in the region between 50º-150º while for 405 nm the standard deviations are considerably lower. Apart of the inherent limitations in the measurements in this scattering range, the results suggests that these -$F_{12}/F_{11}$ values at 660 and 515 nm are very sensitive to changing conditions in the aerosol that is sampled. Moreover, the other region that presents remarkable standard deviations for all wavelengths is the region of scattering angles above 170º. Those regions with large standard deviations are very sensitive to any change in particle type and size, which was demonstrated both from theoretical computations (Mischenko et al., 2002) and in laboratory measurements (Gomez-Martin et al., 2021)".

**REFERENCES**

**Espinosa, W. Reed, et al. "Retrievals of aerosol optical and microphysical properties from Imaging Polar Nephelometer scattering measurements." Atmospheric Measurement Techniques 10.3 (2017): 811-824.**

**Espinosa, W. Reed, et al. "Retrievals of aerosol size distribution, spherical fraction, and complex refractive index from airborne in situ angular light scattering and absorption measurements." Journal of Geophysical Research: Atmospheres 124.14 (2019): 7997-8024.**

**Zhou, Y., Levy, R. C., Remer, L. A., Mattoo, S., & Espinosa, W. R. (2020). Dust aerosol retrieval over the oceans with the MODIS/VIIRS dark target algorithm: 2. Nonspherical dust model. Earth and Space Science, 7, e2020EA001222. https://doi.org/10.1029/2020EA001222**

**Reviewer 2**

**Thank you for your detailed response to my comments. You could convince me that the lidar data for the specific dust outbreaks do not add further information to your study. And I hope that in future you'll find some nice comparisons of the lidar ratio at ground and in the lowest altitudes of the lidar observations. It is good that you have the lidar experts at the same university. Just a side note: Please take care with the altitude above sea level (SSC output) and the lidar data which are usually reported above ground level.**

- We thank the referee for the comments and understandings, they have served to improve our knowledge and the manuscript. As stated in the first review process, there have been some intensive field campaigns to achieve this comparison between polar nephelometry and lidar. We hope to obtain results soon.

**It is also sad that no size distributions were available for your observations. For an ACP publication I would have loved to see them, because the scope of the journal aims for a more holistic view on the single observations. That is why I just could mark "fair" for the scientific significance. Nevertheless, the article should be published. I strongly support your request to get bimodal size distribution which differentiate the optical properties (especially refractive index) between fine and coarse modes. Ideally, you would wait with the publication until this issue is solved to have a more complete modeling part in your study. However, I don't want to further postpone the publication and agree to do it without.**

- We thank the referee for supporting the publication of our manuscript. We are working hard in collaboration with GRASP developers to optimize the retrievals. We are also working in evaluating the irregular-hexahedral model in the retrievals, which is time consuming

**Concerning the uncertainties in Fig. 6, 7 & 9: Even if it just represents the atmospheric variability within the one hour of measurements, these error bars are important. In your Fig. R5, one sees that the results in the bottom left figure are clearly separated, whereas in the bottom center figure the variability is larger and the results are more similar. I would recommend to show these uncertainties in the paper as well and not only in the response to the reviewers. Please add these uncertainties to your 3 figures.**

- We thank referee suggestions. In the revised manuscript we have added the error bars representing the standard deviation to Figures 6, 7 and 9.

**Technical corrections:**

**Some new references in the introduction contain some letters of the surname, e.g., J. B. Renard or Reed Espinoza. It should not be the case.**

**EARLINET/ACTRIS – abbreviations should be written out once.**

We thank the referee for pointing out these issues. Corrections have been added to the manuscript. Between Lines 239-240:

"Aerosols, Clouds and Trace gases Research Infrastructure (ACTRIS; https://www.actris.eu/)."

And between Lines 338-339:

"lidar measurements in AGORA in the framework of the European Aerosol Research Lidar Network (EARLINET; https://www.earlinet.org/ )"

---

## Author Response (AR3)

Dear Editor,

We acknowledge your efforts in editing our manuscript. Attached is the final version with a reduced abstract (259 words) and more concise conclusion sections

Sincerely

---

## Author Response (AR4)

Response to Editor

The authors greatly acknowledge the editor for carefully reading the manuscript and providing constructive comments. This document contains the author's responses.

**Editor's comments**

Authors response

Changes in the manuscript

**Thank you for providing replies to the second round of Referee comments. I understand that you have addressed all their concerns and hope that they are satisfied as well. I am thus happy to accept your work for publication in ACP. However, please take the time to consult the ACP guidelines regarding Abstract and Conclusions before uploading the final version of your manuscript (https://www.atmospheric-chemistry-and-hysics.net/policies/guidelines_for_authors.html)**

We have reduced the abstract to approximately 250 words, and now it is given by

This work investigates the scattering matrix elements during different Saharan dust outbreaks over Granada (South-East Spain) in 2022 using a Polarized Imaging Nephelometer (PI-Neph) capable of measuring continuously the phase function ($F_{11}$) and the polarized phase function ($-F_{12}/F_{11}$) at three different wavelengths (405, 515 and 660 nm) in the range 5° - 175°. The focus is in two extreme dust events ($PM_{10}$ > 1000 $\mu gm^{-3}$) in March 2022. During the peaks of these events $F_{11}$ and $-F_{12}/F_{11}$ show the classical patterns observed for dust samples in laboratory measurements available in the Amsterdam-Granada Light Scattering database at all wavelengths. However, for the moments prior and after the peaks the results reveal important sensitivity in $-F_{12}/F_{11}$ at 405 nm. For the other wavelengths, however, this difference in $-F_{12}/F_{11}$ is not evident. Moreover, no remarkable changes are found in $F_{11}$ that is always characterized by strong predominance of forward scattering. The analyses of more frequent and moderate events registered in summer 2022 ($PM_{10}$ between 50 and 100 $\mu gm^{-3}$) revealed $F_{11}$ and $-F_{12}/F_{11}$ patterns like those observed prior and after the extreme events. The combination of PI-Neph measurements with additional in-situ instrumentation allowed a typing classification that revealed the peaks in the extreme dust events as pure dust, while for the rest of cases remarked a mixture of dust with urban background pollution. In addition, simulations with the Generalized Retrieval of Atmosphere and Surface Properties (GRASP) code explain the different patterns in $-F_{12}/F_{11}$ with changes in the refractive indexes and with the different contributions of the fine and coarse mode.

Additionally, we have shortened the conclusions section, keeping focus in the most relevant results and findings (Lines 789 - 858)

[revised manuscript text omitted]

**Additional comments\***

There were some technical comments that we apologize for not addressing before. One was related to the inclusion of the "competing interest section". Now it is given between Lines 859-860

**Competing interests**

The authors declare that they have no conflict of interest.

Also, we were required to ensure that the colour schemes used in our maps and charts allow readers with colour vision deficiencies to correctly interpret your findings. We were not aware of this and we apologize for not following the guidelines We strongly support this way to make the manuscript more accessible to colour vision deficiencies people. We check Fig. 6, 7 and 8 using the Coblis – Color Blindness Simulator (https://www.color-blindness.com/coblis-color-blindness-simulator/) and revise the colour schemes accordingly.

[Figure]

**Figure 6. Hourly averages of phase function ($F_{11}$) and polarized phase function ($-F_{12}/F_{11}$) on 15th - 16th March 2022 for four different stages of the evolution of the extreme Saharan dust outbreak: (a) 15th March 07:00 UTC before the Saharan dust outbreak reached the station, (b) 15th March 12:00 UTC when the Saharan dust begins to reach the station, (c) 15th March 17:00 UTC associated with the peak of the extreme Saharan dust intrusion, and (d) 16th March 13:00 UTC when Saharan dust starts to withdrawn. Error bars correspond to the standard deviation of the hourly averages.**

[Figure]

**Figure 7. Hourly averages of phase function ($F_{11}$) and polarized phase function ($-F_{12}/F_{11}$) on 24th - 25th March 2022 for four different stages of the evolution of the extreme Saharan dust outbreak: (a) 24th March 13:00 UTC before the Saharan dust outbreak reached the station, (b) 24th March 21:00 UTC when the Saharan dust starts to reach the station, (c) 25th March 09:00 UTC associated with the peak of the extreme Saharan dust outbreak, and 25th March 20:00 UTC when dust begins to withdrawn. Error bars correspond to the standard deviation of the hourly averages.**

[Figure]

**Figure 9. Phase function ($F_{11}$) and polarized phase function ($-F_{12}/F_{11}$) (for different moderate dust events: 15th April 202, 25th July 2022 and 30th August 2022. Error bars correspond to the standard deviation of the hourly averages.**